# Recent dynamic changes on Fleming Glacier after the disintegration of Wordie Ice Shelf, Antarctic Peninsula

Peter Friedl[1], Thorsten C. Seehaus[2], Anja Wendt[3], Matthias H. Braun[2], Kathrin Höppner[1]

[1]German Remote Sensing Data Center (DFD), German Aerospace Center (DLR), Oberpfaffenhofen, 82234, Germany
[2]Institute of Geography, Friedrich-Alexander-University Erlangen-Nuremberg, Erlangen, 91058, Germany
[3]Bavarian Academy of Sciences and Humanities, Munich, 80539, Germany

*Correspondence to*: Peter Friedl (peter.friedl@dlr.de)

**Abstract.** The Antarctic Peninsula is one of the world`s regions most affected by climate change. Several ice shelves retreated, thinned or completely disintegrated during recent decades, leading to acceleration and increased calving of their tributary glaciers. Wordie Ice Shelf, located in Marguerite Bay at the south-western side of the Antarctic Peninsula, completely disintegrated in a series of events between the 1960s and the late 1990s. We investigate the long-term dynamics (1994–2016) of Fleming Glacier after the disintegration of Wordie Ice Shelf by analysing various multi-sensor remote sensing datasets. We present a dense time series of Synthetic Aperture Radar (SAR) surface velocities that reveals a rapid acceleration of Fleming Glacier in 2008 and a phase of further gradual acceleration and upstream propagation of high velocities in 2010–2011.The timing in acceleration correlates with strong upwelling events of warm Circumpolar Deep Water (CDW) into Wordie Bay, most likely leading to increased submarine melt. This, together with continuous dynamic thinning and a deep subglacial trough with a retrograde bed slope close to the terminus probably has induced unpinning of the glacier tongue in 2008 and gradual grounding line retreat between 2010 and 2011. Our data suggest that the glacier`s grounding line had retreated by ~6–9 km between 1996 and 2011, which caused ~56 km$^2$ of the glacier tongue to go afloat. The resulting reduction in buttressing explains a median speedup of ~1.3 m d$^{-1}$ (~27 %) between 2008 and 2011, which we observed along a centreline extending between the grounding line in 1996 and ~16 km upstream. Current median ice thinning rates (2011–2014) along profiles in areas below 1000 m altitude range between ~2.6 to 3.2 m a$^{-1}$ and are ~70 % higher than between 2004 and 2008. Our study shows that Fleming Glacier is far away from approaching a new equilibrium and that the glacier dynamics are not primarily controlled by the loss of the former ice shelf anymore. Currently, the tongue of Fleming Glacier is grounded in a zone of bedrock elevation between ~-400 and -500 m. However, about 3–4 km upstream modelled bedrock topography indicates a retrograde bed which transitions into a deep trough of up to ~-1100 m at ~10 km upstream. Hence, this endangers upstream ice masses, which can significantly increase the contribution of Fleming Glacier to sea level rise in the future.

# 1 Introduction

Recent studies have shown that the Antarctic Peninsula ice masses are strong contributors to sea level rise. In a consolidated effort Shepherd et al. (2012) estimated the contribution between 2005 and 2010 to 36 ± 10 Gt a$^{-1}$ corresponding to 0.1 ± 0.03 mm a$^{-1}$ SLE (sea level equivalent). This is considerably higher than their reported ice mass loss for the period from 1992 to 2000 of 8 ± 17 Gt a$^{-1}$ (Shepherd et al., 2012). Huss and Farinotti (2014) computed from their ice thickness reconstruction of the northern and central Antarctic Peninsula a maximum potential sea level rise contribution of 69 ± 5 mm.

Rott et al. (2014) estimated the total dynamic ice mass loss for the glaciers along the Nordenskjöld Coast and the Sjögren-Boydell glaciers after ice shelf disintegration to be 4.21 ± 0.37 Gt a$^{-1}$ between 2011–2013. Seehaus et al. (2015; 2016) revealed similar values for tributary glaciers of the former Larsen-A and Prince-Gustav-Channel ice shelves. On the western Antarctic Peninsula south of -70° increased ice discharge and considerable thinning rates have been reported for various ice shelf tributaries (Wouters et al., 2015).

The main cause for the current increased ice discharge on the Antarctic Peninsula is the dynamic response of tributary glaciers to the disintegration and basal thinning of several ice shelves (e.g. Angelis and Skvarca, 2003; Pritchard et al., 2012; Rignot, 2006; Wouters et al., 2015; Wuite et al., 2015). With the reduction or loss of the buttressing effect of the ice shelves (Fürst et al., 2016; Mercer, 1978) due to thinning or disintegration, the tributary glaciers accelerate and show imbalance (Rignot et al., 2005; Rott et al., 2014; Scambos et al., 2004).

For the south-western Antarctic Peninsula Rignot et al. (2013) demonstrated that basal melt of George VI, Wilkins, Bach and Stange Ice Shelves exceeded the ablation induced by calving. For Wordie Ice Shelf high basal melt rates of 23.6 ± 10 m a$^{-1}$ (2003–2008) and 14.79 ± 5.26 m a$^{-1}$ (2009) have been reported by Rignot et al. (2013) and Depoorter et al. (2013) respectively. However, the presented melt ratios (i.e. the ratio between basal melt and the sum of calving flux and basal melt) differ between 46 % (Rignot et al., 2013) and 82 % (Depoorter et al., 2013).

Wilkins Ice Shelf experienced amplified basal thinning controlled by small-scale coastal atmospheric and oceanic processes that assist ventilation of the sub-ice-shelf cavity by upper-ocean water masses (e.g. variations in wind stress or reduced freshwater fluxes from runoff and ice-shelf basal melt) until ~8 years before break-up events took place in 2008 and 2009 (Braun and Humbert, 2009; Padman et al., 2012). Subsequent changes in ice dynamics and stresses leading to break-up have been observed (Rankl et al., 2016). On George VI Ice Shelf, surface lowering is linked to enhanced basal melt caused by an increased circulation of warmed Circumpolar Deep Water (CDW) (Holt et al., 2013) and a 13 % increase in ice flow was observed between 1992 and 2015 for its tributary glaciers (Hogg et al., 2017).

However, how the dynamic response after ice shelf loss progresses and how long this process lasts, is frequently unknown. Seehaus et al. (2016) showed that significant temporal differences in the adaptation of glacier dynamics in response to ice shelf decay can occur and that those can only be resolved, if dense time series of satellite-based measurements are available. Wendt et al. (2010) also concluded for Wordie Ice Shelf that its former tributaries were still far from reaching a new equilibrium after retreat and collapse of the ice shelf starting in the 1960's. However, given the limited data used in previous

studies, Wendt et al. (2010) pointed out that a much closer monitoring is required to verify this. A recent comparison of stacked surface velocities of Fleming Glacier derived from InSAR in 2008 with velocities obtained from Landsat 8 feature tracking in 2014 and 2015 revealed that the glacier had sped up by ~400–500 m a[-1] (Walker and Gardner, 2017; Zhao et al., 2017). However, the question why the magnitude of change was much higher than recorded elsewhere at the western

Antarctic Peninsula over a similar time period remained unanswered so far. In this study we investigate the glacier dynamics of Fleming Glacier after the disintegration of Wordie Ice Shelf on the south-western Antarctic Peninsula. Our study ties in with previous works in the region, but covers a much longer time period at a much higher temporal resolution. We provide a dense time series of ice velocity measurements from Synthetic Aperture Radar (SAR) satellite data for the time period 1994–2016 for Fleming Glacier. In order to investigate the observed changes in ice dynamics, we conducted an in-depth analysis

of other geophysical and geodetic remote sensing data such as airborne Light Detection and Ranging (LiDAR) and satellite-borne laser altimetry, radio echo sounding for ice thickness, bistatic and monostatic SAR data as well as optical satellite images. We derive frontal retreat, surface velocity changes, ice elevation changes, grounding line positions and estimate the area of freely floating ice from hydrostatic equilibrium.

## 2 Study site

The former Wordie Ice Shelf was located in Marguerite Bay on the south-western Antarctic Peninsula. The ice shelf was originally fed by several major input units (Fig. 1). Among these, Fleming Glacier is the biggest. It has a current length of approx. 80 km and is up to 10 km wide at its tongue. With a speed of more than 8 m d[-1] close to its calving front (Fig. 1), Fleming Glacier is also the fastest flowing glacier in Wordie Bay. Fleming Glacier merges with Seller and Airy Glacier ~8 km upstream of their joint calving front. Together with Rotz Glacier, which merges with Seller Glacier ~28 km upstream of

the front, all four glaciers form the Airy-Rotz-Seller-Fleming glacier system, spanning a total catchment area of about 7000 km[2] (Cook et al., 2014).

Starting in the 1960s, Wordie Ice Shelf ran through a stepwise disintegration process (Fig. 1), which was controlled by pinning points (i.e. ice rises/rumples). Analyses of satellite imagery suggest that the ice shelf was temporarily grounded and stabilized at these pinning points until one of the next rapid break-up events took place (Doake and Vaughan, 1991;

Reynolds, 1988; Vaughan, 1993; Vaughan and Doake, 1996). However, during phases of ice front retreat, instead of protecting the ice shelf against decay, ice rises that were embedded in the ice shelf appeared to behave as indenting wedges, contributing to weakening the ice shelf and accelerating break-up (Doake and Vaughan, 1991; Vaughan, 1993). It is likely that the collapse of Wordie Ice Shelf was triggered by a combination of amplified ablation due to rising air temperatures (Doake and Vaughan, 1991), enhanced tidal action as a consequence of relaxed sea-ice conditions in Marguerite Bay

(Reynolds, 1988) and increased basal melt rates on ice shelves in the Bellingshausen Sea due to rising ocean temperatures (Depoorter et al., 2013; Holland et al., 2010; Pritchard et al., 2012; Rignot et al., 2013).

After the last big disintegration event in 1989, the ice shelf was split into a northern and a southern part (Doake and Vaughan, 1991). In the early 1990s most of the remaining floating ice in Wordie Bay consisted only of the protruding, unconfined tongues of the disconnected tributary glaciers. These tongues disappeared between 1998 and 1999, so that in 1999 Fleming Glacier was already calving near its 1996 grounding line (Rignot et al., 2005). Wendt et al. (2010) found the remaining area of floating ice to be only 96 km$^2$ in 2009. At this time there was virtually no contiguous ice shelf left and only the glaciers of the Prospect unit and Hariot`s unnamed neighbouring glacier still possessed floating glacier tongues (Wendt et al., 2010). During the following years (2010–2015) the fronts of the glaciers in Wordie Bay remained quite stable, except at the Prospect system where the once interconnected floating ice tongues of the three glaciers disconnected and some floating ice was lost. This resulted in an total area of 84 km$^2$ of ice shelf in Wordie Bay in 2015, if taking the grounding line of 1996 as a baseline (Rignot et al., 2005; Rignot et al., 2011a) and ignoring any grounding line migration.

While in the early 1990's an acceleration was not yet observed from the visual inspection of optical satellite imagery (Doake and Vaughan, 1991; Vaughan, 1993), Rignot et al. (2005) found substantial dynamic thinning and an increase of surface velocities by 40–50 % in 1996 against 3 point measurements by Doake (1975) in 1974 on Fleming Glacier (Fig. 1). The higher velocities as well as further thinning were also confirmed through Global Navigation Satellite System (GNSS) measurements in 2008 and airborne LiDAR surveys in 2004 and 2008 respectively (Wendt et al., 2010). Then, further speed up of ~400–500 m a$^{-1}$ was recorded between 2008 and 2014/2015 (Walker and Gardner, 2017; Zhao et al., 2017).

## 3 Data

We used a broad remote sensing data set in order to investigate the changes in ice dynamics at Fleming Glacier between 1994 and 2016 after the disintegration of Wordie Ice Shelf. Tab. 1 gives an overview of the specifications and the time coverages of the sensors used. The Bedmap2 digital elevation model (DEM) of Antarctica (Fretwell et al., 2013), resampled to 100 m resolution, was taken as a topographic reference for orthorectification of the surface velocity fields and for the derivation of local incidence angles required for the conversion from slant to ground range displacement. Over the Antarctic Peninsula the Bedmap2 DEM provides a seamless compilation of data from the improved ASTER GDEM (from ASTER stereo images acquired between 2000 and 2009) (Cook et al., 2012), the SPIRIT DEM (from SPOT stereo images acquired in 2007 and 2008) and the NSIDC DEM (from ICESat data acquired between 2003 and 2005) (DiMarzio et al., 2007). Calibrated and multi-looked SAR intensity images, Landsat 7 imagery and an existing dataset of ice shelf outlines (Ferrigno, 2008) were taken as a reference for the delineation of the ice shelf/glacier front (Tab. S1).

Calculations of elevation change rates were based on airborne LiDAR measurements, satellite-borne laser altimeter measurements and two DEMs derived from SAR interferometry (Tab. 1). The two DEMs covering the Airy-Rotz-Seller-Fleming glacier system were calculated from bistatic TerraSAR-X/TanDEM-X (TSX/TDX) Coregistered Single look Slant range Complex (CoSSC) strip map (SM) acquisitions on 2011-11-21 and monostatic TSX/TDX CoSSC SM acquisitions on 2014-11-03 (Fritz et al., 2012; Krieger et al., 2013). The TSX/TDX data were selected as close as possible to the dates of

two NASA Operation IceBridge (OIB) Airborne Topographic Mapper (ATM) flights in 2011 and 2014, in order to be able to correct for radar penetration depth biases.

A simulated phase from a subset of the TanDEM-X global DEM with a spatial resolution of 12 m (Rizzoli et al., 2017) was used to facilitate phase unwrapping during the generation of the two TSX/TDX DEMs. The TanDEM-X global DEM was also used as a reference for the absolute height adjustment of the TSX/TDX DEMs.

For the determination of the floating area on the tongue of Fleming Glacier, we used information on ice thickness and surface elevation from several Pre-IceBridge (PIB) and OIB flight lines across the Airy-Rotz-Seller-Fleming glacier system between 2002-11-26 and 2014-11-16 (Tab. 1). Depending on the date of acquisition, ice thickness data was recorded by different versions of the Coherent Radar Depth Sounder (CoRDS) (Tab. 1).

Our estimation of the recent grounding line of Fleming Glacier was based on a combination of the information on hydrostatic equilibrium with bedrock topography data, profiles of surface velocities and elevation change rate patterns inferred from the 2011–2014 TSX/TDX data. Information on bedrock topography was taken from the modelled bedrock grid of Huss and Farinotti (2014), which represents the most detailed dataset on bedrock topography available for the Antarctic Peninsula. The dataset was generated by subtracting modelled ice thickness from the improved ASTER GDEM by Cook et al. (2012). Ice thickness was derived by constraining a simple model based on the shallow ice approximation for ice dynamics with observational data of ice thickness (OIB) and surface velocity (Rignot et al., 2011b). Where available the modelled ice thickness was corrected with OIB ice thickness data, leading to more precise ice thickness values in such areas. On average, the local uncertainty in ice thickness of the dataset is 95 m, but uncertainties can reach 500 m for deep troughs without nearby OIB measurements. Since OIB coverage is fairly good across Fleming Glacier, uncertainties in modelled ice thickness are relatively low in this area. However, a comparison of bedrock elevations from Huss and Farinotti (2014) with bottom elevations calculated from OIB ATM and CoRDS measurements shows that although the modelled bedrock reflects the general subglacial topography well, the absolute difference in bottom elevation can be even more than 100 m (Fig. S5). One possible reason for this is a difference between ATM heights and the refined ASTER GDEM, which transfers to bedrock elevation.

# 4 Methods

## 4.1 Surface velocities

For each sensor consecutive pairs of coregistered single look complex SAR images were processed using an intensity offset tracking algorithm (Strozzi et al., 2002). A moving window was used to calculate surface displacements in azimuth and slant range direction between two SAR intensity images by localizing the peaks of an intensity-cross correlation function. The technique requires the definition of a tracking patch size and a step size (i.e. the distances in range and azimuth between the centres of two consecutive moving windows, Tab. 1). The parameters were chosen according to the sensor specifications, the temporal baseline between the acquisitions and the expected displacement.

During the tracking procedure the implementation of a cross correlation threshold of 0.05 assured the removal of low quality offset estimates. Post-processing of the velocity fields comprises an additional filtering (Burgess et al., 2012) based on the comparison of the orientation and magnitude of the displacement vectors relative to their surrounding vectors. This algorithm discards over 99 % of unreasonable tracking results. The filtered displacement fields were then transferred from slant range geometry into ground range geometry, geocoded and orthorectified. The procedure to determine the error of the velocity measurements is described in the Supplemental material, Sect. S2. The errors for each velocity field and the proportion of velocity vectors removed by the filter are listed in Tab. S3.

## 4.2 Elevation change

We derived ice thinning rates on Fleming Glacier for 2004–2008 and 2011–2014 by comparing ellipsoid heights of the PIB (ATM L1B, 2004), the Centro de Estudios Científicos Airborne Mapping System (CAMS, 2008) and the OIB (ATM L1B, 2011, 2014) airborne LiDAR datasets. The vertical accuracy of the ATM L1B elevation data is estimated to be better than 0.1 m (Krabill et al., 2002; Martin et al., 2012). For the CAMS data, vertical accuracy is 0.2 m (Wendt et al. (2010). Before subtraction, overlapping data of the originally closely spaced measurements were condensed to a common set of median surface elevations with an equal spacing of 50 m in along and across track direction. The locations of the resulting points of differential elevation measurements are shown in Fig. 4. The uncertainty of the gridded measurements can be attributed to both physical topographical features and measurement error. We approximated the uncertainty of each gridded median surface elevation similar as recommended in the IceBridge ATM L2 user guide (https://nsidc.org/data/ILATM2/versions/2/) by calculating the normalized median absolute deviation (NMAD) of all measurements in a 50 m grid cell. For each LiDAR dataset the median NMAD value of all grid cells was taken as the total uncertainty in surface elevation. Total uncertainties in surface elevation were 0.46 m (2004), 0.39 m (2008), 0.38 m (2011) and 0.43 m (2014). For each pair of overlapping gridded median surface elevations we derived the uncertainty in elevation change rate by calculating the root of the sum squares (RSS) of the corresponding NMAD values and dividing the result by the difference in acquisition time (years). The total uncertainty in elevation change rate for a complete dataset was calculated as the median of the elevation change rate uncertainties of all grid cells. The total uncertainties in elevation change rate were 0.32 m a$^{-1}$ for 2004–2008 and 0.24 m a$^{-1}$ for 2011–2014.

Additionally to the airborne LiDAR measurements ice elevation change rates for the period 2004–2008 were calculated from ellipsoid heights measured by the Ice, Cloud and Land Elevation Satellite (ICESat). Saturation of the 1064 nm Geoscience Laser Altimeter System (GLAS) detector can occur over ice, leading to a distorted echo waveform (Schutz et al., 2005). Hence we applied a saturation elevation correction provided on the GLA12 product prior to subtracting both tracks and excluded elevation measurements with flagged invalid saturation correction values from our analyses. Since both repeat tracks are not overlapping but separated by ~150 m in across track direction, we linearly interpolated the elevation data of the 2004 track onto the latitude values of the 2008 data prior to subtraction, as described in Fricker and Padman (2006). In order to keep the error induced by interpolation low, elevation values were only allowed to be interpolated between two

footprint centre locations with an along track spacing of ~170 m. This assured that existing gaps in the real data were preserved. Shuman et al. (2006) report a relative accuracy of 0.25 m for ICESat elevations measured on surface slopes between 1.5 and 2.0°. The mean surface slopes along the two ICESat elevation profiles were 1.9°. Hence, taking into account further possible inaccuracies of 0.15 m due to interpolation, we estimated the accuracy of the ICESat ice elevations to be ±

0.4 m. By calculating the RSS of the ICESat elevation uncertainties and dividing the result by the time interval between the measurements (years), we estimated the uncertainty in ICESat elevation change rates to be 0.13 m a$^{-1}$.

A map of elevation change rates between 2011 and 2014 was calculated by differencing two TSX/TDX-DEMs. Both DEMs have a spatial resolution of 10 m. To generate the DEMs we applied a differential interferometric approach, which facilitates phase unwrapping by incorporating the topographic information of a reference DEM (Vijay and Braun, 2016). A subset of

the TanDEM-X global DEM, covering the two TSX/TDX-DEMs, was chosen to be the reference DEM.

Before differencing, the TSX/TDX-DEMs must be vertically referenced. For this purpose the vertical offset between the DEMs and the TanDEM-X global DEM was measured over stable areas (i.e. tops of nunataks and rock outcrops, which were not affected by image distortions) at altitudes between 150 m and 1000 m (Fig. S1). Both DEMs were adjusted according to the median values of all ground control measurements (n=35452), which were -5.1 m for the 2011 DEM and 0.48 m for the

2014 DEM.. After subtracting the vertically registered DEMs, the elevation differences were converted into yearly elevation change rates. We assessed the accuracy of the vertical registration over another set of stable areas at altitudes between 150 m and 1300 m (Fig. S1). The absolute median value of the extracted change rates was 0.37 m a$^{-1}$ which primarily accounts for errors related to the vertical registration.

However, since radar signals can penetrate several meters into snow and ice, depending on the radar frequency and the

dielectricity of the medium (Mätzler, 1987; Rignot et al., 2001), an additional bias is induced on glaciated areas when differencing interferometric DEMs from different times and/or frequencies (Berthier et al., 2016; Seehaus et al., 2015; Vijay and Braun, 2016). Since the TSX/TDX data was acquired only 4–7 days apart from the ATM data, differences in elevation change rates between the two datasets can be primarily attributed to differences in penetration depth at the TSX/TDX acquisitions in 2011 and 2014 and remaining vertical registration errors. In order to compare the TSX/TDX data with the

ATM data, we extracted the TSX/TDX elevation change rates at the locations of the differential OIB ATM measurements using a buffer with a radius of 25 m and calculated the median for each point. Hypsometric reference values were taken from the resampled Bedmap2 DEM which we converted to ellipsoidal heights using the included geoid correction layer. The comparison between elevation change rates obtained from the 2011–2014 OIB ATM flights and the 2011–2014 TSX/TDX data after the vertical registration of the DEMs showed a maximum overestimation of ice thinning of 1.25 m a$^{-1}$ for the

TSX/TDX measurements (Fig. S4 a, b). However, the general trend of the elevation change rates fits well to those calculated from the LiDAR data and significant differences in elevation change were only measured in the lower areas of the glacier tongue. In the upper areas (above 600 m altitude) the difference between ATM and TSX/TDX elevation change rates was close to 0 m. Here the snow volume was likely completely frozen on both dates of acquisition, so that the penetration bias cancelled out. A backscatter comparison showed lower values in 2014 than in 2011 in areas below 600 m altitude, whereas

the backscatter in the upper areas above 600 m altitude was similar for both dates (Fig. S4 d). We corrected the TSX/TDX data with a local polynomial model based on the elevation change rate differences between the ATM and the TSX/TDX data (Fig. S4 b). We applied this correction to all glaciated areas below 1000 m and clipped the TSX/TDX elevation change rate map accordingly. The RMSE between the cubic fits of the ATM elevation change rates and the extracted values from the

corrected TSX/TDX map was 0.02 m a$^{-1}$ (Fig. S4 c). Nevertheless, penetration depths of the radar signal may be spatially variable at the same altitude e.g. due to local differences in surface melt or ice properties. Hence we checked the validity of the applied penetration depth correction by comparing the corrected TSX/TDX elevation change rates with an independent validation track of 2011–2014 ATM dh/dt rates (see Fig. 4 for location). Figure S4 e shows that the elevation change rates of both datasets are in good agreement. The RMSE between the cubic fits of the data was 0.24 m a$^{-1}$. Hence we estimate the

uncertainty due to remaining penetration depth differences to be 0.3 m a$^{-1}$. Together with the error of vertical registration, this resulted in a total error of 0.67 m a$^{-1}$ for the TSX/TDX ice thinning rates.

For our analyses of elevation change we compared ice thinning rates from the PIB and the CAMS airborne laser altimeter data (2004–2008) with rates obtained from the OIB data (2011–2014) as well as elevation change rates from the TSX/TDX data (2011–2014) with those derived from ICESat in 2004 and 2008. For the comparison between the TSX/TDX and the

ICESat data, ice thinning rates were extracted from the TSX/TDX map at the GLAS centre locations of the 2008-10-04 track. To take into account the 70 m footprint of the GLAS instrument, we applied a buffer with a radius of 35 m and calculated the median from the extracted values at each point.

### 4.3 Floating area (hydrostatic height anomalies) and estimation of recent grounding line

In order to determine the floating area on the tongue of the Airy-Rotz-Seller-Fleming glacier system at different points in

time, we derived hydrostatic height anomalies $\Delta e$ from the PIB and OIB elevation and ice thickness measurements between 2002 and 2014. For every measuring point of ice thickness, $\Delta e$ was calculated similar to Fricker (2002) by subtracting a theoretical freeboard height in hydrostatic equilibrium $e_{he}$ from a measured orthometric ATM ice surface elevation $e$:

$$\Delta e = e - e_{he} \qquad\qquad (1)$$

Regions on the glacier tongue where $\Delta e \leq 0$ were considered as freely floating. Before deriving hydrostatic height

anomalies, we merged simultaneously acquired ice thickness and ATM data by calculating the median elevation within a buffer of 50 m at each ice thickness measurement. As the ATM heights were originally measured relative to the WGS84 ellipsoid, we converted the ellipsoidal ATM values to orthometric heights prior to the buoyancy calculations. For the conversion we used kriged geoid values calculated for a mean tide system with the EIGEN-6C4 global gravity field model (Förste et al., 2014). We calculated $e_{he}$ by applying a modified formula after Griggs and Bamber (2011):

$$e_{he} = (H_i + \delta) - \frac{H_i \cdot \rho_i}{\rho_w} \qquad\qquad (2)$$

where $H_i$ is the measured PIB or OIB ice thickness, i.e. the ice thickness derived under the assumption that all ice is homogeneous and firn free, $\rho_i$ is the ice density of pure ice, $\rho_w$ is the density of sea water and δ is the firn density correction factor, i.e. the difference between the actual thickness of the firn layer above the glacier ice and the thickness that the firn would have if it were at the density of pure ice (Griggs and Bamber, 2011; van den Broeke, Michiel et al., 2008). Details on the uncertainties of all variables used for the calculations as well as the assessment of error propagation are provided in the Supplemental material, Sect. S3.

The buoyancy calculations provided information on the limit of hydrostatic equilibrium between 2002 and 2014 at several locations on the glacier system. As demonstrated in Seehaus et al. (2015), clear patterns of low or negative ice thinning rates in dh/dt maps reveal areas of floating ice, since buoyancy can cause originally grounded ice to bounce and/or decreases the effect of ice thinning on ice surface elevation to ~10 %.

Moreover, the grounding line marks the transition between two fundamentally different flow regimes of grounded and freely floating ice. Whereas the flow dynamics of grounded ice are dominated by vertical shear and controlled by basal drag, flow of floating ice is drag free and dominated by longitudinal stretching and lateral shear (Schoof, 2007). The difference in flow dynamics of grounded and floating ice can result in pronounced changes in surface velocity close to the grounding line (Stearns, 2007; Stearns, 2011). Furthermore, Rignot et al. (2002) demonstrated that if ungrounding occurs, the resulting flow acceleration usually affects both the floating and the grounded part of the glacier, but is largest near the grounding zone. Thus, velocity profiles can serve as additional information for locating the grounding line (Stearns, 2007; Stearns, 2011).

Bedrock elevation data can reveal subglacial topographic features which act as pinning points for the glacier. Hence, our estimations of recent and previous grounding line positions were based on information on hydrostatic equilibrium from the hydrostatic height anomaly calculations as well as maps and profiles of TDX/TSX 2011–2014 elevation change rates, modelled bedrock topography (Huss and Farinotti, 2014) and surface velocities. Wherever possible, we gave preference to information on hydrostatic equilibrium for the final decision of the recent grounding line location. For selected profiles across the glacier, recent and previous (2008) grounding line positions were estimated by combining evidence from elevation change, bedrock topography and surface velocity. In the remaining areas, the recent grounding line was interpolated by combining information on elevation change rate patterns in the TDX/TSX 2011–2014 dh/dt map with information on bedrock topography.

## 5 Results

### 5.1 Surface velocities

Figure 2 shows the multi sensor time series (1994–2016) of SAR intensity tracking derived velocities along a centreline profile on Fleming Glacier (Fig. 1). The profile extends from the grounding line location in 1996 to 16 km upstream. The relative distance of the glacier front to the 1996 grounding line is shown on the left side of the plot. Distances in the subsequent text are given in reference to the grounding line of 1996. Positive values relate to positions on the glacier

upstream of (behind) the former grounding line, while negative values refer to locations seawards of the 1996 grounding line. Figures 3a, b depict absolute and relative velocity changes for 1 km bins along the centreline profile for the periods 1997–January 2008, January 2008–April 2008, 2010–2011, April 2008–2011 and 2011–2015. In 2011 the location of the glacier front reached its most inland position. We excluded all measurements seaward of the 2011 glacier front from our analyses, since this is the section of the profile where frontal change took place and we did not want to compare velocities measured on sea ice or ice mélange.

Figure 2 shows that glacier velocities were rather stable between 1994 and January 2008 although almost all of the remaining floating tongue got lost between 1998 and 1999. After 1999 the glacier front remained comparatively steady close to the grounding line location in 1996 for almost 10 years. The NMAD of the median velocities between 1994 and January 2008 was 0.06 m d$^{-1}$(Fig. 2). Figure 3a reveals that the median velocity difference between 1997 and January 2008 along the centreline profile was just 0.07 m d$^{-1}$ with maximum velocity differences not exceeding 0.2 m d$^{-1}$. Between January and April 2008 a rapid and almost constant acceleration of ~0.4 m d$^{-1}$ was noticeable along the centreline until ~13 km upstream (Fig. 3a) and velocities >5 m d$^{-1}$ propagated ~8 km inland (Fig. 2, orange and red colours). The median relative increase in surface velocity between 3 and 13 km upstream was ~8 %, with a maximum increase of ~10 % at ~13 km upstream (Fig. 3b). Simultaneously, the front of Fleming Glacier retreated behind the 1996 grounding line for the first time. Since 2008 the glacier tongue has not advanced seaward of the 1996 grounding line position (Fig. 2).

The velocity pattern persisted until March 2010, when a second phase of acceleration began. Our velocity time series shows that velocities >5 m d$^{-1}$ gradually propagated further inland within one year, until they reached ~12 km upstream in early 2011. Velocity change during this time period increased towards the glacier front and reached a maximum value of ~0.9 m d$^{-1}$ or ~16 % at ~6 km upstream. Large changes in surface velocity close to the 2011 front in the periods January 2008–2011, 2010–2011 and 2011–2016 were ignored, since they do not represent real dynamic change, but result from comparing the inherently higher frontal velocities in 2011 with lower velocities of the floating glacier tongue in 2008, 2010 and 2016. If looking at the complete period between January 2008 and 2011 the increase in median surface velocity between ~4 and ~7 km upstream was ~1.3 m d$^{-1}$ or ~27 % (Fig. 3a, b). If ignoring velocity change in the vicinity of the 2011 glacier front, the highest acceleration values of >1.4 m d$^{-1}$ or ~28 % were recorded at ~ 6 km upstream of the 1996 grounding line (Fig. 3a). The amount of absolute and relative acceleration abruptly drops at ~12 km (Fig. 3 a, b). If excluding measurements at the 2011 front, no further marked changes in velocities were detected along our centreline profile after 2011. From 2011 to 2016 median velocity change was just 0.06 m d$^{-1}$ between 4 and 16 km upstream and maximum values were smaller than 0.2 m d$^{-1}$ (Fig. 3 a). For the same time period the NMAD of the median centreline velocities was just 0.02 m d$^{-1}$ (Fig. 2).

Surface velocities on 2013-12-24 at the three measuring sites of Doake (1975) ~50 km upstream (Fig. 1) were very similar to those measured in 1996 and 2008. Furthermore, the flow directions in 2013 were like those in 2008 and 1974. However, surface velocities derived from Landsat 8 feature tracking suggest that in 2015 velocities at the three measuring sites had increased by ~20% in comparison to 2008 (Walker and Gardner, 2017; Zhao et al., 2017) (Tab. 2).

## 5.2 Elevation change

Figure 4 shows elevation change rates on the Airy-Rotz-Seller-Fleming glacier system for the period between 2011 and 2014. The entire area undergoes a considerable drawdown. On Fleming Glacier the highest ice thinning rates with peak values of more than ~6 m a$^{-1}$ were recorded in a zone extending from ~8 to ~14 km upstream. On Seller Glacier the ice loss

exceeds ~6 m a$^{-1}$ at about 7 km upstream. In general, ice thinning decreases towards higher altitudes. A tendency to lower negative or even positive elevation change rates was observed on the lower parts of the joint Fleming and Seller glacier tongue between 0 and up to ~9 km upstream. The pattern was not as clear as on Airy Glacier, where a distinct area of low ice thinning rates was detected between 0 and ~4 km upstream.

Figures 5 a) and b) show comparisons of elevation change rates for the times prior to (2004–2008) and after the glacier

acceleration (2011–2014).The location of the data is shown in Fig. 4. In Fig. 5 a) elevation change rates from PIB ATM-CAMS measurements (2004–2008) are plotted together with rates from ATM measurements in 2011 and 2014. The large scattering of the data is due to the highly crevassed surface of the glacier tongue, where a purely horizontal displacement of crevasses can cause apparent positive and negative elevation differences. Therefore, a median filter was applied to the data before adjustment of a cubic function. Fig. 5 b) shows ice thinning rates from ICESat tracks in 2004 and 2008 together with

rates calculated from 2011–2014 TSX/TDX data. Note that the ATM data in Fig. 5 a) and the ICESat data in Fig. 5 b) refer to different profiles.

Figure 5 shows that prior to the speedup in 2008, Fleming Glacier has already been affected by pronounced surface lowering. A clear trend of increasing ice thinning rates towards the glacier front is visible for 2004–2008 on both profiles. During this period the maximum negative elevation change rates were found close to the 1996 grounding line. Here the cubic

regression functions imply that the ice surface lowered at a maximum of ~3.8 m a$^{-1}$ for the CAMS-ATM measurements and at ~4.6 m a$^{-1}$ for the ICESat data. For all median elevation change rates presented below, we calculated the NMAD in order to account for the statistical dispersion of the input data. The median ice thinning rates measured during 2004–2008 for the cubic fits were 1.5 ± 0.6 m a$^{-1}$ on the CAMS-ATM flightpath and 1.9 ± 1 m a$^{-1}$ on the ICESat track. The OIB ATM and the TSX/TDX elevation change rates between 2011 and 2014 reveal a significant change in pattern for the time after the glacier

flow acceleration. A tendency to lower ice thinning rates is present towards the glacier front and high negative elevation change rates can be found in a zone 10–15 km upstream, with maximum ice losses of ~3.7 m a$^{-1}$ for the ATM, and ~4.1 m a$^{-1}$ for the TSX/TDX cubic regression functions. The median elevation change rates were -3.2 ± 0.8 m a$^{-1}$ and -2.6 ± 1.2 m a$^{-1}$ for the cubic fits of the 2011–2014 TSX/TDX data (Fig. 5 b) and the 2011–2014 ATM data (Fig. 5 a), respectively. Despite of lower ice thinning rates measured towards the ice front in 2011–2014, our data show an overall median increase of ice

thinning rates along the profiles of ~1.1–1.3 m a$^{-1}$ or ~70 % between the periods from 2004 to 2008 and from 2011 to 2014. However, in some areas 10–15 km upstream, ice thinning rates even doubled in the latter period.

## 5.3 Floating area (hydrostatic height anomalies) and estimation of recent grounding line

Figure 6 depicts the results of the hydrostatic height anomaly calculations from PIB and OIB elevation and ice thickness data acquired before (2002–2004) and after the speedup of Fleming Glacier (2011–2014). Detailed plots showing the results of the hydrostatic height anomaly calculations along PIB and OIB flight lines can be found in the Supplemental Material, Fig. S5 a–e.

The hydrostatic height anomaly data of 2002 and 2004 (Fig. 6, Track 1 and 2) clearly reveal that the ice inland of the 1996 grounding line was not floating at these times. However, the same calculations for data acquired in 2011 and 2014 (Fig. 6, Track 3–5) as well as patterns of low and positive elevation change rates in the TSX/TDX 2011–2014 dh/dt map (Fig. 4, S7) suggest that after the final stage of glacier acceleration in 2011 an area of about 56 km$^2$ (referring to the front in 2014) of the formerly grounded glacier tongue of the Airy Rotz Seller Fleming system had been afloat.

The bedrock elevation model of Huss and Farinotti (2014) exhibits, that the boundary of the area showing flotation follows bedrock ridges (Fig. 6). Those confine a subglacial trough underneath the Airy-Rotz-Seller-Fleming glacier system. The ridges reach up to ~9 km upstream of the 1996 grounding line. For most regions of the glacier tongue we estimate the current grounding line to coincide with these ridges at an elevation between ~-400 and -500 m. On Fleming and Seller Glacier our estimation of the recent grounding line also largely coincides with the extent of lower ice elevation change rates apparent in the TSX/TDX 2011–2014 dh/dt map (Fig. S7). However, on Airy Glacier a distinct area of low ice thinning rates on the lower part of the glacier tongue indicates floatation (Fig. 4, S7), whereas hydrostatic equilibrium in 2011 suggests that the glacier is grounded on a hump which reaches to the subglacial trough (Fig. 6).

We extracted data of surface velocities, TDX/TSX 2011–2014 elevation change rates and bedrock topography along four profiles on Airy and Fleming Glacier in order to estimate recent grounding line positions and those in 2008 after the first acceleration phase (Fig. S6 a–d). The locations of the profiles as well as the deduced grounding line locations are shown in Fig. 6. After the first acceleration phase in 2008 the front of Fleming Glacier had retreated behind the 1996 grounding line for the first time. Hence, on Fleming Glacier the grounding line must have been situated upstream of the 1996 position at this time. The profile plots in Fig. S6 b, c suggest that after the first acceleration phase in 2008 the grounding line was not located as far upstream as after the second acceleration phase between 2010 and 2011. However, since the estimation of the 2008 grounding line positions was based on surface velocities and modelled bedrock topography only, their precise locations remain unclear. Hence, the 2008 grounding line positions indicated in Fig. 6 are just a best guess based on the data we have in hand. A more detailed discussion on how the grounding line positions were finally decided from the profiles is provided in the Supplemental Material, Fig. S6 a–d. All in all, we estimated the current (2014) grounding line of Fleming Glacier to be located ~6–9 km upstream of its 1996 position. Its likely recent location is consistent with the maximum extent of upstream propagation of high velocities in 2008 on the centreline profile (Fig. 6).

**6 Discussion**

Our results confirm the previously detected acceleration of Fleming Glacier in response to the stepwise break-up and disintegration of Wordie Ice Shelf (Rignot et al. 2005). Median elevation change rates of -1.5 m a$^{-1}$ and -1.9 m a$^{-1}$ between 2004 and 2008 may suggest that Fleming Glacier had not reached a new equilibrium even almost 20 years after the partial disintegration of the ice shelf in 1989. Nevertheless, our dense velocity time series shows that surface velocities remained fairly stable between 1994 and 2007.

Between January and April 2008 the glacier had abruptly accelerated and high velocities had propagated upstream. Between March 2010 and early 2011 a second phase of acceleration was detected during which the speedup gradually propagated further upstream. This two-step acceleration of different characteristic has not been detected before and is important for the explanation of the strong recent dynamic changes on Fleming Glacier. Notwithstanding, the median speedup of ~1.3 m d$^{-1}$ which we recorded between 2008 and 2011 is in good agreement with an acceleration of ~ 400–500 m a$^{-1}$ reported by Walker and Gardner (2017) and Zhao et al. (2017) for the period 2008–2014/2015. However, a comparison of our velocities in 2013 with their velocities in 2015 at the three measuring sites of Doake (1975) ~50 km upstream suggests that the recent speedup had not propagated up to these locations prior to 2015.

Abrupt speedups of tributary glaciers are often recorded as a direct consequence of loss of the buttressing force or major calving events (e.g. Seehaus et al., 2015). However, we did not observe any major calving event, which could have been responsible for the observed acceleration in 2008 or afterwards. In Greenland seasonal velocity fluctuations have been linked to both enhanced basal sliding due to the penetration of surface melt water to the ice-bedrock interface and inter-annual differences in drainage efficiency (Moon et al., 2014; Sundal et al., 2011; Zwally et al., 2002). However, the meltwater production on Fleming Glacier is considered to be generally not sufficient to percolate to the glacier bed (Rignot et al., 2005). Furthermore, a trend of cooling air temperatures is reported for the Antarctic Peninsula since the end of the 1990s (e.g. Turner et al., 2016). Although decadal mean surface temperatures in the period 2006–2015 were 0.2 °C higher than in 1996–2005 at San Martin station (~120 km north of Wordie Ice Shelf), warming rates have decreased markedly since the decade 1996–2005 and show a cooling trend in 2006–2015 (Oliva et al., 2017). This may have further reduced surface melt during recent years. All in all, enhanced basal sliding due to percolating meltwater is likely not the explanation for the observed increase in flow velocities. However, we do not rule out that as a consequence of the acceleration, basal sliding increased in grounded areas by meltwater generated from greater basal frictional heat. Hydrostatic height anomalies calculated from OIB ice thickness and surface elevation data, TSX/TDX elevation change rates, surface velocities and modelled bedrock topography suggest that the current grounding line of Fleming Glacier is located ~6–9 km upstream of its 1996 position, following the edges of a subglacial trough. By calculating hydrostatic equilibrium for a small subsection of OIB tracks at the outer glacier front close to the margins of Mount Balfour (Figure 6, Track 5 and Figure 4 grey dots), Walker and Gardner (2017) found that the grounding line had retreated by ~500 m between 2002 and 2014. However, they did not calculate hydrostatic equilibrium for the glacier`s centre and referred to the grounding line in 1996 instead. We now

provide an estimate of the current grounding line on the central part of the glacier for the first time and show that it had substantially retreated. This is additional key information for the explanation of the recent glaciological changes on Fleming Glacier, since ungrounding causes parts of the glacier tongue to go afloat, which reduces buttressing and basal friction. This in turn provokes the glacier to speed up and to dynamically thin. We hence propose that unpinning and grounding line retreat are the main causes of the observed strong acceleration of Fleming Glacier. This is an advance in the understanding of the recent processes at the glacier and updates the interpretation of Walker and Gardner (2017), who attributed the observed changes to increased frontal ablation and could not rule out that the acceleration is a delayed response to ice shelf disintegration in 1989.

Fairly stable velocities between 1994 and January 2008 as well as hydrostatic height anomalies in 2002 and 2004 do not indicate that ungrounding from the 1996 grounding line position had happened prior to January 2008. Although we were not able to give a precise estimate of the grounding line in 2008, the fact that during the acceleration phase in 2008 the glacier front had retreated behind the 1996 grounding line for the first time, shows that the 2008 grounding line must have been located upstream of the 1996 position. However, our data suggest that in 2008 the grounding line had not retreated to the edge of the subglacial trough at ~6–9 km upstream, yet. The rapidity of the acceleration in 2008 indicates that resistance to glacier flow must have abruptly been reduced. This is characteristic of a response to sudden unpinning rather than to gradual grounding line retreat. We hence propose that in 2008 the frontal part of the glacier abruptly detached from a pinning point (likely a sill) located at the 1996 grounding line. A cavity underneath the ice has probably already existed. Between 2008 and early 2010 Fleming Glacier was possibly grounded and stabilized on a gentle hill ~2.5–4 km upstream of the 1996 grounding line. The second phase of gradual acceleration and upstream propagation of high velocities between March 2010 and early 2011 is likely a response to further gradual grounding line retreat to the recent position ~6-9 km upstream. On our centerline profile highest changes in ice velocity were recorded ~6 km upstream, which is where the glacier now starts to be grounded. Furthermore, 70 % higher median ice thinning rates in the period between 2011 and 2014 in comparison to the period 2004–2008 point to increased dynamic thinning and mass loss after grounding line retreat. The highest negative elevation change rates migrated upstream and can now be found in the vicinity of the estimated current grounding line. A tendency of lower ice thinning rates towards the glacier front, which was detected along the 2011–2014 OIB ATM profiles as well as in the 2011–2014 TSX/TDX dh/dt map, indicates floatation of the glacier tongue. This has neither been observed by Zhao et al. (2017), who did not specifically analyse elevation change for this time period, nor by Walker and Gardner (2017), who calculated elevation change from the same OIB ATM dataset. The latter binned elevation change in 5 km intervals and excluded all data up to 5.5 km landward of the 1996 grounding line, which is the part of the profile where the trend towards lower ice thinning is most prominent.

Rignot et al. (2013) and Depoorter et al. (2013) reported high basal melt rates of $23.6 \pm 10$ m a$^{-1}$ (2003–2008) and $14.79 \pm 5.26$ m a$^{-1}$ (2009) for the remaining parts of Wordie Ice Shelf, respectively. The magnitude of basal thinning is comparable to those found for ice shelves in the Amundsen Sea sector, where the influx of relatively warm CDW onto the continental shelf is thought to be the dominant driver for recent substantial grounding line retreat, acceleration and dynamic thinning of

several glaciers (Turner et al., 2017). Periodical pulses of warm CDW are also known to flood onto the continental shelf of Marguerite Bay (Holland et al., 2010). Significant warming of Antarctic Continental Shelf Bottom Water (ASBW) of 0.1° to 0.3°C decade[-1] since the 1990s were recorded in the Bellingshausen Sea region and linked to increased warming and shoaling of CDW (Schmidtko et al., 2014). Cook et al. (2016) proposed that oceanic melt induced by an increased shoaling

of relatively warm CDW is responsible for an accelerated frontal retreat of tidewater glaciers in the south-western Antarctic Peninsula since the 1990s. Other studies reported considerable thinning of the nearby George VI Ice Shelf (Hogg et al., 2017; Holt et al., 2013) and other ice shelves on the south-western Antarctic Peninsula (e.g. Rignot et al., 2013) due to increased basal melt. The onset of Fleming Glacier`s speedup between January and April 2008 corresponds well with observations of Wouters et al. (2015). They reported first signs of a near simultaneous increase of ice mass loss for glaciers

all across the western Antarctic Peninsula south of -70° around 2008 and an unabated rapid ice loss since 2009. For the glaciers on Western Palmer Land Hogg et al. (2017) showed that part of the ice loss after 2009 can be attributed to dynamic thinning triggered by ocean driven melt.

Walker and Gardner (2017) found that in 2008/2009 and 2010/2011 exceptional warm water intrusions into Wordie Bay occurred due to upwelling CDW in response to phases of anomalously strong north-westerly winds during strong La Niña

and positive SAM (Southern Annular Mode) events. Highest temperatures were not only recorded at depths between 100–200 m but also at 400 m, which is close to where Fleming Glacier is grounded. The coincident timing with the two phases of glacier acceleration substantiates the link between ocean warming and our observed dynamic changes. It is very likely that submarine ice melting was increased during phases of strong CDW upwelling and that this has triggered unpinning from the 1996 grounding line position in 2008 as well as further gradual grounding line retreat in 2010–2011.

The strong basal melt rates proposed by Rignot et al. (2013) for 2003–2008 further suggest that basal melt has already occurred prior to 2008. This, together with increased dynamic thinning towards the ice front between 2004 and 2008 has likely weakened the ice at the pinning point, which may have fostered unpinning in 2008. Furthermore, the bed topography reveals that the trough underneath the joint Airy-Rotz-Seller-Fleming glacier tongue has a retrograde slope on its central part (Fig. S6 b, 0–6 km). Such a bed topography is known to be an unstable configuration for the glacier (e.g. DeConto and

Pollard, 2016; Favier et al., 2014; Rignot et al., 2014; Schoof, 2012), which may have promoted gradual grounding line retreat between 2010 and 2011.

The bedrock topography of Fleming Glacier also shows a retrograde bed slope starting at ~3–4 km upstream of the current grounding line, which transitions into a pronounced deep trough (up to 1100 m below sea level) at about 10 km upstream. Hence, if grounding line retreat exceeds the edge of this trough, destabilisation like on Thwaites Glacier and in the Pine

Island Bay region is possible, which would involve further rapid grounding line retreat and amplified mass loss in the future (Favier et al., 2014; Rignot et al., 2014).

## 7 Conclusions

We present a detailed history of the glacier dynamics of Fleming Glacier after the retreat and disintegration of Wordie Ice Shelf. While previous studies analysed only rather limited amounts of velocity data (Rignot et al., 2005; Walker and Gardner, 2017; Wendt et al., 2010; Zhao et al., 2017), we now show a complete time series of SAR surface velocities of

Fleming Glacier at high temporal resolution for the period 1994–2016. These data enable a much better temporal constraint and characterisation of glaciological changes in the region. By combining this unique dataset with recently published data on oceanic forcing (Walker and Gardner, 2017) and new data on surface elevation change from TSX/TDX interferometry, ICESat laser altimetry and airborne LiDAR, as well as modelled bedrock topography and hydrostatic equilibrium, we are able to relate precisely dated events of acceleration to increased dynamic ice thinning and ocean driven grounding line

retreat. This complements previous studies in the region and provides new and more detailed information on the glaciological changes and their drivers.

Our results show that until 2008 the dynamics of Fleming Glacier were primarily controlled by the impacts of break-up events of Wordie Ice Shelf before the early 1990s. The retreat of the ice shelf reduced glacier buttressing and led to an increase in surface velocities (Rignot et al., 2005), which in turn caused the glacier to dynamically thin. The last floating ice

shelf parts were lost between 1998 and 1999, but this showed no detectable effects on glacier flow dynamics.

After two decades of rather stable velocities, the glacier abruptly accelerated between January and April 2008. Our interpretation is that this happened due to the detachment of the glacier tongue from a pinning point located at the 1996 grounding line position. The unpinning was likely fostered by weakening of the ice due to basal melt and dynamic thinning prior to 2008. Further gradual retreat of the grounding line between 2010 and 2011, an increase in surface velocities of ~27

% as well as ~70 % higher ice thinning rates show that ungrounding in 2008 has initiated a new phase of dynamic imbalance. The unfavourable retrograde bed slope underneath Fleming Glacier probably amplified the grounding line retreat. The coincident timing of reported strong upwelling events of warm CDW with the two phases of acceleration and grounding line retreat shows that enhanced basal melt due to increased shoaling of warm CDW most likely played a major role for the recent changes at Fleming Glacier. The reduction in buttressing due to unpinning and grounding line retreat is able to explain

why the magnitude of velocity change was much higher than in other places at the western Antarctic Peninsula during this time.

Today Fleming Glacier and the other glaciers of the Airy-Rotz-Seller-Fleming glacier system are far away from reaching a new equilibrium. The modelled subglacial topography of Fleming Glacier upstream of the recent grounding line is characterized by some smaller troughs which are separated by chains of gentle hills. Pronounced oceanic forcing will

presumably continue, since the SAM is forecasted to be shifted further poleward, which will foster conditions like those during the strong La Niña/+SAM events in 2008/2009 and 2010/2011 (Abram et al., 2014; Fogt et al., 2011; Walker and Gardner, 2017). Thus, further retreat of the grounding line and more dynamic thinning are possible on Fleming Glacier. If ungrounding would reach upstream to the retrograde bed slope at about ~3–4 km from the current grounding line and further

to the deep subglacial trough, this can trigger a positive feedback loop of rapid grounding line retreat, flow acceleration, dynamic thinning, increased calving and mass loss. However, on Airy and Seller glaciers the more favourable subglacial geometry of an overall landward steepening slope may slow down or prevent further grounding line retreat in the future

## Author contributions

5 Peter Friedl processed and analysed all SAR, optical, elevation and ground penetrating radar data. Thorsten C. Seehaus processed the ALOS PALSAR data and supported the interferometric DEM generation. Anja Wendt provided the CAMS LiDAR data and gave valuable comments. Matthias H. Braun coordinated the research and wrote the manuscript jointly with Peter Friedl. Kathrin Höppner initiated the project, co-coordinated the research in close cooperation with Matthias H. Braun and contributed to discussions of the results throughout. All authors revised the manuscript.

## Competing interests

The authors declare no competing financial interests.

## Acknowledgements

This project was mainly funded by the DLR VO-R Young Investigator Group "Antarctic Research". Thorsten C. Seehaus was funded by Deutsche Forschungsgemeinschaft (DFG) in the framework of the priority program "Antarctic Research with 15 comparative investigations in Arctic ice areas" by a grant to M.B. (BR 2105/9-1). Matthias H. Braun and Thorsten C. Seehaus would like to thank the HGF Alliance "Remote Sensing of Earth System Dynamics" (HA-310) and Marie-Curie-Network International Research Staff Exchange Scheme IMCONet (EU FP7-PEOPLE-2012-IRSES, Project Reference: 318718) for additional support. TerraSAR-X and TanDEM-X data were provided by DLR through the projects HYD3008 and XTI_GLAC7015. We would like to thank Rainer Lorenz (German Remote Sensing Data Centre, DLR) for his support 20 retrieving the ERS-1/-2 SLC data from the ESA ERS-1/-2 Processing and Archiving Facility (D-PAF) at DLR and the ENVISAT ASAR SLC data from the ESA ENVISAT Processing and Archiving Centre (D-PAC) at DLR. All ERS, ENVISAT and TSX/TDX data were received at the German Antarctic Receiving Station GARS O'Higgins of DLR. The CAMS data were acquired and kindly made available by the Centro de Estudios Científicos, Chile. Furthermore, we are grateful to Christian Kienholz (University of Alaska Fairbanks) and Bill Hauer (Alaska Satellite Facility) for providing the 25 Radarsat-1 SLC data. Other remote sensing data was kindly made available by ESA, NASA, JAXA, USGS and CReSIS. Finally, we would like to thank Paul Wachter, Saurabh Vijay, the tree anonymous reviewers and the editor, Bert Wouters, for their helpful comments on the manuscript.

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

**Figures**

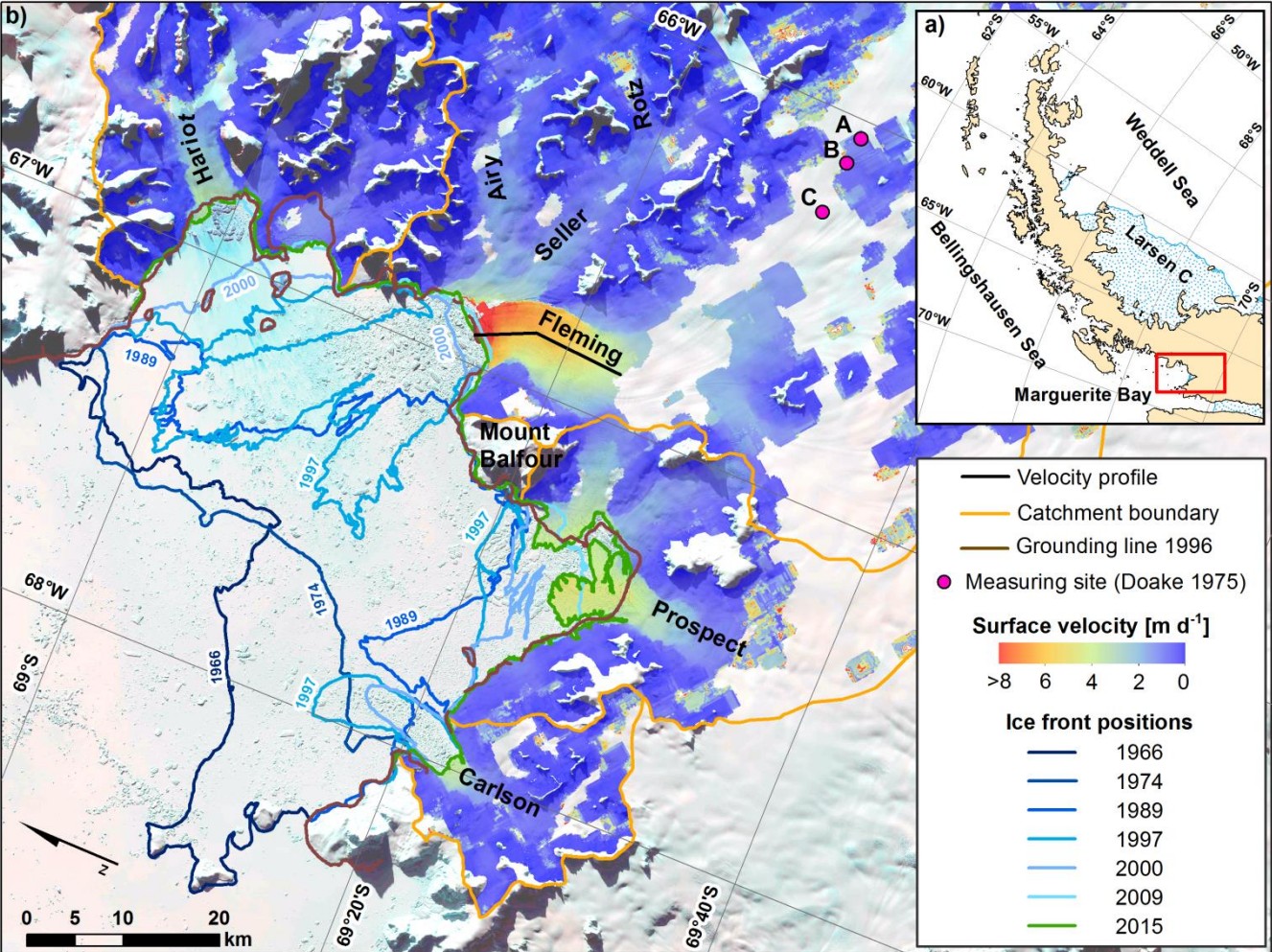

**Figure 1: (a)** Location of Wordie Bay at the Antarctic Peninsula. Map base: SCAR Antarctic Digital Database, version 6.0. **(b)** Surface velocity field and frontal positions of Wordie Ice Shelf between 1966 and 2015. Surface velocities were derived from Sentinel-1 acquisitions acquired on 28-08-2015 and 09-09-2015. Front positions (blue and green lines) were taken from existing datasets or manually mapped from calibrated and multi-looked SAR intensity images. For detailed information on the data sources used for the frontal delineation see Supplemental material Tab. S1. The grounding line in 1996 (brown line) was derived from ERS-1/2 double difference interferometry (Rignot et al., 2005; Rignot et al., 2011a). Black line: Extraction profile for the velocity time series presented in Sect. 5. Orange Line: Glacier system catchment boundaries from the SCAR Antarctic Digital Database, version 6.0. Pink dots: Sites of the velocity measurements undertaken by Doake, 1975 in 1974. Background: Mosaic of two Landsat-8 "Natural Colour" images, acquired on 2015-09-16 ©USGS.

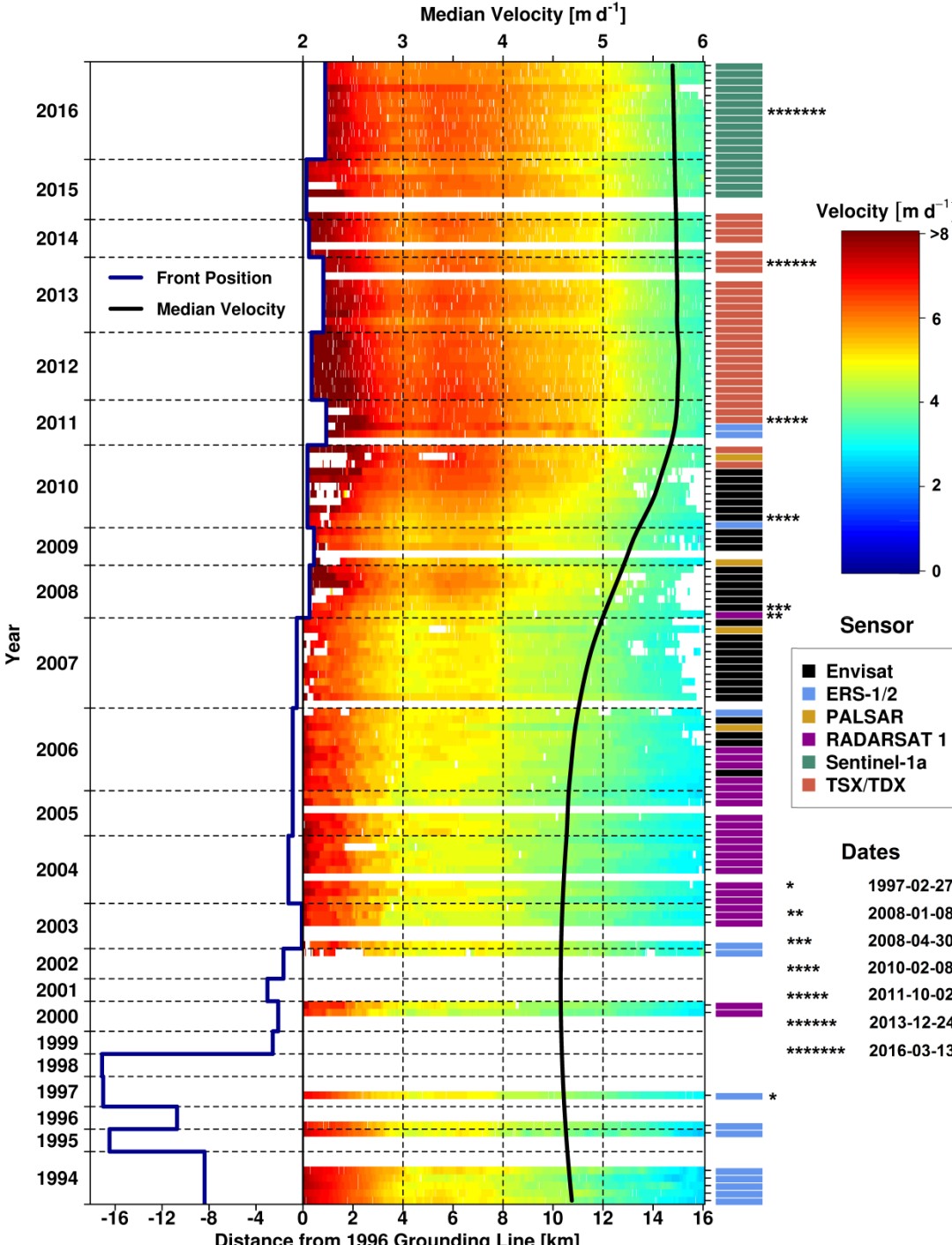

**Figure 2:** Left side (blue line): relative distance of the glacier front to the grounding line position in 1996. See Tab. S1 for data used for front mapping. Right side: velocity time series and smoothed median velocities (black line) derived from multi sensor SAR intensity tracking along a centreline profile on Fleming Glacier starting at the 1996 grounding line (Fig. 1). Dates mentioned in the text and in Fig. 3a, b are indicated with black stars. For all other dates see Tab. S3.

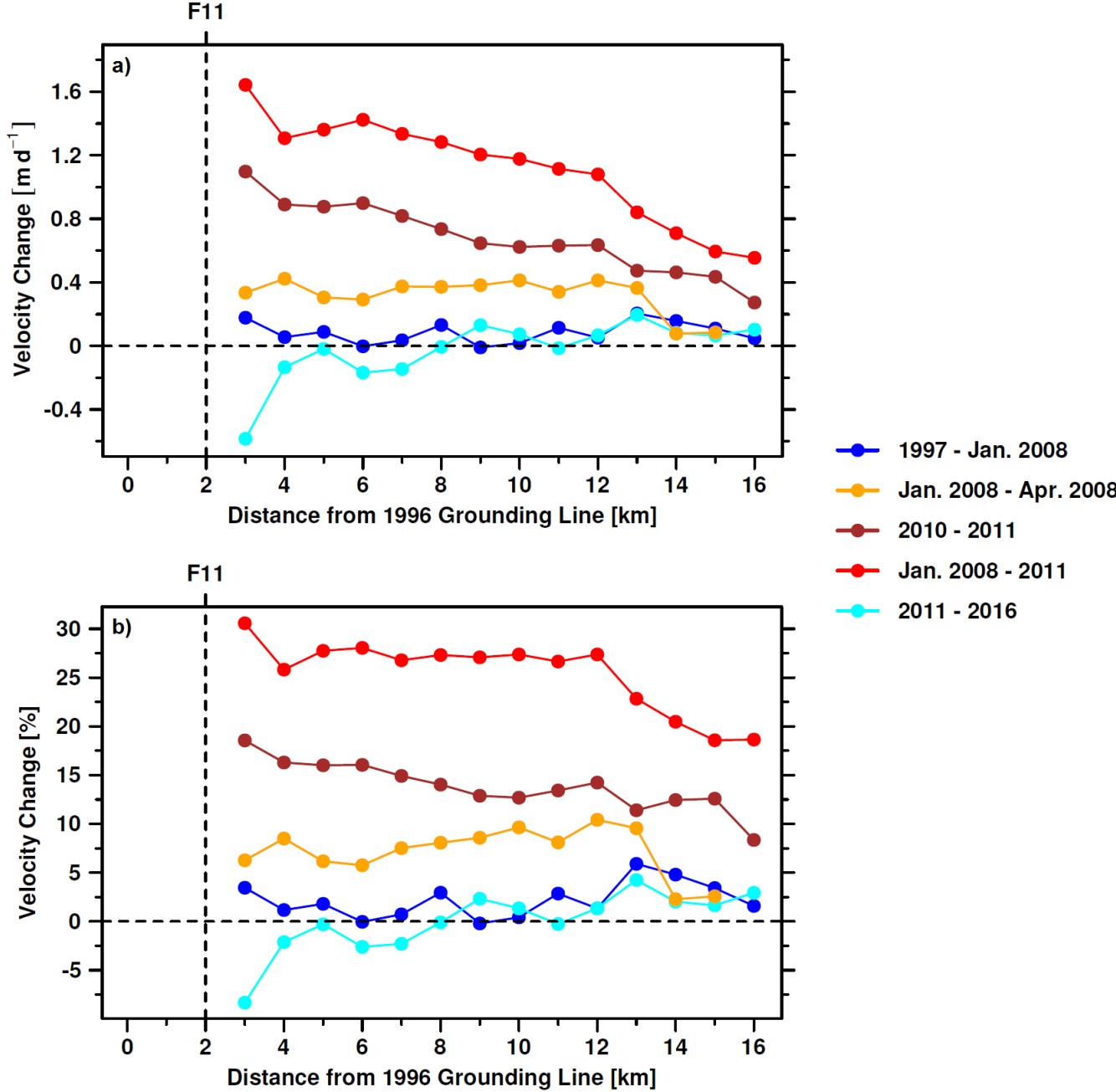

**Figure 3:** Absolute (**a**) and relative (**b**) velocity change along the centreline profile on Fleming Glacier (Fig. 1) for 1997-02-27 to 2008-01-05, 2008-01-05 to 2008-04-30, 2010-02-08 to 2011-10-02, 2008-01-05 to 2011-10-02 and 2011-10-02 to 2016-03-13. F11: front position in 2011. See Tab. S1 for data used for front mapping.

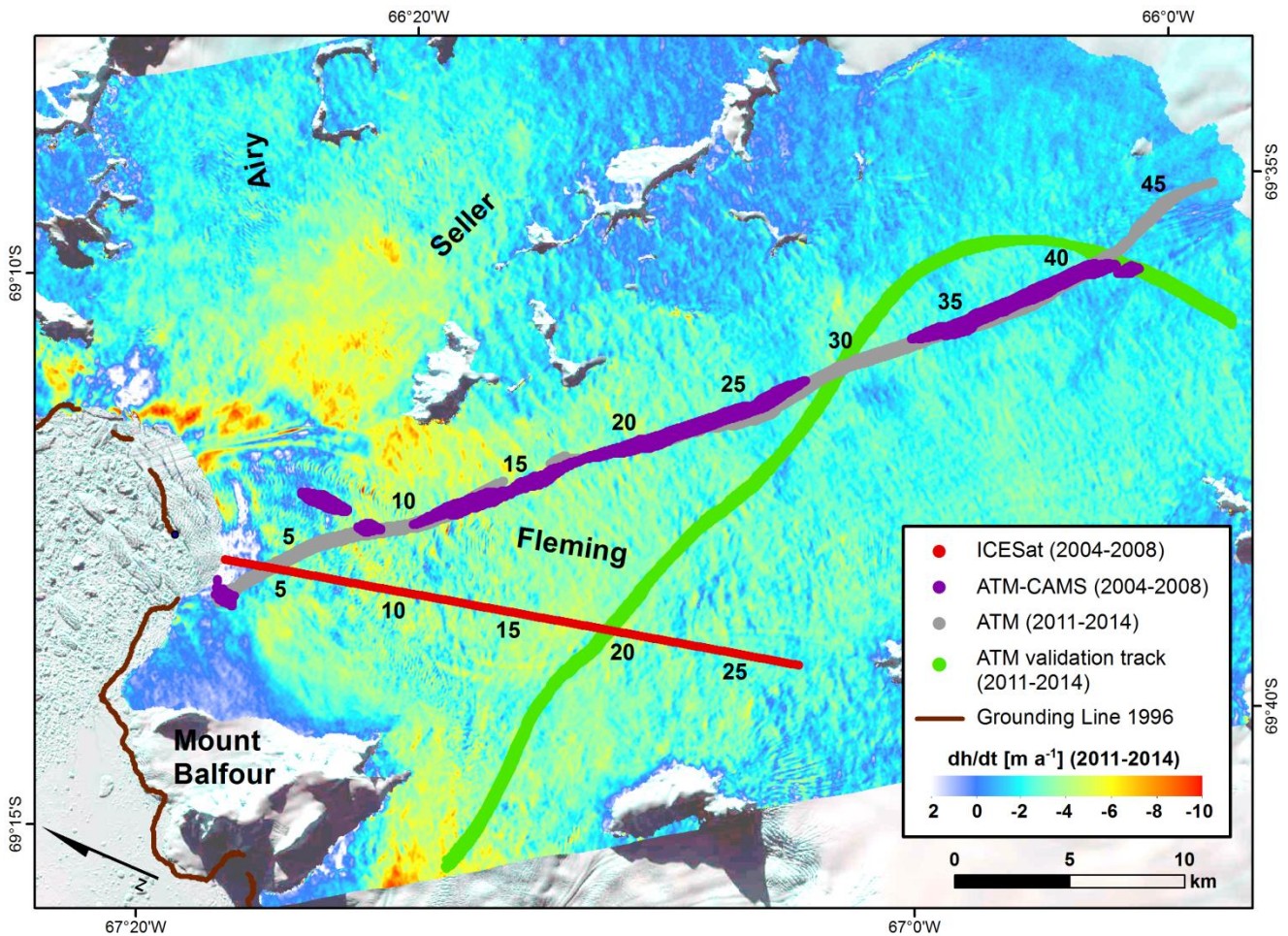

**Figure 4:** Glacier surface elevation change on Fleming Glacier between 2011 and 2014 derived from TSX/TDX bistatic and monostatic acquisitions. Red dots: ICESat track on 2008-10-04 taken as reference for ICESat 2004–2008 dh/dt calculations. Purple dots: common locations of PIB ATM and CAMS LiDAR measurements in 2004 and 2008. Grey dots: common locations of OIB ATM LiDAR measurements in 2011 and 2014. Green dots: common locations of OIB ATM LiDAR measurements in 2011 and 2014 taken for validation of the 2011–2014 TSX/TDX dh/dt measurements. Brown line: grounding line in 1996 from Rignot et al. (2005) and Rignot et al. (2011a). Numbers indicate distances to the 1996 grounding line. Background: Mosaic of two Landsat-8 "Natural Color" images, acquired on 2015-09-16 ©USGS.

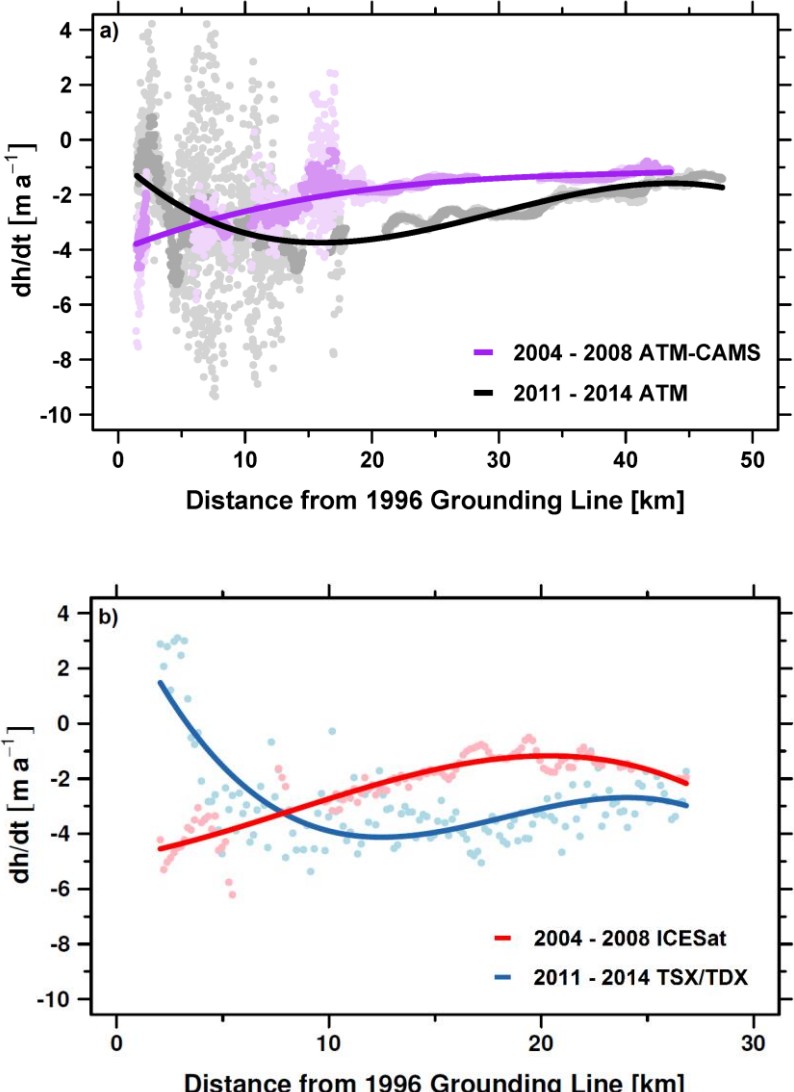

**Figure 5: (a)** Elevation change rates on Fleming Glacier 2004–2008 and 2011–2014 plotted against distance from the 1996 grounding line. Light purple dots: change rates from PIB ATM and CAMS LiDAR measurements in 2004 and 2008. Purple dots: median filtered elevation change rates 2004–2008. Light grey dots: change rates from OIB ATM LiDAR measurements in 2011 and 2014. Dark grey dots: median filtered elevation change rates 2004–2008. See Fig. 4 for flight path locations. Purple and black lines: Cubic functions fitted to the median filtered elevation change rates. **(b)** Elevation change rates on Fleming Glacier 2004–2008 and 2011–2014 plotted against distance from the 1996 grounding line. Light red dots: change rates from ICESat measurements in 2004 and 2008. Light blue dots: change rates between 2011 and 2014, extracted from the TSX/TDX dh/dt map along the 2008 ICESat track (see Fig. 4 for location). Red and blue lines: Cubic functions fitted to both datasets.

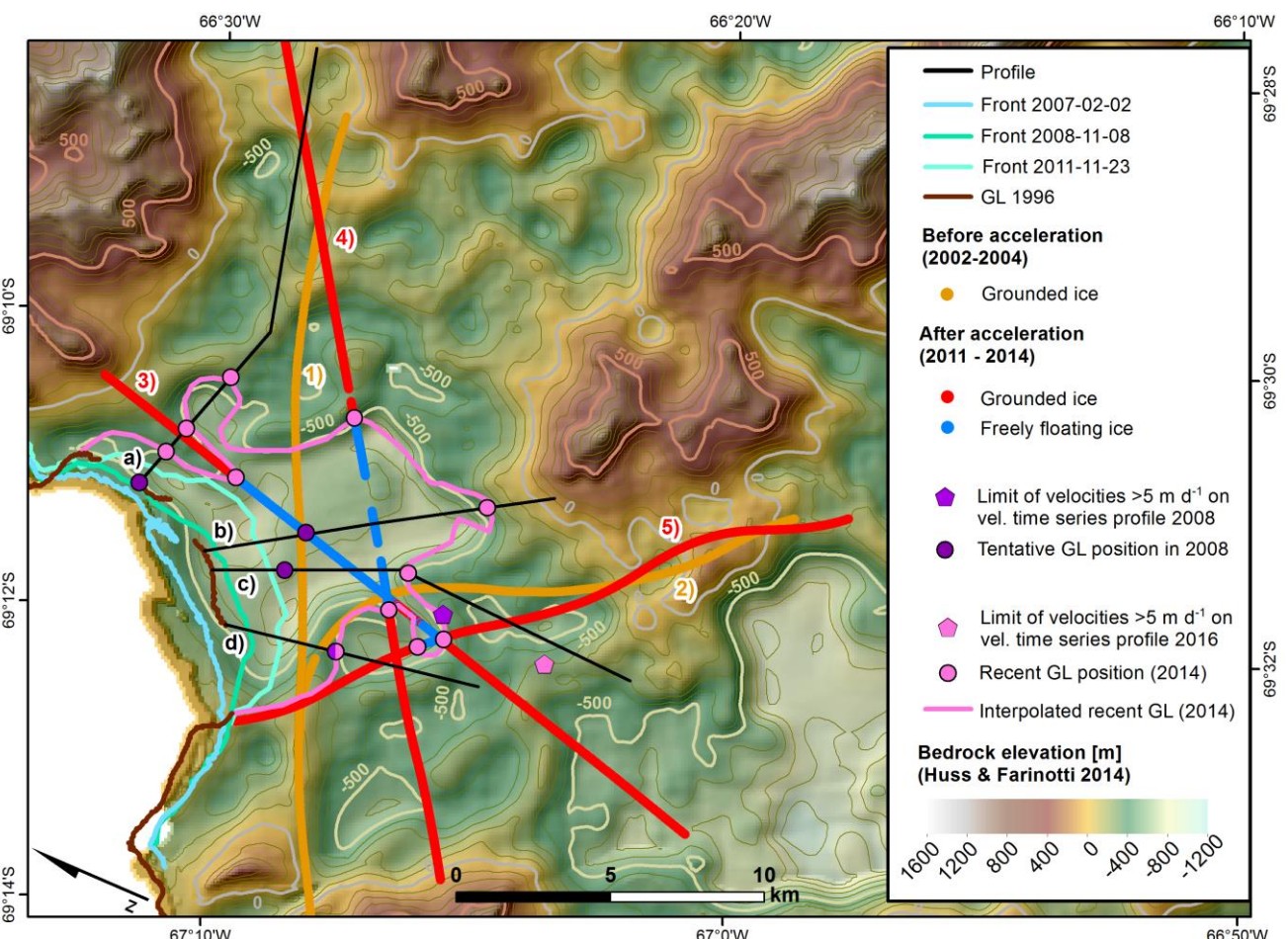

**Figure 6:** Floating area of Fleming Glacier and estimation of the recent grounding line. Black lines 1–4: Profiles for extraction of modeled bedrock elevations, surface velocities and elevation change rates (see Fig. S6 a–d). Elevation contours are shown at an interval of 100 m. Light blue and cyan lines: glacier fronts on 2007-02-02 and 2011-11-23. Brown line: grounding line in 1996 (Rignot et al., 2005; Rignot et al., 2011a).Orange dots: Grounded ice before acceleration as derived from PIB LiDAR and ice thickness data.. Dates of PIB flights: 1) 2002-11-26, 2) 2004-11-18. Background: bedrock elevation from Huss and Farinotti (2014). Blue and red dots: Freely floating and grounded ice after acceleration as derived from OIB laser altimeter and ice thickness data. Dates of OIB flights: 3) 2011-11-17, 4) 2014-11-16, 5) 2014-11-10. Purple pentagon: Location of the limit of velocities >5 m d$^{-1}$ on the velocity time series profile in 2008 (Fig. 1 and 2). Purple circle: likely grounding line location in 2008. Purple circles with cross: possible grounding line locations in 2008. Pink pentagon: Location of the limit of acceleration on the velocity time series profile in 2015 (Fig. 1 and 2). Pink circles: Estimated positions of the recent grounding line (2014) obtained from buoyancy calculations, surface velocities, elevation change rate patterns and/or modeled bedrock elevations on profiles a–d. Pink line: Final solution of the interpolated recent (2014) grounding line.

# Tables

**Table 1:** Sensors and data used in this study and their main specifications. Intensity tracking parameters are provided in pixels [p] in slant range geometry.

**SAR sensors**

| Platform | Sensor | Mode | SAR band | Repetition cycle [d] | Time interval [yyyy-mm-dd] — [yyyy-mm-dd] | Tracking patch sizes [p x p] | Tracking step size [p x p] |
|---|---|---|---|---|---|---|---|
| ERS-1/2 | AMI SAR | IM | C band | 35/3/1 | 1994-01-26 — 2011-06-29 | 64x320 | 5x25 |
| RADARSAT 1 | SAR | ST | C band | 24 | 2000-09-07 — 2008-01-17 | 128x512 | 5x20 |
| Envisat | ASAR | IM | C band | 35 | 2006-02-15 — 2010-10-10 | 64x320 | 5x25 |
| ALOS | PALSAR | FBS | L band | 46 | 2006-06-25 — 2010-11-23 | 128x384 | 10x30 |
| TerraSAR-X/ TanDEM-X | SAR | SM | X band | 11 | 2008-10-14 — 2015-01-30 | 512x512 | 25x25 |
| Sentinel-1a | SAR | IW | C band | 12 | 2015-08-28 — 2016-08-22 | 640x128 | 50x10 |

**LiDAR/Laser Altimeter**

| Mission | Sensor | Type | Wavelength [nm] | Footprint [m] | Dates [yyyy-mm-dd] | Estimated accuracy | Reference |
|---|---|---|---|---|---|---|---|
| Pre-IceBridge (PIB) | ATM (L1B) | LIDAR | 532 | 1 | 2002-11-26 2004-11-18 | 0.10 m | Krabill (2012) |
| ICESat | GLAS | Laser Altimeter | 1064 | 70 | 2004-05-18 2008-10-04 | 0.4 m | Zwally et al. (2014) |
| CECS/FACH | CAMS | LiDAR | 900 | 1 | 2008-12-07 | 0.20 m | Wendt et al. (2010) |
| Operation IceBridge (OIB) | ATM (L1B) | LiDAR | 532 | 1 | 2011-11-17 2014-11-10 2014-11-16 | 0.10 m | Krabill (2010, updated 2016) |

**Ice Thickness**

| Mission | Sensor | Type | Bandwidth [MHz] | Sample spacing [m] | Dates [yyyy-mm-dd] |
|---|---|---|---|---|---|
| Pre-IceBridge (PIB) | ICoRDS-2 | Radar | 141.5-158.5 | ~130 | 2002-11-26 |
| Pre-IceBridge (PIB) | ACoRDS | Radar | 140-160 | ~30 | 2004-11-18 |
| Operation IceBridge (OIB) | MCoRDS | Radar | 180-210 | ~15 | 2011-11-17 |
| Operation IceBridge (OIB) | MCoRDS 2 | Radar | 165-215 | ~15 | 2014-11-10 2014-11-16 |

**Table 2:** Comparison of surface velocities and flow directions at Fleming Glacier obtained by an optical survey in 1974 (Doake, 1975), SAR interferometry in 1996 (Rignot et al., 2005),GPS measurements in 2008 (Wendt et al., 2010) and SAR intensity tracking in 2013. Velocities and flow directions in 2013 were derived from intensity tracking applied on two TSX/TDX acquisitions on 2013-12-19 and
5  2013-12-30. Velocities in 2015 are from (Zhao et al., 2017). For the locations of the measuring sites see Fig. 1.

| Location | 1974 | | 1996 | | 2008 | | 2013 | | 2015 | |
|---|---|---|---|---|---|---|---|---|---|---|
| | Magnitude | Direction | Magnitude | Direction | Magnitude | Direction | Magnitude | Direction | Magnitude | Direction |
| | m a$^{-1}$ | ° | m a$^{-1}$ | ° | m a$^{-1}$ | ° | m a$^{-1}$ | ° | m a$^{-1}$ | ° |
| A (69.505° S, 66.049° W) | 146 ± 4 | 277 ± 5 | 244 ± 10 | 285 | 205.5 ± 0.02 | 275.8 ± 0.1 | 205 ± 22 | 272 | 271 ± 20 | NA |
| B (69.502° S, 66.123° W) | 175 ± 4 | 272 ± 5 | 271 ± 10 | 287 | NA | NA | 244 ± 22 | 271 | 299 ± 20 | NA |
| C (69.500° S, 66.267° W) | 201 ± 4 | 283 ± 5 | 306 ± 10 | 300 | 312.8 ± 0.04 | 286.3 ± 0.1 | 323 ± 22 | 284 | 356 ± 20 | NA |