# Peer review of "Recent dynamic changes on Fleming Glacier after the disintegration of Wordie Ice Shelf, Antarctic Peninsula"

_The Cryosphere, 2017_

## Referee Comment (RC1) · Anonymous Referee #1 · 13 Jul 2017

General Comments:

Friedl et al. present a study on glacier retreat and changes in ice flow for Antarctic Peninsula outlet glaciers after disintegration of Wordie Ice Shelf. The work is based on analysis of remote sensing data from various sources. It extends the period of observations on frontal retreat, flow acceleration and glacier thinning that was reported by Rignot et al. (2005) and Wendt et al. (2010) up to the year 2009. Of particular interest is the production of a close time series of surface velocities, up to the year 2016, including the filling of gaps from previous years. The analysis of surface elevation change, comparing the change 2011 to 2014 to that of previous years, and the time

series of frontal retreat are also very relevant for describing glacier behaviour. The review on previous work and the presentation and interpretation of the observations are presented in coherent manner at large. However, there are various individual points that are not well explained or questionable. The presented data sets are very useful for characterizing the glacier behaviour during the last two decades, but the discussion and conclusions focus on the description of the observed phenomena and do not provide any substantial new insights into the processes leading to the observed changes.

Specific Comments:

P1, L16: The conclusion on "pronounced basal melt at the grounding line" is not based on any direct observations for these glaciers.

P1, L19: The length of the centreline (for which this values is valid) should be specified

P1, L21: Fig. 5 shows in the downstream part of the profiles for 2011-2014 thinning rates that are smaller than for 2004 to 2008. 60% to 70% higher rates are only evident for a subsection of the profile shown in Fig. 5.

P1, L26: Hardly possible that the glaciers draining into Wordie Bay have "a huge potential for an increase in sea level rise".

P2, L4: 4.21 Gt/a refers to Larsen-A and Prince-Gustav-Channel glaciers, 2011-2013.

P2, L13, L14: Rignot et al. (2014), Suppl. Material, show for Wordie Ice Shelf clear dominance of calving losses compared to ablation, rather than "basal melt exceeding the ablation induced by calving".

P2, L15: Please explain the "small coastal atmospheric and oceanic processes".

P4, L4: Date and source for the Bedmap2 DEM section over the study area should be mentioned (may have some impact for geocoding and analysis of surface elevation change).

P4, L24: May mention here that the OIB and Huss bedrock data are compared in Fig.

S5.

P5, L15: point clouds of differential elevation measurements are shown in Fig. 5. (not Fig. 4).

P5, L34: It is not meaningful using a low resolution DEM from a different epoch for performing DEM differencing with high resolution TanDEM-X data, if an up-to-date high resolution DEM from TanDEM-X is available.

P6, L19 to L28: The differences between ATM and (uncorrected) TanDEM-X rates of elevation change (up to 2m/a) need further explanations and checks. 2m/a corresponds to 6 m difference for the 3 year time span. The area for the profile is located in the percolation zone. Typical values for TanDEM-X penetration bias in the percolation zone are about 4 m (see e.g. Wessel et al., ISPRS Annals. VL-III-7, doi:10.5194/isprsannals-III-7-9-2016). If the morphology of the snow and firn medium is the same on both dates, the penetration bias cancels out for DEM differencing. A difference in dh/dt of 6 m versus optical data can only be explained if the snow morphology is completely different on both dates (e.g. melting surface snow without penetration vs. completely frozen snow volume). This should show up clearly in the backscatter signatures.

P7, L23: "determent" is probably the wrong word here.

P8, L9: It seems there was some slowdown after 1994, and gradual acceleration started in 2003. The selected graphic representation is not favourable for capturing such features.

P9, L7: "Figure 5 shows that prior to the speedup (2004–2008) Fleming Glacier was already affected by pronounced surface lowering." No elevation change data prior to 2004-2008 are shown in Fig. 5.

P9, L28: "a vast part of the formerly grounded glacier tongue". "vast part" is a subjective impression; be specific.

P10, L14: "A possible location of the grounding line after the initial ungrounding in

2008….". Confusing statement. Was the area floating before 2008 and then became grounded again?

P10, L25: The Rothera station data (near Wordie I.S.) do not show a cooling trend for recent years. Oliva et al. (Science of the Total Environment 580, 210–223, 2017) report higher mean annual air temperature for 2006 to 2015 than in the previous decades.

P10, L31, L32: Here it is stated that "flow acceleration usually affects both the floating and the grounded part of the glacier, but is largest near the grounding zone", and also "This is consistent with our observation of a highest relative speedup by $\sim$32–35 % between 7 and 11 km upstream". If the highest speedup is 7 to 11 km upstream, the second statement is not consistent with the first one.

P11, L4: Any direct observations supporting the statement that basal melt is particularly effective in the grounding zone?

P11, L6: In which respect is the bedrock topography "unfavourable"?

P11, L7, L8: Fig. 5 shows thinning rates only for grounded ice. This does not provide any clear link to basal melt rates of the floating part in the grounding Zone

P12, L12, L13: "Our data suggest that enhanced basal melt at the grounding line due to increased shoaling of warm CDW most likely played a major role for the recent changes at Fleming Glacier." This conclusion is not based on the measurements presented in this study, but rather on measurements in other locations and reported in other publications.

P12, L21: Please explain the expected "fatal effects on the stability and sea level contribution".

P19, L5: Fig. 1 caption. Please check the dates for Sentinel-1 data. One year time span for velocity retrieval?

P24, L5: Fig. 6 caption. "Fulfillment of floating condition". Does this refer to grounded

or floating ice?

P24, L7: "Fulfillment of the hydrostatic equilibrium condition". Same question as for P24, L5.

L25, Table 1: SAR, column 6 shows single dates (not "Time Interval")

Supplement Table S2: Pixel size should be accurately specified in along track (or LOS) and across track direction. These can be quite different even for a single sensor (e.g. Sentinel-1).

---

## Short Comment (SC1) · 16 Aug 2017

The paper adds really great new data and understanding on the recent evolution of the Fleming glacier system. Too late for the authors to see before submission was the new paper of Zhao et al epsl which has a different focus but some potential to discuss and compare results. I think the new manuscript by Gardner et al is also relevant.
* * *

---

## Author Comment (AC1) · 28 Aug 2017

We thank Matt King for his informative comment. We are aware of the new paper by Zhao et al. and we will definitely discuss and compare their results on elevation change of the Fleming glacier system with our findings in a revised version of the manuscript. The paper of Gardner et al. will also be considered if it has been accepted for publication by then.
* * *

---

## Author Comment (AC2) · 25 Sep 2017

We would like to thank the anonymous referee for the detailed and constructive review of our paper. This is just a short statement to his comments. A detailed answer together with a supplementary pdf containing the changes in the manuscript will be provided after we have received the second review. The individual points mentioned by the referee are clear and corrections of these points will contribute to a further improvement of the paper. We will address these points accordingly in a revised version of the paper. We would like to point out, that meanwhile an interesting new paper by Walker and Gardner (2017) was published that intensively investigated melt at the ice-ocean

boundary in Wordie Bay. Their investigation was coupled to global atmospheric circulation patterns. One of their key questions "Why was the response of Fleming Glacier to western Antarctic Peninsula wide periods of oceanic forcing far much stronger than the response of any other glacier in the region?" remained unanswered. Our analysis provides unprecedented details on the recent dynamic changes of Fleming Glacier. In contrast to the paper by Walker and Gardner that only analysed three stacked velocity fields from 2008, 2014 and 2015, we provide a very dense time series of satellite data for surface velocities since the availability of SAR data in the region. The data enabled us to identify two pronounced phases of acceleration for the first time: a sudden acceleration and upstream propagation of high velocities in 2008 and a phase of further gradual acceleration and upstream propagation between 2010 and 2011. Additionally, we investigate elevation changes derived from ICESsat and imaging satellite sensors, and conduct buoyancy calculations from Operation Ice Bridge data for the center of the glacier tongue. This allows us to better determine when speed-up occurred in response to ungrounding and where the new grounding line is located. Interestingly the timing of the acceleration phases is well correlated to periods of distinct warm water intrusions into Marguerite Bay, identified by Walker and Gardner (2017). Hence, our analysis provides new insights in the timing of the processes and reasons for the observed changes in Wordie Bay. Thus, our data set is highly complementary to the work by Walker and Gardner (2017) and our interpretation supports their analysis, but enabling a much better temporal constraint of the processes. We have summarized our interpretation in a small view graph (Fig. 1).
* * *
[Figure]

**before 2008**

- Dynamic thinning due to ice shelf loss
- Basal melt at grounding line by ASBW

Stable velocities

Grounding Line 1996

Pinning point exerts buttressing

**2010 - 2011**

- Gradual grounding line retreat
- Gradual decrease in buttressing

Gradual acceleration

New Grounding Line

**2008**

- Loss of pinning point
- Part of ice tongue goes afloat
- Decrease in buttressing

Acceleration

New Grounding Line

**today**

- Increased velocities
- Increased dynamic thinning
- Steepening
- Mass loss

Grounding Line

Basal melt at grounding line and beneath glacier tongue

**Fig. 1.**

---

## Referee Comment (RC2) · Anonymous Referee #2 · 25 Oct 2017

Friedl et al. present a dense time series of surface velocities and elevation change rates of the Fleming Glacier system in the past two decades from multi-source remote sensing data. This study identified two significant acceleration phases occurred in 2008 and 2010-2011 after the Wordie Ice Shelf has nearly disappeared in front of the Fleming Glacier. The determination of floating area based on the hydrostatic height anomalies, bedrock data, elevation change rates, and acceleration phase is very helpful to explain the rapid dynamic changes after 2008. However, some points in this study are not well explained and need further rewording. Some minor wording and grammar errors also need to be fixed.

[Figure]

Here are some general comments:

1) It is not clear about the epoch of the 2008 grounding line position. Please clarify it, particularly regarding locations in Fig. 6 and Fig. 6S.

2) The method used to determine the grounding line position in 2008, 2011 and 2015 from the velocity data, surface-thinning rate, and the bedrock data is not clear enough in the method (Sect. 4.3, P7, L19-27) and result section (Sect. 5.3), especially regarding the determination of the grounding line position in 2008. From Fig. 6, it is appeared that the 2015 grounding line is mainly based on the ridges of modelled bedrock combined with the hydrostatic height anomalies along OIB flight lines. However, profiles in Fig. S5 show that how much the Bedmap2 bedrock and Huss and Farinotti's modelled bedrock can differ from the PIB and OIB measurements. Then how reliable are the hills used to decide the grounding line location? In the analysis of Fig. 6S, it is better to explain how the different evidence (velocity profiles, elevation change profiles, hydraulic height anomalies, and modelled bedrock) is being combined to determine the estimated 2008 and 2015 grounding line. If they are not consistent with each other, please explain the final selection.

3) This study (P11, L27-29) concludes that the basal melting driven by ocean-warming is the dominant trigger of the grounding line retreat and glacier acceleration. However this lacks direct evidence, namely the basal or frontal melting rates underneath the ice shelf. Given the little ice shelf left in 2008, it is not very convincing to say that the basal melting underneath the ice shelf dominates the ungrounding process.

Specific Comments:

P1, L8: change to "... regions most affected by climate change"

P1, L14: The finding about glacier ungrounded in 2008 between January and March has not been confirmed by observation or measurements. Please reedit this sentence.

P1, L24: "Currently, the tongue of Fleming Glacier is grounded in a zone of bedrock

elevation of -400 m" This statement is not mentioned in the main text. Please state this in the main text and make the elevation color bar in Fig. 6 clearer for reading the elevation values. It will be helpful to add elevation contours.

P1, L26: You can't say "this endangers" "a huge potential". Add a suitable verb before "a huge potential". "huge" is not appropriate here.

P2, L4: Add "be" before the number "4.21".

P3, L3: "former tributary glaciers" is confusing since the glaciers still exist, Suggest instead "... originally fed by several major tributary glaciers (Fig.1). Among these, Fleming Glacier is ...".

P3, L10: "Here the ice shelf was temporarily grounded and stabilized ..." is confusing. Does "Here" mean at a time point or at the location of the pinning points? Is there evidence of grounding of the previously floating Wordie Ice Shelf? Do you mean it was stabilized temporarily? Do you mean "the next rapid break up event" occurred in 1989? If yes, please specify it.

P3, L18-19: Please add the ice front position in 1997 and 2000 in Fig. 1. It will help reader to understand your statement here.

P4, L21-25: The information about the modelled bedrock topography from Huss and Farinotti 2014 is not enough. Given its role, indication of origins and uncertainties should be included in the paper.

P7, L18: Modify "Sect. S2" to "Sect. S3".

P7, L22: "mitigates" is hardly correct for 90

P7, L22-23: It is not clear how you decide the grounding line position from the surface velocities. Please clarify. If you define the grounding line based on where the maximum velocity increase occurred, please specify it with relevant references.

P7, L23: replace "determent" - do you mean "determinant"?

P8, L18: Please comment on the higher velocity changes in the 2011 glacier front in Fig. 3 and why it is ignored.

P8, L23-24: Suggest updating Table2 by adding the velocity in 2015 from Walker and Gardner, 2017 at three sites.

P8, L27: As I could tell, the highest ice thinning rates for Fleming Glacier from Fig. 4 (orange color) is higher than 6 m a-1.

P9, L2: "For the location of the data see Fig. 4" → "The location of the data is shown in Fig. 4".

P9, L13: Missing space after the full stop.

P9, L19-20: It would be useful to compare your thinning rate with that from Zhao et al., 2017 and Walker and Gardner, 2017.

P10, L1-3: The "We extracted" part of the statement should explicitly refer to Fig. 6S, and the location of the four profiles should refer to Fig. 6.

P10, L3-4: For "The plots", you mean Fig. 6 or Fig. S6? It would be clearer to say "Those profile plots (Fig S6 1-4) suggest . . .". Which glacier or profile are you talking about for the 2008 grounding line position?

P10, L5-6: "possible" and "likely" are same to me. It's not clear how the grounding line positions in 2008 are decided on Profile 2, 3, and 4.

P10, L7-15: Is detailed discussion of a supplementary figure in the main text proper? Consider including Fig. S6-3.

P10, L18-20: About "no significant acceleration since 1996 . . .", if you compare the velocity at three sites with the 2015 velocity from Walker and Gardner, 2017, you will find some speed up from 1996 to 2015. Alternatively you need to qualify with "by 2013".

P10, L27-28: Besides the surface melt, basal melting water generated from the basal
frication heating could be another trigger of enhanced basal sliding. It's hard to rule out the possibility of enhanced basal sliding unless you have further evidence.

P10, L30-31: If this is your method to decide the grounding line position from the velocity data, please move this sentence to Sect. 4.3.

P11, L12: Please cite references for the Twaites Glacier and the Pine Island Bay.

P24, L4: Modify "Linens" to "Lines".

Supplement Material

It is better to use consistent zero distance mark for the various profiles in both the main text (Fig. 2, 3, 5, 6) and the supplementary material (Fig. S6). The use of different starting points (the 1996 grounding line position or the 2007 ice front position) for describing the distance is confusing. If it is too difficult to shift the origin in the supplementary figures, the GL96 grounding line needs to be clearly marked.

P12: Modify the legend of a) and b) from "Ice Surface/Bottom elevation (OIB)" to "Ice Surface/Bottom elevation (PIB)".

P15, L3: Modify "profiles 1-5" to "profiles 1-4". Also consider labeling figure parts as S6.a-d for consistency with other sections. Anyway add labels to the profile figures.

P15, L4: Can't find OIB ice surface/bottom in those figures.

P15, L5: Modify "grounding line" to "GL".

P15, L6-7: Can't find Green, Red and Blue lines in those figures.

P16, L7-8: It's hard to tell the 2008 grounding line position upstream the 2011 front without direct evidence. Please explain it.

P16, L10-11: Please add GL11 in Fig. S6.2.

P17, L3: Move "in 2008" after "the grounding line".
P17, L5-6: Along Profile2, it's hard to tell this hill in Fig. 6. Considering the uncertainty of bedrock ïïjĹyellow and brown line in Fig. S6.3), is it possible that this smaller hill is not physically real? On P18, L5-6 you suggest that "the topography may be distorted in the modelled bedrock data".

---

## Editor Comment (EC1) · B. Wouters (Editor) · 16 Nov 2017

**Review of: Recent dynamic changes on Fleming Glacier after the disintegration of Wordie Ice Shelf. Friedel et al., 2017**

**Summary**

The topic of this paper, Fleming Glacier, is extremely interesting given the significant changes reported by previous publications, so I was very keen to read these new results. The authors present new ice velocity, elevation change and grounding line position data acquired from a range of airborne and satellite based instruments. The ice velocity results are nice, and I believe constitute the most complete time series of velocity measurements over Fleming glacier which provides new insight into the timing of ice speedup on this sector.

However, after reading the manuscript there are a number of major flaws with the methods employed to derive surface elevation change and to measure the grounding line retreat. As it stands, the problem with the techniques make it highly likely that both the magnitude of the elevation change signal, and the grounding line retreat, may not be correctly reported in this paper. For example, cross calibrating the DEM elevations over sea ice, which varies annually and can range in thickness from 0-5m, is extremely unsatisfactory. Moreover, even if this correction is accepted, the authors estimate that known X-band penetration bias can account for ~50% of the dh/dt signal across the basin. Even if the estimate of elevation change is accepted, the associated error measurements do not reflect spatial variability in the data quality, and are unrealistically small given the spread of the raw data. For example, it is stated that the CAMS-ATM dh/dt data has an error of 0.2m/yr in regions where the point measurements at the same location range from -0.5 to -7 m/yr. These errors must therefore be revised. Regarding the estimate of grounding line retreat, the technique is unproven and is un-validated in this paper. Even if the authors demonstrate that the technique can be trusted, as it stands the results presented in this paper are contradictory because regions measured to be grounded are located within the new floating ice shelf area.

On top of these technical issues, I have a few more minor concerns about the use of scientific terminology throughout the paper, and the overly simplistic nature of the analysis and discussion sections. The specifics of these and other concerns are documented in detail below.

My criticisms of the methods and results presented in the paper are major, and will be time consuming and require a significant effort to properly address. The implications of these concerns is that I believe the magnitude and spatial pattern of the elevation change and grounding line retreat data may be incorrectly reported in this paper. This is significant as I do not have confidence in two of the three core datasets presented, therefore, it is my recommendation that this paper is not suitable for publication in its current form.

**Specific Edits**

P1 L21 – Edit 'far upstream of the glacier', using this as a location is ambiguous, better use a fixed reference location, like the calving front in 'x' year.
P1 L26 – Edit paper to quantify 'much larger ice masses'.
P2 L4 – Edit missing to 'be'.
P2 L7 – Edit ice 'shelf' tributaries.

P2 L30 – Edit 'explain' to 'investigate'. I'd argue at this point the authors haven't demonstrated they can explain the observed signal.

P3 L23 – Poor wording, edit out 'got' both times.

P4 L2 – Edit 'dynamic' and check use throughout paper. Previous publications (e.g. Rignot et al 2005) have demonstrated that Fleming exhibited dynamic imbalance during their study period, but as the authors state this paper spans a longer time period, and at this point it hasn't been proven that Fleming is dynamically imbalanced for the full duration of their study period.

P5 L6 – More informative to state what % of each velocity image is removed during the filtering process as this should be an indicator of the quality of the tracking output. The filtering should remove 100% of the unreasonable results, otherwise its not a very good filter!

P5 L16 – 0.2 m is the accuracy of the original point measurements; the authors are using the dataset after re-gridding it so state the accuracy of this dataset instead or as well as the accuracy of the raw data. The accuracy is also different for different sensors, so the authors should provide statistics for each dataset.

P6 L2/3 – This is correction extremely unsatisfactory. Sea ice is a complex parameter, and is certainly not a stable/constant reference surface for precise cross calibration of elevation measurements. In the Antarctic, sea ice can range from 0 to 5 m thickness with very large spatial and temporal variability, snow depth on sea ice is not routinely measured but can account for half the thickness retrieval, and ocean height varies with tides, atmospheric pressure etc. When deriving the correction, the authors have not attempted to account for inter-annual variability in sea ice thickness so this must be addressed before any confidence can be had in the elevation change measurements. This is critical because the range of thickness variability is the same order of magnitude as the dh/dt signal calculated from the DEM differencing. The authors must revise the manuscript to characterise the temporal variability of the sea ice over which they are cross calibrating the DEM's, and to rule out any influence from this factor on the end elevation change. If this effect can be proven to be negligible, the authors should also state the size of the correction, and which DEM was adjusted, in the manuscript. Having said all this, I suggest the authors dont cross calibrate the DEM's over sea ice at all as it hugely reduces the confidence that I believe we can have in these results.

P6 L20 – The authors calculation that up to 2 m/a of the thinning rate can be attributed to TSX DEM penetration bias. This is a huge error which accounts for ~50% of the dhdt signal present across the majority of the basin. The ICESat and ATM tracks that this error was calculated from have extremely limited coverage, and don't pass through the region with the highest thinning rates, therefore its possible that this number might even be an underestimate. For example, other studies have shown that the penetration bias in DEM's derived from TSX/TDM data over snow covered terrain can be as large as 4m (e.g. Dehecq et al 2016), which is the same magnitude as the dh/dt signal presented in figure 4. The large size of the known errors relative to the size of the signal, combined with the limited data that has been used to characterise the error makes it very difficult to have confidence in the thinning rates presented here. Other auxiliary datasets such as atmospheric temperature data, or SAR backscatter images might also be used to characterise the onset and spatial pattern of melt in the study area. The authors description of how they have accounted for this source of error is cursory given its size relative to the dh/dt signal in the study area. As surface melt and therefore penetration is known to have large spatial variability, I recommend that the authors revise their approach to account for this spatial variation across the basin, as a polynomial fit derived from a single track of airborne data will definitely not capture the magnitude or pattern of this effect across the study area.

P6 L27 – Assuming an error of 0.2m/a just because its one order of magnitude higher than the direct inter-comparison with the ATM data the dh/dt was calibrated against isn't satisfactory. Errors are spatially variable, so the authors should revise their approach.

P6 L33 – Why use a 35 m buffer if the ICESat footprint is known to be 70 m? I recommend the authors use the same footprint size as the aim is to do the most direct comparison possible.

P7 L15 – Cite a reference for the source of the firn density correction variable.

P7 L25 – As far as I'm aware, this method of detecting grounding line position has not been proven in peer reviewed literature. Although the logic behind it is reasonable, (i.e. if the ice is in hydrostatic equilibrium it must be floating), factors such as the spatial resolution and error on each input dataset will severely limit the sensitivity of the technique for detecting grounding line position, let alone change in grounding line position. To be convinced that the technique works, I recommend grounding line retreat from this method is evaluated against known retreat rates, in the Amundsen sea for example. If suitable data isn't available to validate this technique in another area, then alternatively a proven technique can be employed to evaluate the hydrostatic technique in this study area. For example, ERS-2 SAR data with a 3-day temporal baseline was acquired in this area in 2011, so if coherent, this should be used to produce a grounding line estimate from the proven quadruple difference interferometry technique (Rignot et al, 1998). At a minimum the authors must state the error on their estimate of grounding line position from hydrostatic height anomaly, and it follows that if the uncertainty on the measurement is greater than the change in position assumed, then the method is not viable.

P8 L3 – The Figure 2 z-scope is not easy to interpret. I suggest the authors re-plot this information as a standard x/y line plot of the time series of flow line ice speeds, with an inset showing change in calving front position.

P8 L14 – Poor sentence wording. Edit.

P8 L28 – 8 to 14 km upstream of which location. Edit sentence to be more precise.

P8 L30 – Edit text to quantify 'lower parts'.

P8 L31 – Edit manuscript to remove all 'could be detected' wording. You are stating what results you have observed, so it 'has been detected', not the less affirmative 'could be'.

P9 L4 – The scatter on figure 5a is very large and must be addressed given that it is significantly greater than the previously stated errors. I recommend that a) distance markers are annotated onto the ATM and ICESat track locations on Figure 4 so we can see how this corresponds to the x axis distance scale in Figure 5. My interpretation of figure 5a is that the elevation change measurements are unusable between 0 and 20 km of the grounding line, which looks like its about up to the 'g' on the Fleming annotation on figure 4. This is the key area of interest, so vastly limits the usefulness of these datasets. B) state the method used to calculate the lines of best fit, e.g. moving average, polynomial fit? How has the clearly erroneous data been removed? C)

P9 L10 – Based on figure 5a, stating that the CAMS-ATM show elevation change of 4.1 ± 0.2 m/a is not credible. The raw data shown in figure 5a shows that at this location the elevation change ranges between -0.5 to 7 m/yr, so the error of 0.2m/yr is effectively meaningless. Please revise the error estimate here, and throughout the rest of the results paragraph.

P9 L12 – The fact that Fleming is thinning between 04-08, doesn't prove that the catchment hasn't reached an equilibrium since shelf collapse in 1989. The two effects may be entirely uncorrelated, so although its possible, without a continuous dataset I don't think it can be proven one way or the other. I recommend the authors revise this wording. Changing 'shows' to 'might suggest' would be more factually correct.

P9 L22 - Although I don't like the method, it's clear how the hydrostatic equilibrium has been calculated along the airborne tracks. Can the authors clarify what method they have used to draw the grounding line connecting the dots in Figure 6? For example, according to their own data, a section of grounded ice on track 'c)' is included in the now 'floating' area. I recommend the authors revise the line as their data shows it isn't correct.

P9 L30 – Based on the above comment, the number stated for the area of the floating shelf

will also need to recalculated.

P9 L33 – Quantify 'several km'.

P10 L25 – Although Turner et al 2016 shows that the long term air temperatures are decreasing the situation may be more complicated than that, and Sundal et al (2011) showed that a simple linear relationship between melt water vs lubrication is not currect, as melt induced speed-up can be offset by drainage efficiencies. I'd revise this text to avoid oversimplifying these relationships.

P11 L4 – Remove sentence about basal melt. This hasn't been measured in this study so is just a generic assumption, and no reference provided to previous study evidencing statement.

P11 L8 – Same statement as above re basal melt inference.

P11 L11 – Really poor sentence wording. Edit to be more diligent with regards to terminology. Stability is a specific process, i.e. 'unstoppable' retreat that will continue to propagate even if environmental conditions returned to their original state. Glacier imbalance and grounding line retreat can occur stably. I haven't seen evidence presented in this paper of about the likely future instability of Fleming, so tighten up language.

P11 L12 – edit Thwaites

P11 L18 – Again I feel the analysis here is overly simplistic. Wouters et al were the first to present the rapid thinning rates and mass loss from the Western Palmer Land region, but subsequent publications (Hogg et al 2017) have shown that only ~30% of this should be attributed to ocean induced dynamic imbalance. Revise text to reflect known complexity. Discussion general – the authors have stated results from other regions of WAIS/AP, however this really isn't tied very coherently into how this impacts on the results they have presented on Fleming.

P12 L10 – as previously stated I do not think the authors have proven dynamic instability on Fleming. Imbalance maybe, but instability, no. Equally, attributing the signal to ocean induced dynamic forcing without properly evaluating any oceanographic or atmospheric data is poor. These interlinked processes are very complex, and really hard to disentangle. Although its entirely plausible that ocean forcing is responsible, I do not think the analysis presented in this paper has proven it.

P12 L18 – 'ocean forcing is likely to continue'. Do the authors present any evidence to support this statement, or is it just a guess?

---

## Author Comment (AC3) · 14 Dec 2017

Anonymous Referee #1 Received and published: 13 July 2017

We would like to thank the reviewer again for the constructive and helpful comments on our manuscript.

All comments have been considered and a list of responses and changes in the manuscript is given below. Responses are written in bold face type and changes in the manuscript are written in *blue*.

Friedl et al. present a study on glacier retreat and changes in ice flow for Antarctic Peninsula outlet glaciers after disintegration of Wordie Ice Shelf. The work is based on analysis of remote sensing data from various sources. It extends the period of observations on frontal retreat, flow acceleration and glacier thinning that was reported by Rignot et al. (2005) and Wendt et al. (2010) up to the year 2009. Of particular interest is the production of a close time series of surface velocities, up to the year 2016, including the filling of gaps from previous years. The analysis of surface elevation change, comparing the change 2011 to 2014 to that of previous years, and the time series of frontal retreat are also very relevant for describing glacier behaviour. The review on previous work and the presentation and interpretation of the observations are presented in coherent manner at large. However, there are various individual points that are not well explained or questionable. The presented data sets are very useful for characterizing the glacier behaviour during the last two decades, but the discussion and conclusions focus on the description of the observed phenomena and do not provide any substantial new insights into the processes leading to the observed changes.

We have largely restructured and reformulated the Discussion and the Conclusion sections. In particular we have included a discussion of recently published data on atmospheric driven CDW upwelling events and elevation change in Wordie Bay. By linking these findings to our results on high resolution surface velocity change, ice thinning and grounding line retreat, we are able to provide new insights in the timing of the processes and reasons for the observed changes in Wordie Bay.

Specific Comments:

P1, L16: The conclusion on "pronounced basal melt at the grounding line" is not based on any direct observations for these glaciers.

We understand the reviewer's concerns. However, Depoorter et al. (2013) and Rignot et al. (2013) observed high basal melt rates for the remaining parts of the ice shelf, similar to those found for ice shelves in the Amundsen Sea sector (see answer P2, L13, L14). Furthermore the coincident timing of exceptional warm water intrusions in 2008/2009 and 2010/2011 into Marguerite Bay found by Walker and Gardner (2017) and the two acceleration phases found by us indicates that ocean warming and the observed changes in glacier dynamics are strongly correlated. Anomalously low sea ice extent in Marguerite Bay observed in 2008 and 2010 suggests that the warming events affected the upper water column, but Walker and Gardner (2017) found that the years 2009-2011 had the highest temperatures also at depths of 400 m. Hence, it is very likely that submarine ice melting was increased during phases of strong CDW upwelling events and that this has triggered unpinning and grounding line retreat. We rewrote the abstract in such a way that it points out that our conclusion on basal melt at the grounding line is not based on direct observations made in this study but on reasonable assumptions upon existing evidence. We also discuss this in more detail in the restructured Discussion and Conclusion parts.

**P1, L19: The length of the centreline (for which this values is valid) should be specified **We thank the reviewer for this suggestion. We have changed the sentence into:**

The resulting loss in buttressing is able to explain a remarkable median speedup of ~1.3 m d-1 (~30 %) between 2007 and 2011 observed along a centreline extending between the grounding line in 1996 and ~16 km upstream.

P1, L21: Fig. 5 shows in the downstream part of the profiles for 2011-2014 thinning rates that are smaller than for 2004 to 2008. 60% to 70% higher rates are only evident for a subsection of the profile shown in Fig. 5.

Despite of lower ice thinning rates measured towards the ice front in 2011-2014, our data show an overall increase of median ice thinning rates of  $\sim 1.1-1.3$  m a-1 or  $\sim 70$  % between the periods from 2004 to 2008 and from 2011 to 2014. Median elevation change rates were calculated for entire profiles, including subsections with lower and increased ice thinning rates. Hence the numbers refer to full profiles not just to the upstream parts. In some upstream parts ice thinning rates increased by even more than 100%. We have changed the corresponding section in chapter 5.2 in such a way that we hope it makes this now clearer to the reader:

Despite of lower ice thinning rates measured towards the ice front in 2011–2014, our data show an overall median increase of ice thinning rates along the profiles of ~1.1–1.3 m  $a^{-1}$  or ~70 % between the periods from 2004 to 2008 and from 2011 to 2014. However, in some areas 10–15 km upstream, ice thinning rates even doubled in the latter period.

P1, L26: Hardly possible that the glaciers draining into Wordie Bay have "a huge potential for an increase in sea level rise".

Our choice of words was probably too dramatic. We have changed the sentence into:

Hence, this endangers upstream ice masses, which can significantly increase the contribution of Fleming Glacier to sea level rise in the future.

**P2, L4: 4.21 Gt/a refers to Larsen-A and Prince-Gustav-Channel glaciers, 2011-2013. We apologize for this inaccuracy. We have changed the sentence into:**

Rott et al. (2014) estimated the total dynamic ice mass loss for the glaciers along the Nordenskjöld Coast and the Sjögren-Boydell glaciers after ice shelf disintegration to be  $4.21 \pm 0.37$  Gt a-1 between 2011–2013.

P2, L13, L14: Rignot et al. (2014), Suppl. Material, show for Wordie Ice Shelf clear dominance of calving losses compared to ablation, rather than "basal melt exceeding the ablation induced by calving".

We agree with the reviewer. Indeed Fig. 1 in Rignot et al. (2013) indicates that mass loss from Wordie Ice Shelf is much bigger from calving than from basal melt. However, mass losses of George VI, Wilkins, Bach and Stange ice shelves are shown to be clearly dominated by basal melt. Tab. 1 in Rignot et al. (2013) reveals that mass loss at Wordie Ice Shelf is calculated to be 7.6  $\pm$  3 Gt/a for calving and 6.5  $\pm$  3 Gt/a for basal melt, which means that ~54 % of mass loss can be attributed to calving and ~46 % to basal melt. However, the calculated basal melt rate of 23.6 ± 10 m/a is the second largest value obtained for Antarctica in this study. Interestingly a similar study of Antarctic wide basal melt rates by Depoorter et al. (2013) attributes 82 % of the mass loss of Wordie Ice Shelf to basal melt and only 18 % to calving flux (Fig. 1, Depoorter et al., 2013). The values correspond to a calving flux of  $2 \pm 3$  Gt/a and a mass loss of  $10 \pm 4$  Gt/a (or 14.79 ± 5.26 m/a respectively) due to basal melt (Table 1 in Suppl. Material, Depoorter et al., 2013). The high basal melt rates at Wordie Ice Shelf are similar to those found for ice shelves in the Amundsen Sea sector (Table 1 in Suppl. Material and Fig. 2, Depoorter et al., 2013). The differences between the two studies seem to be mainly caused by the different datasets used for the computations (e.g. Rignot et al., 2013 used ice thickness from Bedmap-2 whereas Depoorter et al., 2013 calculated ice thickness from satellite borne surface elevation data). Despite of the differences, both studies show that Wordie Ice Shelf undergoes pronounced basal melt. Consequently we have changed the section into:

For the south-western Antarctic Peninsula Rignot et al. (2013) demonstrated that basal melt of George VI, Wilkins, Bach and Stange Ice Shelves exceeded the ablation induced by calving. For Wordie Ice Shelf high basal melt rates of  $23.6 \pm 10 \text{ m a}^{-1}$  and  $14.79 \pm 5.26 \text{ m a}^{-1}$  have been reported by Rignot et al. (2013) and Depoorter et al. (2013) respectively. However, the presented melt ratios (i.e. the ratio between basal melt and the sum of calving flux and basal melt) differ between 46 % (Rignot et al., 2013) and 82 % (Depoorter et al., 2013).

P2, L15: Please explain the "small coastal atmospheric and oceanic processes".

Padman et al. (2012) found that changes in basal melt at Wilkins Ice Shelf are primarily determined by small-scale coastal atmospheric and oceanic processes that can alter the depth of the cold winter water layer, rather than by large-scale atmospheric forcing of benthic inflows of warm CDW along troughs cutting across the continental shelf. Such small-scale processes include complex interactions between coastal wind stress, surface radiation balance, sea ice and freshwater fluxes (including land runoff, sea-ice production and melt, and feedbacks between the ice shelf and regional hydrography and circulation). For example, increased freshwater fluxes from runoff and ice-shelf basal melt consolidated by downwelling forced by enhanced wind stress can lead to a depression of the cold winter water layer and hence to a reduction of basal melt. For the sake of clarity we have changed the corresponding sentence into:

Wilkins Ice Shelf experienced amplified basal thinning controlled by small-scale coastal atmospheric and oceanic processes that assist ventilation of the sub-ice-shelf cavity by upper-ocean water masses (e.g. variations in wind stress or reduced freshwater fluxes from runoff and ice-shelf basal melt) until ~8 years before break-up events took place in 2008 and 2009 (Braun and Humbert, 2009; Padman et al., 2012).

P4, L4: Date and source for the Bedmap2 DEM section over the study area should be mentioned (may have some impact for geocoding and analysis of surface elevation change).

The Bedmap2 DEM is a combination of several DEMs covering all or part of Antarctica (Fig. 5 and Tab. 3, Fretwell et al., 2013). The multiple surface elevation datasets were gridded together into a seamless DEM of Antarctica. On the Antarctic Peninsula north of 70° S the Bedmap2 DEM is entirely based on the improved ASTER GDEM by Cook et al. (2012), providing a vertical accuracy of  $\pm 26$  m. The ASTER GDEM is compiled from stacked photogrammetric DEMs from ASTER scenes acquired between 2000 and 2009 that are unspecified in the final product. On the Antarctic Peninsula south of 70° S the Bedmap2 DEM consists of data from the SPIRIT DEM, compiled from SPOT images, acquired in 2007-2008 (vertical accuracy is  $\pm 6m$ ) and from the NSIDC DEM (DiMarzio et al., 2007), derived from ICESat data acquired in 2003–2005 (vertical accuracy varies from  $\pm 0.4$  m to  $\pm 20$  m).

We used the Bedmap2 DEM as a source of topographic information for orthorectification of the velocity fields and for the derivation of local incidence angles required for the conversion from slant to ground range displacement. Here consistency of the elevation data is more important than the date. Since the extent of our Sentinel-1 scenes exceeded  $70^{\circ}$  S (i.e. the extent of the improved ASTER GDEM by Cook et al., 2012), the coverage of the Bedmap2 DEM allowed us to use a single consistent topographic reference for all velocity products. This is important when comparing velocities obtained from the different sensors.

We would like to clarify that we did not use the Bedmap2 DEM for the calculation of elevation changes between the Bedmap2 DEM and the TanDEM-X DEMs. Of course in this case the dates of the data would have been important. Instead, we used the differential phase between the simulated phase of the Bedmap2-DEM and the topographic phase of TanDEM-X to support phase unwrapping. However in the revised version of the manuscript we now use a subset of the TanDEM-X Global DEM for this procedure, as described in answer P5, L34.

Notwithstanding the above, we of course agree with the reviewer that providing more information on the Bedmap2-DEM is an asset. We have therefore changed the corresponding section into:

The Bedmap2 digital elevation model (DEM) of Antarctica (Fretwell et al., 2013), resampled to 100 m resolution, was taken as a topographic reference for orthorectification of the surface velocity fields and for the derivation of local incidence angles required for the conversion from slant to ground range displacement. Over the Antarctic Peninsula the Bedmap2 DEM provides a seamless

compilation of data from the improved ASTER GDEM (from ASTER stereo images acquired between 2000 and 2009) (Cook et al., 2012), the SPIRIT DEM (from SPOT stereo images acquired in 2007 and 2008) and the NSIDC DEM (from ICESat data acquired between 2003 and 2005) (DiMarzio et al., 2007).

P4, L24: May mention here that the OIB and Huss bedrock data are compared in Fig. S5. We have added a reference to Fig. S5 at the end of the corresponding sentence.

P5, L15: point clouds of differential elevation measurements are shown in Fig. 5. (not Fig. 4).

In Fig. 4 the locations of the measuring points are depicted which form the point clouds shown in Fig. 5. For more clarity we have changed the corresponding sentence into:

The locations of the resulting points of differential elevation measurements are shown in Fig. 4.

P5, L34: It is not meaningful using a low resolution DEM from a different epoch for performing DEM differencing with high resolution TanDEM-X data, if an up-to-date high resolution DEM from TanDEM-X is available.

As already mentioned in answer P4, L4 we originally subtracted a simulated topographic phase from the Bedmap2 DEM from the TanDEM-X phase prior to the unwrapping of the TanDEM-X phase and re-added the subtracted height-information afterwards. The fact that then only a differential phase has to be unwrapped, facilitates the phase unwrapping and makes it more robust. So the only purpose of subtracting the phase was facilitating the phase unwrapping rather than calculating elevation changes between Bedmap2 and TanDEM-X. Nevertheless, in the revised manuscript we now use a subset of the TanDEM-X global DEM with a spatial resolution of 12 m (Rizzoli et al., 2017) for the procedure. This is because we have also changed the vertical referencing in such a way that we now use the TanDEM-X global DEM to vertically adjust the TanDEM-X DEMs (see answer P6, L19 to L28).

We consequently have changed the corresponding sentence into:

A subset of the TanDEM-X global DEM, covering the two TSX/TDX-DEMs, was chosen to be the reference DEM.

We also have added a sentence to chapter 3 (Data):

A simulated phase from a subset of the TanDEM-X global DEM with a spatial resolution of 12 m (Rizzoli et al., 2017) was used to facilitate phase unwrapping during the generation of the two TSX/TDX DEMs. The TanDEM-X global DEM was also used as a reference for the absolute height adjustment of the TSX/TDX DEMs.

P6, L19 to L28: The differences between ATM and (uncorrected) TanDEM-X rates of elevation change (up to 2m/a) need further explanations and checks. 2m/a corresponds to 6 m difference for the 3 year time span. The area for the profile is located in the percolation zone. Typical values for TanDEM-X penetration bias in the percolation zone are about 4 m (see e.g. Wessel et al., ISPRS Annals. VL-III-7, doi:10.5194/isprsannals-III-7-9-2016). If the morphology of the snow and firm medium is the same on both dates, the penetration bias cancels out for DEM differencing. A difference in dh/dt of 6 m versus optical data can only be explained if the snow morphology is completely different on both dates (e.g. melting surface snow without penetration vs. completely frozen snow volume). This should show up clearly in the backscatter signatures.

We thank the reviewer for raising this important issue. We completely agree with the reviewer that a penetration bias of 6 m is pretty large for the percolation zone. We found that our approach of vertically referencing the TSX/TDX DEMs by calculating mean offsets over sea ice was not optimal. We therefore changed the vertical registration procedure in such a way that we now use a subset of the TanDEM-X global DEM at 12 m resolution as an absolute height reference. Before differencing, both TSX/TDX digital elevation models were vertically adjusted to the TanDEM-X global DEM according to their median offsets measured over stable ground. This improved the vertical registration and resulted in a more realistic penetration depth bias.

The maximum difference between ATM and uncorrected TDX/TSX elevation change rates was 1.25 m/a, corresponding to a 3.75 m difference for the 3 year time span. However, differences in elevation change were only measured in the lower areas of the glacier tongue, where surface melt occurs. In the upper areas (above 600 m altitude) the difference between ATM and

TSX/TDX elevation change rates was close to 0 m. Here the medium was likely completely frozen on both dates of acquisition, so that the penetration bias cancelled out. A comparison of backscatter values showed that in areas below 600 m altitude the backscatter of the 2014 acquisition was lower than the backscatter of the 2011 acquisition. In the upper areas above 600 m, however, the backscatter values were similar.

We have updated the TSX/TDX elevation change rates in the text and all figures that contain TSX/TDX elevation change data (i.e. Fig. 4, Fig. 5 b, Fig. S4 a-c and Fig. S6.1-4).

We have also added a Figure S4 d showing the differences in backscatter between both acquisition dates.

Furthermore we have updated Fig. S1 for the new areas over stable ground used for vertical adjustment and error estimation.

**Finally we have largely changed the corresponding section in the text:**

Before differencing, the TSX/TDX-DEMs must be vertically referenced. For this purpose the median vertical offset between the DEMs and the TanDEM-X global DEM was measured over stable areas (i.e. tops of nunataks and rock outcrops, which were not affected by image distortions) at altitudes between 150 m and 1000 m (Fig. S1), before both DEMs were adjusted accordingly.

After subtracting the vertically registered DEMs, the elevation differences were converted into yearly elevation change rates. We assessed the accuracy of the vertical registration over another set of stable areas at altitudes between 150 m and 1300 m (Fig. S1). The absolute median value of the extracted change rates was 0.37 m  $a^{-1}$  which primarily accounts for errors related to the vertical registration.

The comparison between elevation change rates obtained from the 2011–2014 OIB ATM flights and the 2011–2014 TSX/TDX data after the vertical registration of the DEMs showed a maximum overestimation of ice thinning of 1.25 m  $a^{-1}$  for the TSX/TDX measurements (Fig. S4 a, b). However, the general trend of the elevation change rates fits well to those calculated from the LiDAR data and significant differences in elevation change were only measured in the lower areas of the glacier tongue. In the upper areas (above 600 m altitude) the difference between ATM and TSX/TDX elevation change rates was close to 0 m. Here the snow volume was likely completely frozen on both dates of acquisition, so that the penetration bias cancelled out. A backscatter comparison showed lower values in 2014 than in 2011 in areas below 600 m altitude, whereas the backscatter in the upper areas above 600 m altitude was similar for both dates (Fig. S4 d).

**P7, L23: "determent" is probably the wrong word here.**

**The word has been deleted during reformulation of the section according to the suggestions of reviewer #2.**

P8, L9: It seems there was some slowdown after 1994, and gradual acceleration started in 2003. The selected graphic representation is not favourable for capturing such features.

We thank the reviewer for this helpful comment. We have added a curve showing the (smoothed) median velocities measured along the centreline profile to Fig. 2. The median absolution deviation (NMAD) of the median velocities between 1994 and 2007 is just 0.06 m d-1. Hence, velocities were pretty stable during this time. A comparison of velocities acquired in October 2007 and October 1995 shows almost no difference in median surface velocities. A clear sudden strong acceleration is first visible in 2008 and a strong gradual acceleration is apparent between 2010 and 2011. However, we have changed the corresponding section into:

Figure 2 shows that glacier velocities were rather stable between 1994 and 2007. The normalised median absolution deviation (NMAD) of the median velocities during this time was 0.06 m  $d^{-1}$ . A comparison of the velocities on 1995-10-27 and 2007-10-23 (Fig. 3a) along the centreline profile reveals that the median velocity difference between 1995 and 2007 was just 0.04 m  $d^{-1}$ .

P9, L7: "Figure 5 shows that prior to the speedup (2004–2008) Fleming Glacier was already affected by pronounced surface lowering." No elevation change data prior to 2004-2008 are shown in Fig. 5. We apologize for the misleading phrasing. In Fig. 5 elevation change rates are shown that were calculated from data acquired in 2004 and 2008. The time period 2004–2008 does not refer to the speedup but to elevation change. We have changed the section into:

Figure 5 shows that prior to the speedup in 2008, Fleming Glacier has already been affected by pronounced surface lowering. A clear trend of increasing ice thinning rates towards the glacier front is visible for 2004–2008.

P9, L28: "a vast part of the formerly grounded glacier tongue". "vast part" is a subjective impression; be specific.

**We thank the reviewer for this helpful advice. We have changed the corresponding section into:** *However, the same calculations for data acquired in 2011 and 2014 (Fig. 6, Track 3–5) as well as patterns of low and positive elevation change rates in the TSX/TDX 2011–2014 dh/dt map (Fig. 4, S7) show that after the final stage of glacier acceleration in 2011 an area of about 56 km2 (referring to the front in 2014) of the formerly grounded glacier tongue of the Airy Rotz Seller Fleming system had been floating*

P10, L14: "A possible location of the grounding line after the initial ungrounding in 2008. . ...". Confusing statement. Was the area floating before 2008 and then became grounded again? We apologize for the misleading phrasing. Until 2008 the frontal part of the glacier tongue was likely pinned to a subglacial ridge (probably a sill) located at the 1996 GL position. However, an ice cavity between the pinning point and a smaller hump  $\sim 2.5-3$  km upstream of the pinning point at the 1996 GL position and from then on the glacier was suddenly only grounded at the hump  $\sim 2.5-3$  km upstream. The grounding line was stabilized there until between 2010 and 2011 it further gradually retreated to its present position  $\sim 7-9$  km upstream. We have changed this section accordingly during restructuring of the Discussion part. We have further added a view graph to the supplement (Fig. S 8) showing our interpretation of the ungrounding process, in order to make it better understandable to the reader.

P10, L25: The Rothera station data (near Wordie I.S.) do not show a cooling trend for recent years. Oliva et al. (Science of the Total Environment 580, 210–223, 2017) report higher mean annual air temperature for 2006 to 2015 than in the previous decades.

We thank the reviewer for pointing us to the publication of Oliva et al. (2017). In contrast to other studies (e.g. Turner et al., 2016) which found an overall cooling trend for the entire Antarctic Peninsula since the late 1990s based on stacked temperature records, this study provides a regional analysis of temperature trends at different stations across the Antarctic Peninsula and the South Shetland Islands (SSI). Indeed Oliva et al. (2017) report higher mean annual air temperatures (MAAT) for the period 2006-2015 at Rothera and San Martin (which is closer to Wordie Ice Shelf than Rothera) than in previous decades. While the decadal mean temperatures at all stations on the SSI and the N-NE Antarctic Peninsula were 0.2-0.9 °C lower in the period 2006-2015 than in the period 1996-2005, at Rothera and San Martin decadal mean temperatures were 0.1 and 0.2 °C higher, respectively (Fig. 1 and Tab. 5, Oliva et al., 2017). However, when looking at decadal temperature trends, Oliva et al. (2017) reveal that although temperatures at Rothera and San Martin do not show a continuous cooling trend as recorded at other stations since the late 1990s, warming rates at both stations have decreased markedly since the decade 1996-2005 and a cooling trend was present between 2006 and 2015 (Tab. 5 and Figs. 2 and 4, Oliva et al., 2017). We consequently reworded the corresponding section into:

Furthermore, a trend of cooling air temperatures is reported for the Antarctic Peninsula since the end of the 1990s (e.g. Turner et al., 2016). Although decadal mean surface temperatures in the period 2006–2015 were 0.2 °C higher than in 1996–2005 at San Martin station (~120 km north of Wordie Ice Shelf), warming rates have decreased markedly since the decade 1996–2005 and show a cooling trend in 2006–2015 (Oliva et al., 2017). This may have reduced surface melt during recent years.

P10, L31, L32: Here it is stated that "flow acceleration usually affects both the floating and the grounded part of the glacier, but is largest near the grounding zone", and also "This is consistent with our observation of a highest relative speedup by \_32–35 % between 7 and 11 km upstream". If the highest speedup is 7 to 11 km upstream, the second statement is not consistent with the first one.

**The sentence has been removed during rewriting of the section.**

P11, L4: Any direct observations supporting the statement that basal melt is particularly effective in the grounding zone?

Same answer as for P1, L16

P11, L6: In which respect is the bedrock topography "unfavourable"?

The trough underneath Fleming Glacier has a retrograde slope. Such a bed topography is known to be an unstable configuration which fosters rapid grounding line retreat (e.g. Rignot et al., 2014). We have added a note on this during rewriting of the Discussion section.

P11, L7, L8: Fig. 5 shows thinning rates only for grounded ice. This does not provide any clear link to basal melt rates of the floating part in the grounding Zone

We agree. However, it has been shown by several other studies that Wordie Bay is subject to intruding warm CDW which causes substantial subglacial melt. Hence, it is very likely that basal melt also occurred prior to 2008 at the pinning point (at the 1996 grounding line). The sentence has been deleted during rewriting of the Discussion section and the central statement has been changed accordingly.

P12, L12, L13: "Our data suggest that enhanced basal melt at the grounding line due to increased shoaling of warm CDW most likely played a major role for the recent changes at Fleming Glacier." This conclusion is not based on the measurements presented in this study, but rather on measurements in other locations and reported in other publications.

We agree with the reviewer. We have changed this during restructuring of the Discussion section. Apart from that, same answer as for P1, L16

P12, L21: Please explain the expected "fatal effects on the stability and sea level contribution".

Modelled bed topography shows another deep subglacial trough with a retrograde sloped bed geometry 3-4 km upstream of the current grounding line. If the grounding line retreats to this trough, this may trigger a positive feedback loop of rapid grounding line retreat, flow acceleration, dynamic thinning, increased calving, and mass loss (marine ice sheet instability). However, we have reformulated and weakened the statement during restructuring of the Conclusion part

P19, L5: Fig. 1 caption. Please check the dates for Sentinel-1 data. One year time span for velocity retrieval?

We apologize for the mistake. The data used for calculating the velocity were acquired on 28-08-2015 and 09-09-2015. We have changed this.

P24, L5: Fig. 6 caption. "Fulfillment of floating condition". Does this refer to grounded or floating ice?

We apologize for the misleading phrasing. Orange dots show grounded ice along PIB flight lines as derived from hydrostatic equilibrium for the time before acceleration. We have changed the caption into:

*Orange dots: Grounded ice before acceleration as derived from PIB LiDAR and ice thickness data. Dates of PIB flights: a) 2002-11-26, b) 2004-11-18.*

P24, L7: "Fulfillment of the hydrostatic equilibrium condition". Same question as for P24, L5. We apologize for the misleading phrasing. Blue dots show freely floating ice and red dots show grounded ice along OIB flight lines as derived from hydrostatic equilibrium for the time after acceleration. We have changed the caption into:

Blue and red dots: Freely floating and grounded ice after acceleration as derived from OIB laser altimeter and ice thickness data. Dates of OIB flights: c) 2011-11-17, d) 2014-11-16, e) 2014-11-10.

L25, Table 1: SAR, column 6 shows single dates (not "Time Interval") Supplement

We thank the reviewer for pointing us to the misleading labelling of the column. Indeed the dates in the column refer to time intervals of SAR data used in the study. We have changed the labelling of the column to make this clearer to the reader.

Table S2: Pixel size should be accurately specified in along track (or LOS) and across track direction. These can be quite different even for a single sensor (e.g.Sentinel-1).

We agree with the reviewer that pixel sizes can largely differ between the across track and the along track direction. Hence, to make a conservative estimate of velocity errors, we always chose the coarser resolution value to be the image resolution  $\Delta x$  used in Formula 1 (S 2). In order to make this clearer to the reader we follow the recommendation of the reviewer by adding two columns to the table which show the resolution in ground range and azimuth. We also have changed the caption of Table S2 accordingly:

Ground range resolution  $\Delta x_{GR}$ , azimuth resolution  $\Delta x_{AZ}$  and image resolution  $\Delta x$  used in Formula 1 (S2) for calculating velocity errors. For most of the sensors  $\Delta x_{GR}$  is coarser than  $\Delta x_{AZ}$ , except for Sentinel-1a that has a coarser resolution in azimuth direction and TSX/TDX which have fairly equal resolutions in both directions. In order to make a conservative estimate of velocity errors, always the coarser resolution value was chosen to be  $\Delta x$

**References**

- Braun, M. and Humbert, A.: Recent Retreat of Wilkins Ice Shelf Reveals New Insights in Ice Shelf Breakup Mechanisms, IEEE Geosci. Remote Sensing Lett., 6, 263–267, doi: 10.1109/LGRS.2008.2011925, 2009.
- Cook, A. J., Murray, T., Luckman, A., Vaughan, D. G. and Barrand, N. E.: A new 100-m Digital Elevation Model of the Antarctic Peninsula derived from ASTER Global DEM: methods and accuracy assessment, Earth Syst. Sci. Data, 4, 129–142, doi: 10.5194/essd-4-129-2012, 2012.
- Depoorter, M. A., Bamber, J. L., Griggs, J. A., Lenaerts, J T M, Ligtenberg, S R M, van den Broeke, M R and Moholdt, G.: Calving fluxes and basal melt rates of Antarctic ice shelves, Nature, 502, 89–92, doi: 10.1038/nature12567, 2013.
- DiMarzio, J., Brenner, A., Schutz, R., Shuman, C. A. and Zwally, H. J.: GLAS/ICESat 500 m laser altimetry digital elevation model of Antarctica, Boulder, Colorado USA, 2007.
- Fretwell, P., Pritchard, H. D., Vaughan, D. G., Bamber, J. L., Barrand, N. E., Bell, R., Bianchi, C., Bingham, R. G., Blankenship, D. D., Casassa, G., Catania, G., Callens, D., Conway, H., Cook, A. J., Corr, H. F. J., Damaske, D., Damm, V., Ferraccioli, F., Forsberg, R., Fujita, S., Gim, Y., Gogineni, P., Griggs, J. A., Hindmarsh, R. C. A., Holmlund, P., Holt, J. W., Jacobel, R. W., Jenkins, A., Jokat, W., Jordan, T., King, E. C., Kohler, J., Krabill, W., Riger-Kusk, M., Langley, K. A., Leitchenkov, G., Leuschen, C., Luyendyk, B. P., Matsuoka, K., Mouginot, J., Nitsche, F. O., Nogi, Y., Nost, O. A., Popov, S. V., Rignot, E., Rippin, D. M., Rivera, A., Roberts, J., Ross, N., Siegert, M. J., Smith, A. M., Steinhage, D., Studinger, M., Sun, B., Tinto, B. K., Welch, B. C., Wilson, D., Young, D. A., Xiangbin, C. and Zirizzotti, A.: Bedmap2: improved ice bed, surface and thickness datasets for Antarctica, The Cryosphere, 7, 375–393, doi: 10.5194/tc-7-375-2013, 2013.
- Oliva, M., Navarro, F., Hrbáček, F., Hernández, A., Nývlt, D., Pereira, P., Ruiz-Fernández, J. and Trigo, R.: Recent regional climate cooling on the Antarctic Peninsula and associated impacts on the cryosphere, The Science of the total environment, 580, 210–223, doi: 10.1016/j.scitotenv.2016.12.030, 2017.
- Padman, L., Costa, D. P., Dinniman, M. S., Fricker, H. A., Goebel, M. E., Huckstadt, L. A., Humbert, A., Joughin, I., Lenaerts, Jan T. M., Ligtenberg, Stefan R. M., Scambos, T. and Van Den Broeke, Michiel R.: Oceanic controls on the mass balance of Wilkins Ice Shelf, Antarctica, J. Geophys. Res., 117, doi: 10.1029/2011JC007301, 2012.
- Rignot, E., Jacobs, S., Mouginot, J. and Scheuchl, B.: Ice-shelf melting around Antarctica, Science (New York, N.Y.), 341, 266–270, doi: 10.1126/science.1235798, 2013.

- Rignot, E., Mouginot, J., Morlighem, M., Seroussi, H. and Scheuchl, B.: Widespread, rapid grounding line retreat of Pine Island, Thwaites, Smith, and Kohler glaciers, West Antarctica, from 1992 to 2011, Geophys. Res. Lett., 41, 3502–3509, doi: 10.1002/2014GL060140, 2014.
- Rizzoli, P., Martone, M., Gonzalez, C., Wecklich, C., Borla Tridon, D., Bräutigam, B., Bachmann, M., Schulze, D., Fritz, T., Huber, M., Wessel, B., Krieger, G., Zink, M. and Moreira, A.: Generation and performance assessment of the global TanDEM-X digital elevation model, ISPRS Journal of Photogrammetry and Remote Sensing, 132, 119–139, doi: 10.1016/j.isprsjprs.2017.08.008, 2017.
- Rott, H., Floricioiu, D., Wuite, J., Scheiblauer, S., Nagler, T. and Kern, M.: Mass changes of outlet glaciers along the Nordensjköld Coast, northern Antarctic Peninsula, based on TanDEM-X satellite measurements, Geophys. Res. Lett., 41, 8123–8129, doi: 10.1002/2014GL061613, 2014.
- Turner, J., Lu, H., White, I., King, J. C., Phillips, T., Hosking, J. Scott, Bracegirdle, T. J., Marshall, G. J., Mulvaney, R. and Deb, P.: Absence of 21st century warming on Antarctic Peninsula consistent with natural variability, Nature, 535, 411–415, doi: 10.1038/nature18645, 2016.
- Walker, C. C. and Gardner, A. S.: Rapid drawdown of Antarctica's Wordie Ice Shelf glaciers in response to ENSO/Southern Annular Mode-driven warming in the Southern Ocean, Earth and Planetary Science Letters, 476, 100–110, doi: 10.1016/j.epsl.2017.08.005, 2017.

---

## Author Comment (AC4) · 14 Dec 2017

**We would like to thank the reviewer for constructive and helpful comments on our manuscript. All comments have been considered and a list of responses and changes in the manuscript is given below. Responses are written in bold face type and changes in the manuscript are written in** *blue***.**

Friedl et al. present a dense time series of surface velocities and elevation change rates of the Fleming Glacier system in the past two decades from multi-source remote sensing data. This study identified two significant acceleration phases occurred in 2008 and 2010-2011 after the Wordie Ice Shelf has nearly disappeared in front the Fleming Glacier. The determination of floating area based on the hydrostatic height anomalies, bedrock data, elevation change rates, and acceleration phase is very helpful to explain the rapid dynamic changes after 2008. However, some points in this study are not well explained and need further rewording. Some minor wording and grammar errors also need to be fixed.

General Comments:

1) It is not clear about the epoch of the 2008 grounding line position. Please clarify it, particularly regarding locations in Fig. 6 and Fig. 6S.
**We have added more information on how we decided the groundling line locations in 2008 to the methods section (Sect. 4.3) and especially to the discussions on Fig. 6S. We are aware that the exact 2008 grounding line positions are vague and we have tried to make this clearer in the text. However, since the glacier front had retreated behind the grounding line 1996 in 2008 for the first time, the 2008 grounding line must have been situated upstream of the 1996 position.**

2) The method used to determine the grounding line position in 2008, 2011 and 2015 from the velocity data, surface-thinning rate, and the bedrock data is not clear enough in the method (Sect. 4.3, P7, L19-27) and result section (Sect. 5.3), especially regarding the determination of the grounding line position in 2008. From Fig. 6, it is appeared that the 2015 grounding line is mainly based on the ridges of modelled bedrock combined with the hydrostatic height anomalies along OIB flight lines. However, profiles in Fig. S5 show that how much the Bedmap2 bedrock and Huss and Farinotti's modelled bedrock can differ from the PIB and OIB measurements. Then how reliable are the hills used to decide the grounding line location? In the analysis of Fig. 6S, it is better to explain how the different evidence (velocity profiles, elevation change profiles, hydraulic height anomalies, and modelled bedrock) is being combined to determine the estimated 2008 and 2015 grounding line. If they are not consistent with each other, please explain the final selection.
**We have largely rewritten Sect. 4.3 Sect. 5.3 and the discussions to Fig. S6 a-d in order to better explain how we finally decided the grounding line positions. We have also addressed possible uncertainties in the bedrock data.**

3) This study (P11, L27-29) concludes that the basal melting driven by ocean-warming is the dominant trigger of the grounding line retreat and glacier acceleration. However this lacks direct evidence, namely the basal or frontal melting rates underneath the ice shelf. Given the little ice shelf left in 2008, it is not very convincing to say that the basal melting underneath the ice shelf dominates the ungrounding process.

**We understand the reviewer`s concerns. However, Depoorter et al. (2013) and Rignot et al. (2013) calculated high basal melt rates of 14.79 ± 5.26 m/a and 23.6 ± 10 m/a respectively for the remaining parts of the ice shelf. These melt rates are similar to those found for ice shelves in the Amundsen Sea sector (Table 1 in Suppl. Material and Fig. 2, Depoorter et al., 2013). ). Hence, it is likely that basal melt also occurs at the grounding line. Furthermore the coincident timing of exceptional warm water intrusions in 2008/2009 and 2010/2011 into Marguerite Bay found by Walker and Gardner (2017) and the two acceleration phases found by us, indicates that ocean warming and the observed changes in glacier dynamics are strongly correlated. Anomalously low sea ice extent in Marguerite Bay, observed in 2008 and 2010, indicates that the warming events affected the upper water column, but Walker and Gardner (2017) found that the years 2009-2011 had the highest temperatures also at depths of > 400 m. Hence, it is very likely that submarine ice melting at the grounding line was increased during phases of strong CDW upwelling events and that this has triggered unpinning in 2008 and gradual grounding line retreat between 2010 and 2011. We rewrote the Discussion, Conclusion and Abstract sections in such a way that it points out that our conclusion on basal melt at the grounding line is not based on direct observations made in this study but on reasonable assumptions upon existing evidence.**

Specific Comments:

P1, L8: change to ". . . regions most affected by climate change"
**The sentence has been changed accordingly.**

P1, L14: The finding about glacier ungrounded in 2008 between January and March has not been confirmed by observation or measurements. Please reedit this sentence.
**The glacier front had retreated behind the 1996 grounding line location for the first time in 2008. Hence, the grounding line must have been located upstream of its 1996 position by this time. Before 2008 no marked changes in surface velocity have been observed, which would have indicated grounding line retreat. Although the precise position of the grounding line after the acceleration in 2008 remains vague, at least it is clear that in 2008 the grounding line was located upstream of its 1996 position. It is likely that confirmed intrusions of warm CDW to Maguerite Bay in 2008 have triggered unpinning of the glacier tongue from a pinning point located at the 1996 grounding line position. However, the sentence has been reformulated during rewriting of the abstract.**

P1, L24: "Currently, the tongue of Fleming Glacier is grounded in a zone of bedrock elevation of -400 m" This statement is not mentioned in the main text. Please state this in the main text and make the elevation color bar in Fig. 6 clearer for reading the elevation values. It will be helpful to add elevation contours.
**We thank the reviewer for this helpful suggestion. We have added elevation contours and a more detailed color bar to Fig. 6. We have also added a note on the elevation level of the recent grounding line (~-400 to -500 m) to the main text (section 5.3):**
*For most regions of the glacier tongue we estimate the current grounding line to coincide with these ridges at an elevation between ~-400 and -500 m.*

P1, L26: You can't say "this endangers" "a huge potential". Add a suitable verb before "a huge potential". "huge" is not appropriate here.
**We have changed the sentence into:**
*Hence, this endangers upstream ice masses, which can significantly increase the contribution of Fleming Glacier to sea level rise in the future.*

P2, L4: Add "be" before the number "4.21".
**We have changed the sentence accordingly:**

P3, L3: "former tributary glaciers" is confusing since the glaciers still exist, Suggest instead ". . . originally fed by several major tributary glaciers (Fig.1). Among these, Fleming Glacier is . . .".
**We have changed this accordingly.**

P3, L10: "Here the ice shelf was temporarily grounded and stabilized . . ." is confusing. Does "Here" mean at a time point or at the location of the pinning points? Is there evidence of grounding of the previously floating Wordie Ice Shelf? Do you mean it was stabilized temporarily? Do you mean "the next rapid break up event" occurred in 1989? If yes, please specify it.
**We apologize for the ambiguous phrasing. "Here" refers to the location of the pinning points (e.g. ice rises/rumples). Based on the analysis of satellite imagery e.g. Vaughan (1993) concluded that the ice front retreat of Wordie Ice Shelf was punctuated by periods of stasis during which the ice front rested on theses pinning points. When embedded in the ice shelf they helped to stabilize the ice shelf by compressing it upstream and providing restraint at the grounding line. However, during the following rapid break up events (not just the 1989 event) the ice rises appeared to behave as indenting wedges, contributing to weakening the ice shelf and accelerating break up. We have reformulated the section into:**
*Analyses of satellite imagery suggest that the ice shelf was temporarily grounded and stabilized at these pining points until one of the next rapid break-up events took place (Doake and Vaughan, 1991; Reynolds, 1988; Vaughan, 1993; Vaughan and Doake, 1996). However, during phases of ice front retreat, instead of protecting the ice shelf against decay, ice rises that were embedded in the ice shelf appeared to behave as indenting wedges, contributing to weakening the ice shelf and accelerating break up (Doake and Vaughan, 1991; Vaughan, 1993).*

P3, L18-19: Please add the ice front position in 1997 and 2000 in Fig. 1. It will help reader to understand your statement here.
**We have changed Fig. 1 accordingly.**

P4, L21-25: The information about the modelled bedrock topography from Huss and Farinotti 2014 is not enough. Given its role, indication of origins and uncertainties should be included in the paper.
**We agree with the reviewer. We have added more detailed information on the dataset to the corresponding section:**
*The dataset was generated by subtracting modelled ice thickness from the improved ASTER GDEM by Cook et al. (2012). Ice thickness was derived by constraining a simple model based on the shallow ice approximation for ice dynamics with observational data of ice thickness (OIB) and surface velocity (Rignot et al., 2011). Where available the modelled ice thickness was corrected with OIB ice thickness data, leading to more precise ice thickness values in such areas. On average, the local uncertainty in ice thickness of the dataset is ±95m but values can reach ±500m for deep troughs without nearby OIB measurements. Since OIB coverage is fairly good across Fleming Glacier, uncertainties in modelled ice thickness are relatively low in this area. However, a comparison of bedrock elevations from Huss and Farinotti with bottom elevations calculated from OIB ATM and CoRDS measurements shows that although the modelled bedrock reflects the general subglacial topography well, the absolute difference in bottom elevation can be even more than 100 m (Fig. S5). One possible reason for this is a difference between ATM heights and the refined ASTER GDEM, which transfers to bedrock elevation.*

P7, L18: Modify "Sect. S2" to "Sect. S3".
**We have changed this.**

P7, L22: "mitigates" is hardly correct for 90
**We have substituted "mitigates" by "decreases".**

P7, L22-23: It is not clear how you decide the grounding line position from the surface velocities. Please clarify. If you define the grounding line based on where the maximum velocity increase occurred, please specify it with relevant references.

**We have added some more sentences with references on how and why information on surface velocity can be used to identify the grounding line:**

*Moreover, the grounding line marks the transition between two fundamentally different flow regimes of grounded and freely floating ice. Whereas the flow dynamics of grounded ice are dominated by vertical shear and controlled by basal drag, flow of floating ice is drag free and dominated by longitudinal stretching and lateral shear (Schoof, 2007). The difference in flow dynamics of grounded and floating ice can result in pronounced changes in surface velocity close to the grounding line (Stearns, 2007; Stearns, 2011). Moreover, Rignot et al. (2002) demonstrated that if ungrounding occurs, the resulting flow acceleration usually affects both the floating and the grounded part of the glacier, but is largest near the grounding zone. Thus, velocity profiles can serve as additional information for locating the grounding line (Stearns, 2007; Stearns, 2011).*

P7, L23: replace "determent" - do you mean "determinant"?
**The word has been deleted during rewriting of the section.**

P8, L18: Please comment on the higher velocity changes in the 2011 glacier front in Fig. 3 and why it is ignored.
**We thank the reviewer for this helpful question. In 2011 the glacier front had an all-time minimum position. Consequently in 2007 and 2015 the ice front was located seaward of the 2011 front. Ice velocities are highest at the glacier front. Hence, at the location of the 2011 front inherently high front velocities were subtracted from inherently lower velocities of floating ice (2011-2015) or vice versa (2007-2011). This results in peaks of velocity change at the 2011 front position. Thus the strong velocity changes at the 2011 front do not represent "real" dynamic change and were ignored. We have added a note on this in the text. Furthermore, we have modified Fig. 3 a, b in such a way that we now do not show velocity change for 2007-2011 and 2011-2015 seaward of the 2011 front anymore. Otherwise here velocity change would result from comparing velocity of the glacier tongue with velocities of sea ice or ice melange.**

*Fig. 3a, b show absolute and relative velocity changes along the centreline profile for the periods 1995–2007, 2007–2011 and 2011–2015. In 2011 the location of the glacier front reached its most inland position. Consequently velocities measured on sea ice or ice mélange in 2011 (seaward of the 2011 front) were excluded from our analyses. Large changes in surface velocity close to the 2011 front in the periods 2007–2011 and 2011–2015 do not represent real dynamic change, but result from comparing the inherently higher frontal velocities in 2011 with lower velocities of the floating glacier tongue in 2007 and 2015.*

P8, L23-24: Suggest updating Table2 by adding the velocity in 2015 from Walker and Gardner, 2017 at three sites.
**We have updated the table and have added a note on the higher velocities in 2015 at the three sites to the text:**

*However, surface velocities derived from Landsat 8 feature tracking suggest that in 2015 velocities at the three measuring sites had increased by ~20% in comparison to 2008 (Walker and Gardner, 2017; Zhao et al., 2017) (Tab. 2).*

P8, L27: As I could tell, the highest ice thinning rates for Fleming Glacier from Fig. 4 (orange color) is higher than 6 m a$^{-1}$.
**The reviewer is right. We apologize for the imprecise formulation. We have changed the sentence into:**

*On Fleming Glacier the highest ice thinning rates with peak values of more than ~6 m a$^{-1}$ were recorded in a zone extending from ~8 to ~14 km upstream.*

P9, L2: "For the location of the data see Fig. 4" ! "The location of the data is shown in Fig. 4".
**We have changed this.**

P9, L13: Missing space after the full stop.
**We have added a space.**

P9, L19-20: It would be useful to compare your thinning rate with that from Zhao et al., 2017 and Walker and Gardner, 2017.
**We have added a comparison of our results and the thinning rates from Walker and Gardner, 2017 to the Discussion section: However, we do not compare ice thinning rates with those of Zhao et al., 2017, since the time intervals they used for computing elevation change differ from ours.**

P10, L1-3: The "We extracted" part of the statement should explicitly refer to Fig. 6S, and the location of the four profiles should refer to Fig. 6.
**We have changed the section into:**
*We extracted data of surface velocities, TDX/TSX 2011–2014 elevation change rates, bedrock topography along four profiles on Airy and Fleming Glacier in order to estimate the recent and the grounding line location in 2008 after the first acceleration phase (Fig. S6 a-d). The location of the profiles as well as the deduced grounding line positions are shown in Fig. 6.*

P10, L3-4: For "The plots", you mean Fig. 6 or Fig. S6? It would be clearer to say "Those profile plots (Fig S6 1-4) suggest . . .". Which glacier or profile are you talking about for the 2008 grounding line position?
**We mean Fig. S6 1-4. We have changed this in the text accordingly.**

P10, L5-6: "possible" and "likely" are same to me. It's not clear how the grounding line positions in 2008 are decided on Profile 2, 3, and 4.
**The sentence has been removed during rewriting of the results section (5.3). We have changed the term "possible" to "tentative" making clear that the 2008 grounding line position is only a rough estimate. We have added more information on how the 2008 grounding line positions were deduced from the profiles to Sect. 4.3, 5.3 and Fig. S6 a-d.**

P10, L7-15: Is detailed discussion of a supplementary figure in the main text proper? Consider including Fig. S6-3.
**We agree with the reviewer. We have removed the section during rewriting of the results section (5.3). We hence have kept the Figure S6 c in the supplemental material.**

P10, L18-20: About "no significant acceleration since 1996 . . .", if you compare the velocity at three sites with the 2015 velocity from Walker and Gardner, 2017, you will find some speed up from 1996 to 2015. Alternatively you need to qualify with "by 2013".
**We have changed the section into:**
*The remarkable median speedup of ~1.3 m d$^{-1}$ which we recorded between 2007 and 2011 is in good agreement with an acceleration of ~ 400–500 m a$^{-1}$ reported by Walker and Gardner (2017) and Zhao et al. (2017) for the period 2008–2015. However, a comparison of our velocities in 2013 with their velocities in 2015 at the three measuring sites of Doake (1975) ~50 km upstream suggests that the recent speedup had not propagated up to these locations prior to 2015.*

P10, L27-28: Besides the surface melt, basal melting water generated from the basal frication heating could be another trigger of enhanced basal sliding. It's hard to rule out the possibility of enhanced basal sliding unless you have further evidence.
**We agree that liquid water generated from fricational heat can enhance basal sliding. Basal frication heat can increase if the glacier accelerates or if it gains mass. Hence, increased fricational heat is more likely a result of the acceleration, rather than the trigger of it. However, we do not want to rule out that in areas where the glacier is currently grounded, as a consequence of initial acceleration, basal sliding is enhanced due to melt water from increased fricational heat.**
**We have changed the section into:**

*All in all, enhanced basal sliding due to percolating meltwater is likely not the explanation for the observed increase in flow velocities. However, we do not rule out that as a consequence of the acceleration, basal sliding had increased in grounded areas by liquid water generated from greater basal fricational heat.*

P10, L30-31: If this is your method to decide the grounding line position from the velocity data, please move this sentence to Sect. 4.3.
**We have moved and reformulated the sentence (see answer P7, L22-23)**

P11, L12: Please cite references for the Twaites Glacier and the Pine Island Bay.
**We have added a reference to Rignot et al. (2014).**

P24, L4: Modify "Linens" to "Lines".
**We have changed this.**

Supplement Material

It is better to use consistent zero distance mark for the various profiles in both the main text (Fig. 2, 3, 5, 6) and the supplementary material (Fig. S6). The use of different starting points (the 1996 grounding line position or the 2007 ice front position) for describing the distance is confusing. If it is too difficult to shift the origin in the supplementary figures, the GL96 grounding line needs to be clearly marked.
**We thank the reviewer for the helpful suggestion. We have changed the profile extent to the 1996 grounding line in Fig. S6.**

P12: Modify the legend of a) and b) from "Ice Surface/Bottom elevation (OIB)" to "Ice Surface/Bottom elevation (PIB)".
**We have changed the legend accordingly.**

P15, L3: Modify "profiles 1-5" to "profiles 1-4". Also consider labeling figure parts as S6.a-d for consistency with other sections. Anyway add labels to the profile figures.
**We have changed the labelling to S6.a-d and have modified the references in the text accordingly.**

P15, L4: Can't find OIB ice surface/bottom in those figures.
**We have deleted the corresponding sentence.**

P15, L5: Modify "grounding line" to "GL".
**We have changed this accordingly.**

P15, L6-7: Can't find Green, Red and Blue lines in those figures.
**The sentence has been deleted.**

P16, L7-8: It's hard to tell the 2008 grounding line position upstream the 2011 front without direct evidence. Please explain it.
**We agree with the reviewer that it is difficult to tell the exact 2008 grounding line position from the modelled bedrock data only. The only thing which is clear is that the grounding line must have been upstream of its 1996 position in 2008, since in 2008 the glacier front had retreated behind the 1996 grounding line location. The position which we state is only a best guess based on the data we have in hand. We hope that we have made this clearer in the rewritten text to Fig. S6 b and in the results section. We have also pointed to the limitations of the bedrock data in the text and have labelled the 2008 position in Fig. S6 b with a question mark.**

P16, L10-11: Please add GL11 in Fig. S6.2.
**We have added the GL11 to Fig. S6 b.**

P17, L3: Move "in 2008" after "the grounding line".
**We have changed this accordingly.**

P17, L5-6: Along Profile2, it's hard to tell this hill in Fig. 6. Considering the uncertainty of bedrock ïïj´Lyellow and brown line in Fig. S6.3), is it possible that this smaller hill is not physically real? On P18, L5-6 you suggest that "the topography may be distorted in the modelled bedrock data".
**Same answer as for P16, L7-8**

**References**

Cook, A. J., Murray, T., Luckman, A., Vaughan, D. G. and Barrand, N. E.: A new 100-m Digital Elevation Model of the Antarctic Peninsula derived from ASTER Global DEM: methods and accuracy assessment, Earth Syst. Sci. Data, 4, 129–142, doi: 10.5194/essd-4-129-2012.

Depoorter, M. A., Bamber, J. L., Griggs, J. A., Lenaerts, J T M, Ligtenberg, S R M, van den Broeke, M R and Moholdt, G.: Calving fluxes and basal melt rates of Antarctic ice shelves, Nature, 502, 89–92, doi: 10.1038/nature12567.

Doake, C. S. M. and Vaughan, D. G.: Rapid disintegration of the Wordie Ice Shelf in response to atmospheric warming, Nature, 350, 328–330, doi: 10.1038/350328a0.

Reynolds, J. M.: The structure of Wordie ice shelf, Antarctic peninsula, Bulletin-British Antarctic Survey, 57–64.

Rignot, E., Jacobs, S., Mouginot, J. and Scheuchl, B.: Ice-shelf melting around Antarctica, Science (New York, N.Y.), 341, 266–270, doi: 10.1126/science.1235798.

Rignot, E., Mouginot, J., Morlighem, M., Seroussi, H. and Scheuchl, B.: Widespread, rapid grounding line retreat of Pine Island, Thwaites, Smith, and Kohler glaciers, West Antarctica, from 1992 to 2011, Geophys. Res. Lett., 41, 3502–3509, doi: 10.1002/2014GL060140.

Rignot, E., Mouginot, J. and Scheuchl, B.: Ice flow of the Antarctic ice sheet, Science (New York, N.Y.), 333, 1427–1430, doi: 10.1126/science.1208336.

Rignot, E., Vaughan, D. G., Schmeltz, M., Dupont, T. and MacAyeal, D.: Acceleration of Pine Island and Thwaites Glaciers, West Antarctica, Annals of Glaciology, 34, 189–194, doi: 10.3189/172756402781817950.

Schoof, C.: Ice sheet grounding line dynamics. Steady states, stability, and hysteresis, J. Geophys. Res., 112, 1720, doi: 10.1029/2006JF000664.

Stearns, L. A.: Outlet glacier dynamics in East Greenland and East Antarctica, Ph.D. thesis, Univ. of Maine, Orono, 2007.

Stearns, L. A.: Dynamics and mass balance of four large East Antarctic outlet glaciers, Ann. Glaciol., 52, 116–126, doi: 10.3189/172756411799096187.

Vaughan, D. G.: Implications of the break-up of Wordie Ice Shelf, Antarctica for sea level, Antarctic Science, 5, doi: 10.1017/S0954102093000537.

Vaughan, D. G. and Doake, C. S. M.: Recent atmospheric warming and retreat of ice shelves on the Antarctic Peninsula, Nature, 379, 328–331, doi: 10.1038/379328a0.

Walker, C. C. and Gardner, A. S.: Rapid drawdown of Antarctica's Wordie Ice Shelf glaciers in response to ENSO/Southern Annular Mode-driven warming in the Southern Ocean, Earth and Planetary Science Letters, 476, 100–110, doi: 10.1016/j.epsl.2017.08.005.

Zhao, C., King, M. A., Watson, C. S., Barletta, V. R., Bordoni, A., Dell, M. and Whitehouse, P. L.: Rapid ice unloading in the Fleming Glacier region, southern Antarctic Peninsula, and its effect on bedrock uplift rates, Earth and Planetary Science Letters, 473, 164–176, doi: 10.1016/j.epsl.2017.06.002.

---

## Author Comment (AC5) · 14 Dec 2017

**We would like to thank the reviewer his helpful comments on our manuscript.**
**All comments have been considered and a list of responses and changes in the manuscript is given below. Responses are written in bold face type and changes in the manuscript are written in** *blue***.**

The topic of this paper, Fleming Glacier, is extremely interesting given the significant changes reported by previous publications, so I was very keen to read these new results. The authors present new ice velocity, elevation change and grounding line position data acquired from a range of airborne and satellite based instruments. The ice velocity results are nice, and I believe constitute the most complete time series of velocity measurements over Fleming glacier which provides new insight into the timing of ice speedup on this sector.

However, after reading the manuscript there are a number of major flaws with the methods employed to derive surface elevation change and to measure the grounding line retreat. As it stands, the problem with the techniques make it highly likely that both the magnitude of the elevation change signal, and the grounding line retreat, may not be correctly reported in this paper. For example, cross calibrating the DEM elevations over sea ice, which varies annually and can range in thickness from 0-5m, is extremely unsatisfactory. Moreover, even if this correction is accepted, the authors estimate that known X-band penetration bias can account for ~50% of the dh/dt signal across the basin. Even if the estimate of elevation change is accepted, the associated error measurements do not reflect spatial variability in the data quality, and are unrealistically small given the spread of the raw data. For example, it is stated that the CAMS-ATM dh/dt data has an error of 0.2m/yr in regions where the point measurements at the same location range from -0.5 to -7 m/yr. These errors must therefore be revised. Regarding the estimate of grounding line retreat, the technique is unproven and is un-validated in this paper. Even if the authors demonstrate that the technique can be trusted, as it stands the results presented in this paper are contradictory because regions measured to be grounded are located within the new floating ice shelf area.

On top of these technical issues, I have a few more minor concerns about the use of scientific terminology throughout the paper, and the overly simplistic nature of the analysis and discussion sections. The specifics of these and other concerns are documented in detail below. My criticisms of the methods and results presented in the paper are major, and will be time consuming and require a significant effort to properly address. The implications of these concerns is that I believe the magnitude and spatial pattern of the elevation change and grounding line retreat data may be incorrectly reported in this paper. This is significant as I do not have confidence in two of the three core datasets presented, therefore, it is my recommendation that this paper is not suitable for publication in its current form.

**Specific Edits**
P1 L21 – Edit 'far upstream of the glacier', using this as a location is ambiguous, better use a fixed reference location, like the calving front in 'x' year.
**The sentence has been deleted during rewriting of the abstract.**

P1 L26 – Edit paper to quantify 'much larger ice masses'.

**It is difficult to exactly predict how much ice will get lost in the future due to the response of the glacier to further grounding line retreat from observational data. However, if the grounding line retreats up to the deep retrograde trough ~3-4 km upstream of its current position, it is likely that the glacier gets into an unstable configuration where rapid grounding line retreat and more mass loss than today can be expected (see answer P11 L11). The quantification of the expected mass loss, however, needs the involvement of ice sheet modelling, which is out of the scope of this paper. Nevertheless, we have changed the sentence into:**

*Hence, this endangers upstream ice masses, which can significantly increase the contribution of Fleming Glacier to sea level rise in the future.*

P2 L4 – Edit missing to 'be'.
**We have changed this accordingly.**

P2 L7 – Edit ice 'shelf' tributaries.
**We have changed this accordingly.**

P2 L30 – Edit 'explain' to 'investigate'. I'd argue at this point the authors haven't demonstrated they can explain the observed signal.
**We have changed this accordingly.**

P3 L23 – Poor wording, edit out 'got' both times.
**We have changed the sentence into:**
*During the following years (2010–2015) the fronts of the glaciers in Wordie Bay remained quite stable, except at the Prospect system where the once interconnected floating ice tongues of the three glaciers disconnected and some floating ice was lost.*

P4 L2 – Edit 'dynamic' and check use throughout paper. Previous publications (e.g. Rignot et al 2005) have demonstrated that Fleming exhibited dynamic imbalance during their study period, but as the authors state this paper spans a longer time period, and at this point it hasn't been proven that Fleming is dynamically imbalanced for the full duration of their study period.
**We agree with the reviewer, that at this point it is not clear, whether the dynamic changes we observe are consequences of the disintegration of the ice shelf or a result of other processes. However, at least we investigate the dynamic changes of the glacier in the time period after the disintegration of the ice shelf. We hence have changed the sentence into:**
*We used a broad remote sensing data set in order to investigate the changes in ice dynamics at Fleming Glacier between 1994 and 2016 after the disintegration of Wordie Ice Shelf.*
**We have also checked the use of the word "dynamic" throughout the paper and have changed it where necessary.**

P5 L6 – More informative to state what % of each velocity image is removed during the filtering process as this should be an indicator of the quality of the tracking output. The filtering should remove 100% of the unreasonable results, otherwise its not a very good filter!
**We apologize for this inaccuracy. The filter removes more than 99 % of erroneous vectors and removes very few false negatives (Burgess et al., 2012). We have changed this in the text accordingly. The filter has been successfully applied in several other peer reviewed studies (e.g. Rankl et al., 2014; Rankl et al., 2016; Seehaus et al., 2015; Seehaus et al., 2016). However, the proportion of velocity vectors removed from each velocity image by the filter is not only depended on the quality of the tracking but also on the coverage of the scene. For example, more erroneous tracking results are likely to be found in areas where no marked features can be used for tracking (e.g. in the interior of the peninsula). If a scene covers more of such areas, the amount of removed velocity vectors will be higher than in other scenes which only cover the glacier tongue. Hence we have calculated the proportion of removed velocity vectors for an area over Fleming Glacier, which is covered by all scenes and which is relevant for our velocity measurements. The area for which the calculations have been done has been added to Fig. S1.**

**Additionally Tab. S3 has been updated with a column containing the proportion of removed measurements.**

P5 L16 – 0.2 m is the accuracy of the original point measurements; the authors are using the dataset after re-gridding it so state the accuracy of this dataset instead or as well as the accuracy of the raw data. The accuracy is also different for different sensors, so the authors should provide statistics for each dataset.

**We thank the reviewer for this important comment. We have now cited additional references to state the accuracies of the different sensors used in this study:**

*We derived ice thinning rates on Fleming Glacier for 2004–2008 and 2011–2014 by comparing ellipsoid heights of the PIB (ATM, 2004), the Centro de Estudios Científicos Airborne Mapping System (CAMS, 2008) and the OIB (ATM, 2011, 2014) airborne LiDAR datasets. The vertical accuracy of the ATM elevation data is estimated to be better than 0.1 m (Krabill et al., 2002; Martin et al., 2012). For the CAMS data, vertical accuracy is 0.2 m (Wendt et al. 2010).*

P6 L2/3 – This is correction extremely unsatisfactory. Sea ice is a complex parameter, and is certainly not a stable/constant reference surface for precise cross calibration of elevation measurements. In the Antarctic, sea ice can range from 0 to 5 m thickness with very large spatial and temporal variability, snow depth on sea ice is not routinely measured but can account for half the thickness retrieval, and ocean height varies with tides, atmospheric pressure etc. When deriving the correction, the authors have not attempted to account for interannual variability in sea ice thickness so this must be addressed before any confidence can be had in the elevation change measurements. This is critical because the range of thickness variability is the same order of magnitude as the dh/dt signal calculated from the DEM differencing. The authors must revise the manuscript to characterise the temporal variability of the sea ice over which they are cross calibrating the DEM's, and to rule out any influence from this factor on the end elevation change. If this effect can be proven to be negligible, the authors should also state the size of the correction, and which DEM was adjusted, in the manuscript. Having said all this, I suggest the authors dont cross calibrate the DEM's over sea ice at all as it hugely reduces the confidence that I believe we can have in these results.

**We thank the reviewer for raising this important issue. According to the previous suggestions of reviewer #1 we have changed the vertical registration procedure in such a way that we now use a subset of the TanDEM-X global DEM at 12 m resolution as an absolute height reference. Before differencing, both TSX/TDX digital elevation models were vertically adjusted to the TanDEM-X global DEM according to their median offsets measured over stable ground. This improved the vertical registration and resulted in a more realistic penetration depth bias.**

**The maximum difference between ATM and uncorrected TDX/TSX elevation change rates was 1.25 m/a, corresponding to an absolute 3.75 m difference for the 3 year time span. However, differences in elevation change were only measured in the lower areas of the glacier tongue, where surface melt can occur. In the upper areas (above 600 m altitude) the difference between ATM and TSX/TDX elevation change rates was close to 0 m. Here the medium was likely completely frozen on both dates of TDX acquisition, so that the penetration bias cancelled out. A comparison of backscatter values showed that in areas below 600 m altitude the backscatter of the 2014 acquisition was lower than the backscatter of the 2011 acquisition. In the upper areas above 600 m, however, the backscatter values were similar.**

**We have updated the TSX/TDX elevation change rates in the text and all figures that contain TSX/TDX elevation change data (i.e. Fig. 4, Fig. 5 b, Fig. S4 a-c and Fig. S6.1-4).**

**We have also added a Figure S4 d) showing the differences in backscatter between both acquisition dates.**

**Furthermore we have updated Fig. S1 with the new areas over stable ground used for vertical adjustment and error estimation.**

**Finally we have largely changed the corresponding section in the text:**

*Before differencing, the TSX/TDX-DEMs must be vertically referenced. For this purpose the median vertical offset between the DEMs and the TanDEM-X global DEM was measured over stable areas (i.e. tops of nunataks and rock outcrops, which were not affected by image distortions) at altitudes between 150 m and 1000 m (Fig. S1), before both DEMs were adjusted accordingly.*

*After subtracting the vertically registered DEMs, the elevation differences were converted into yearly elevation change rates. We assessed the accuracy of the vertical registration over another set of stable areas at altitudes between 150 m and 1300 m (Fig. S1). The absolute median value of the extracted change rates was 0.37 m $a^{-1}$ which primarily accounts for errors related to the vertical registration.*

*...*

*The comparison between elevation change rates obtained from the 2011–2014 OIB ATM flights and the 2011–2014 TSX/TDX data after the vertical registration of the DEMs showed a maximum overestimation of ice thinning of 1.25 m $a^{-1}$ for the TSX/TDX measurements (Fig. S4 a, b). However, the general trend of the elevation change rates fits well to those calculated from the LiDAR data and significant differences in elevation change were only measured in the lower areas of the glacier tongue. In the upper areas (above 600 m altitude) the difference between ATM and TSX/TDX elevation change rates was close to 0 m. Here the snow volume was likely completely frozen on both dates of acquisition, so that the penetration bias cancelled out. A backscatter comparison showed lower values in 2014 than in 2011 in areas below 600 m altitude, whereas the backscatter in the upper areas above 600 m altitude was similar for both dates (Fig. S4 d).*

P6 L20 – The authors calculation that up to 2 m/a of the thinning rate can be attributed to TSX DEM penetration bias. This is a huge error which accounts for ~50% of the dhdt signal present across the majority of the basin. The ICESat and ATM tracks that this error was calculated from have extremely limited coverage, and don't pass through the region with the highest thinning rates, therefore its possible that this number might even be an underestimate. For example, other studies have shown that the penetration bias in DEM's derived from TSX/TDM data over snow covered terrain can be as large as 4m (e.g. Dehecq et al 2016), which is the same magnitude as the dh/dt signal presented in figure 4. The large size of the known errors relative to the size of the signal, combined with the limited data that has been used to characterise the error makes it very difficult to have confidence in the thinning rates presented here. Other auxiliary datasets such as atmospheric temperature data, or SAR backscatter images might also be used to characterise the onset and spatial pattern of melt in the study area. The authors description of how they have accounted for this source of error is cursory given its size relative to the dh/dt signal in the study area. As surface melt and therefore penetration is known to have large spatial variability, I recommend that the authors revise their approach to account for this spatial variation across the basin, as a polynomial fit derived from a single track of airborne data will definitely not capture the magnitude or pattern of this effect across the study area.

**Same answer as for P6 L2/3. Furthermore we understand the reviewer`s concers on our approach of applying a penetration depth correction derived from a single track of altimeter data to the entire study area. However, we have to deal with the data we have in hand. The tongue of the Airy-Rotz-Seller-Fleming system (which is our main study area) has only a width of ~20 km. We therefore think that it is acceptable to assume that melt conditions and the penetration depth bias is similar for this area and just altitude depending.**

P6 L27 – Assuming an error of 0.2m/a just because its one order of magnitude higher than the direct inter-comparison with the ATM data the dh/dt was calibrated against isn't satisfactory. Errors are spatially variable, so the authors should revise their approach.

**Due to the relatively small size of the study area the spatial variability of penetration bias in our study area should be little too (see previous answer). Hence, we think that assuming an error of 0.2 m/a due to penetration bias variability is already a conservative estimate.**

P6 L33 – Why use a 35 m buffer if the ICESat footprint is known to be 70 m? I recommend the authors use the same footprint size as the aim is to do the most direct comparison possible.

**We apologize for the misleading phrasing. We used a buffer with a radius of 35 m, which is equal to the 70 m diameter of the ICESat footprint. We have changed the sentence into:**

*To take into account the 70 m footprint of the GLAS instrument, we applied a buffer with a radius of 35 m and calculated the median from the extracted values at each point.*

P7 L15 – Cite a reference for the source of the firn density correction variable.
**We have added a reference to Griggs and Bamber (2011).**

P7 L25 – As far as I'm aware, this method of detecting grounding line position has not been proven in peer reviewed literature. Although the logic behind it is reasonable, (i.e. if the ice is in hydrostatic equilibrium it must be floating), factors such as the spatial resolution and error on each input dataset will severely limit the sensitivity of the technique for detecting grounding line position, let alone change in grounding line position. To be convinced that the technique works, I recommend grounding line retreat from this method is evaluated against known retreat rates, in the Amundsen sea for example. If suitable data isn't available to validate this technique in another area, then alternatively a proven technique can be employed to evaluate the hydrostatic technique in this study area. For example, ERS-2 SAR data with a 3-day temporal baseline was acquired in this area in 2011, so if coherent, this should be used to produce a grounding line estimate from the proven quadruple difference interferometry technique (Rignot et al, 1998). At a minimum the authors must state the error on their estimate of grounding line position from hydrostatic height anomaly, and it follows that if the uncertainty on the measurement is greater than the change in position assumed, then the method is not viable.

**We understand the reviewer`s concerns. Of course DInSAR would have been the preferred method for grounding line detection. We tried to form interferograms from ERS-1 (1994) and ERS-2 (2011) data with a temporal baseline of 3 days as well as from Seninel-1 data (2016) with a temporal baseline of 6 days. However, due to the high speed of the glacier none of the datasets maintained enough coherence over the relevant areas. Besides strong temporal decorrelation, the ERS-2 data of 2011 suffers from large and unstable Doppler centroid frequencies due to the failure of the gyroscope in 2001, which additionally hampers coherence (Miranda et al., 2003). Hence, we moved to the alternative approach of deriving the recent grounding line positions by calculating hydrostatic equilibrium from ice elevation and ice thickness data. This is a frequently used method in peer reviewed literature (e.g. Enderlin and Howat, 2013; Fricker, 2002; Münchow et al., 2014; Tinto and Bell, 2011). Typically the point of first stable hydrostatic equilibrium is located a few hundred meters seaward of the interferometric grounding line (Rignot et al., 2011). An exception are a few ice streams which have an ice plain (i.e. a region upstream of the grounding line with low surface slopes that is only lightly grounded and where the ice is very close to hydrostatic equilibrium). Here, hydrostatic analysis may place the grounding line some kilometers upstream of the real grounding line (Corr, H. F. J. et al., 2001; Crabtree and Doake, 1982). However, in our datasets we do not see any signs for the presence of an ice plain on Fleming Glacier.**

**In order to demonstrate the applicability of the hydrostatic height anomaly method, we have conducted the same calculations as on Fleming Glacier for the glaciers in the Cabinet Inlet region (Larsen C Ice Shelf) (Fig. 1, 2). Here coherence is also maintained in Sentinel-1 6-day temporal baseline interferograms and no long term grounding line migration has been observed since the last ERS measurement in 1996. This enabled us to compare our hydrostatic calculations with interferometric grounding lines from different epochs. Fig. 1 shows that the hydrostatic calculations are generally in good agreement with the DInSAR measurements. The differences between the 1996 and the 2016 DinSAR grounding line are mainly due to short term tidal grounding line migration. The mean difference between the first hydrostatic equilibrium and the closest DInSAR-grounding line is ~ 300 m.**

**We have also added a sentence on the estimated uncertainty of grounding line positions derived from hydrostatic height anomalies to S3:**

*For all possible sensor-depending errors of $H_i$=750 m the Monte Carlo simulation yielded a standard deviation of < 12 m for $\Delta e$. Thus we assumed the total uncertainty of $\Delta e$ to be ~12 m. Consequently, we assigned locations on our OIB and PIB profiles to be freely floating ice, if the calculated values of $\Delta e$ lay within this range. In the vicinity of the grounding zone the TDX global DEM has a minimum gradient of 1°, and therefore we estimate the uncertainty in the horizontal position of the transition from grounded to freely floating ice to be ~700 m.*

[Figure]

**Figure 1: (a)** Location of the study area at the Antarctic Peninsula. Map base: SCAR Antarctic Digital Database, version 6.0. (**b**) Comparison of floating and grounded ice from hydrostatic height anomalies with grounding lines derived from DInSAR. Brown line: grounding line in 1996 from ERS-1/2 (Rignot et al., 2011). Green line: grounding line in 2016 from Sentinel-1 a/b acquisitions on 2016-09-27, 2016-10-03 and 2016-10-09. Blue and red dots: Freely floating and grounded ice after acceleration as derived from OIB laser altimeter and ice thickness data on 2011-11-14. Background: USGS Landsat Image Mosaic of Antarctica (LIMA).

[Figure]

**Figure 2:** Fulfillment of the hydrostatic equilibrium assumption from hydrostatic height anomaly calculations along the OIB profile from 2011-11-14. The profile extends from southwest to northeast on the map (Fig. 1). Purple dots: PIB/OIB ice surface/bottom elevations. Ice surface elevation is taken from PIB/OIB ATM measurements and ice bottom elevation is calculated by subtracting OIB/PIB ice thickness from ice surface elevation. Yellow line: Bedrock elevation from Bedmap 2 (Fretwell et al., 2013). Red and blue dots: calculated ice surface elevation in hydrostatic equilibrium $e_{he}$ and information on hydrostatic equilibrium (blue: freely floating ice, red: grounded ice)

P8 L3 – The Figure 2 z-scope is not easy to interpret. I suggest the authors re-plot this information as a standard x/y line plot of the time series of flow line ice speeds, with an inset showing change in calving front position.
**We understand the reviewer`s concerns. However, in our view a standard x/y line plot which contains all of our 175 measurements would be very confusing. Furthermore the huge amount of lines would make a distinguishable color coding impossible. Thus we think that our plot is a clear way to show a chronologic time series of all our measurements. The plot also reveals the timing and the propagation of pronounced acceleration very well. However, for additional information we have added a graph of smoothed median center line velocities to the plot.**

P8 L14 – Poor sentence wording. Edit.
**We have changed the sentence into:**
*Since 2008 the glacier tongue never has exceeded the 1996 grounding line anymore*

P8 L28 – 8 to 14 km upstream of which location. Edit sentence to be more precise.
**All distances in the text are relative to the 1996 grounding line. We have stated this at the beginning of section 5: "Distances in the subsequent text are given in reference to the grounding line of 1996." We hope that the reviewer is fine if we do not change the text here.**

P8 L30 – Edit text to quantify 'lower parts'.
**We have changed the section into:**
*A tendency to lower negative or even positive elevation change rates could be observed on the lower parts of the joint Fleming and Seller glacier tongue between 0 and up to ~9 km upstream. However the pattern was not as clear as on Airy Glacier, where a distinct area of low ice thinning rates could be detected between 0 and ~4 km upstream.*

P8 L31 – Edit manuscript to remove all 'could be detected' wording. You are stating what results you have observed, so it 'has been detected', not the less affirmative 'could be'.
**We thank the reviewer for pointing this out. We have changed it throughout the text.**

P9 L4 – The scatter on figure 5a is very large and must be addressed given that it is significantly greater than the previously stated errors. I recommend that a) distance markers are annotated onto the ATM and ICESat track locations on Figure 4 so we can see how this corresponds to

the x axis distance scale in Figure 5. My interpretation of figure 5a is that the elevation change measurements are unusable between 0 and 20 km of the grounding line, which looks like its about up to the 'g' on the Fleming annotation on figure 4. This is the key area of interest, so vastly limits the usefulness of these datasets. B) state the method used to calculate the lines of best fit, e.g. moving average, polynomial fit? How has the clearly erroneous data been removed? C)

**The large scatter of the ATM and the CAMS data between 0 and 20 km of the grounding line is due to the undulated and heavily crevassed surface, where a purely horizontal displacement of crevasses can cause apparent positive and negative elevation differences. Many crevasses are up to 50 m wide and 300 m long. Due to the high spatial resolution and precision of the airborne laser altimeter data, these characteristics of the surface are well reflected in the signal. If looking at the ICESat elevation change of 2004–2008 , which by the way shows a very similar trend like the 2004–2008 ATM-CAMS data, indeed the scatter is smaller, but this is just because of the much lower spatial resolution of the data. Hence, more scatter does not necessarily mean that the data is more "erroneous". In general, the ATM and CAMS data have been shown to be very useful to calculate elevation change over Fleming Glacier (Gardner et al., 2017; Wendt et al., 2010; Zhao et al., 2017).**

**To better account for the large scatter, we now follow the previously published approach by Wendt et al. (2010) by applying a median filter to the data, before calculating the cubic regression functions. However, this did not change the results substantially. We have added a note on this to the text:**

*The large scattering of the data is due to the highly crevassed surface of the glacier tongue, where a purely horizontal displacement of crevasses can cause apparent positive and negative elevation differences. Therefore, a median filter was applied to the data before adjustment of a cubic function.*

**We have also updated Figure 5a and the numbers accordingly. Furthermore, we have followed the suggestion of the reviewer by adding distance markers to Figure 4.**

P9 L10 – Based on figure 5a, stating that the CAMS-ATM show elevation change of 4.1 ± 0.2 m/a is not credible. The raw data shown in figure 5a shows that at this location the elevation change ranges between -0.5 to 7 m/yr, so the error of 0.2m/yr is effectively meaningless. Please revise the error estimate here, and throughout the rest of the results paragraph.

**In order to minimize the influence of scatter, we now apply a median filter before calculating the cubic regression functions of elevation change. Furthermore, for all median ice thinning rates we now calculate the normalised median absolution deviation (NMAD) as a measure of the statistical dispersion ("error") of the input data. We have added a note on this to the results paragraph (5.2):**

*For all median elevation change rates presented below, we calculated the NMAD in order to account for the statistical dispersion of the input data.*

P9 L12 – The fact that Fleming is thinning between 04-08, doesn't prove that the catchment hasn't reached an equilibrium since shelf collapse in 1989. The two effects may be entirely uncorrelated, so although its possible, without a continuous dataset I don't think it can be proven one way or the other. I recommend the authors revise this wording. Changing 'shows' to 'might suggest' would be more factually correct.

**We agree. We have changed the sentence according to the reviewer`s suggestion.**

P9 L22 - Although I don't like the method, it's clear how the hydrostatic equilibrium has been calculated along the airborne tracks. Can the authors clarify what method they have used to draw the grounding line connecting the dots in Figure 6? For example, according to their own data, a section of grounded ice on track 'c)' is included in the now 'floating' area. I recommend the authors revise the line as their data shows it isn't correct.

**The reviewer is right. Originally we assumed that since the data used for calculating the hydrostatic equilibrium on track c was acquired in 2011, the area now may be floating. However, since there is no further evidence for this assumption, we have revised the line according to the reviewer`s suggestion and have updated Fig. 6 and Fig. S7 accordingly.**

**Following the suggestions of reviewer #2, we have also added some more information to the methods section on how the dots have been connected:**

*Wherever possible, we gave preference to information on hydrostatic equilibrium for the final decision of the recent grounding line location. For selected profiles across the glacier, recent and previous (2008) grounding line positions were estimated by combining evidence from elevation change, bedrock topography and surface velocity. In the remaining areas, the recent grounding line was deduced from combining information on elevation change rate patterns in the TDX/TSX 2011–2014 dh/dt map with information on bedrock topography.*

P9 L30 – Based on the above comment, the number stated for the area of the floating shelf will also need to recalculated.
**We have changed the number for the area of floating ice throughout the text to ~56 km $^{2}$.**

P9 L33 – Quantify 'several km'.
**We have changed the sentence into:**
*The ridges reach up to ~ 9 km upstream of the 1996 grounding line.*

P10 L25 – Although Turner et al 2016 shows that the long term air temperatures are decreasing the situation may be more complicated than that, and Sundal et al (2011) showed that a simple linear relationship between melt water vs lubrication is not currect, as melt induced speed-up can be offset by drainage efficiencies. I'd revise this text to avoid oversimplifying these relationships.
**We agree with the reviewer that in Greenland, where strong surface melt in summer leads to enhanced basal sliding due to rapid migration of meltwater to the ice-bedrock interface, the acceleration effect may be offset by efficient drainage. However, the meltwater production on Fleming Glacier is generally not large enough to percolate to the bed (Rignot et al., 2005). No seasonal variation in surface velocities, which would indicate enhanced glacial sliding during summer, is observed. Furthermore, surface melt is likely further reduced due to decreasing air temperatures. Hence, since there is no enhanced basal sliding due to percolating surface meltwater on Fleming Glacier, the mechanisms as described by Sundal et al. (2011) play no role. Nevertheless, we have added a sentence on basal sliding and variations in drainage efficiency:**
*In Greenland seasonal velocity fluctuations have been linked to both enhanced basal sliding due to the penetration of surface melt water to the ice-bedrock interface and inter-annual differences in drainage efficiency (Moon et al., 2014; Sundal et al., 2011; Zwally et al., 2002).*

P11 L4 – Remove sentence about basal melt. This hasn't been measured in this study so is just a generic assumption, and no reference provided to previous study evidencing statement.
**We have largely rewritten the discussion, conclusion and abstract sections. In the revised manuscript we link our observations to high basal melt rates measured by Depoorter et al. (2013) and Rignot et al. (2013) and to exceptional phases of warm CDW intrusions in 2008/2009 and 2010/2011 found by Walker and Gardner (2017). Anyway, the sentence has been deleted during reformulation of this section.**

P11 L8 – Same statement as above re basal melt inference.
**Same answer as for P11 L4**

P11 L11 – Really poor sentence wording. Edit to be more diligent with regards to terminology. Stability is a specific process, i.e. 'unstoppable' retreat that will continue to propagate even if environmental conditions returned to their original state. Glacier imbalance and grounding line retreat can occur stably. I haven't seen evidence presented in this paper of about the likely future instability of Fleming, so tighten up language.
**We apologize for the poor wording. However, it is well known that if grounding line retreat occurs on ice resting on a retrograde bed, this can trigger a positive feedback loop of flow acceleration, dynamic thinning, increased calving, mass loss and further grounding line retreat**

**("marine ice sheet instability" or "tidewater instability") (e.g. DeConto and Pollard, 2016; Favier et al., 2014; Rignot et al., 2014; Schoof, 2012). The subglacial topography of Fleming Glacier reveals a trough 3-4 km upstream of the current grounding line, which has such a retrograde bed. This suggests that once the grounding line has exceeded the edge of the trough, instability is possible. Especially since further oceanic forcing is likely to continue in the future (see answer P12L18), this is a likely future scenario for Fleming Glacier. We have changed the sentence into:**

*Hence, if grounding line retreat exceeds the edge of this trough, destabilisation like on Thwaites Glacier and in the Pine Island Bay region (Favier et al., 2014; Rignot et al., 2014) is possible, which would involve further rapid grounding line retreat and amplified mass loss in the future.*

P11 L12 – edit Thwaites
**We have changed this accordingly.**

P11 L18 – Again I feel the analysis here is overly simplistic. Wouters et al were the first to present the rapid thinning rates and mass loss from the Western Palmer Land region, but subsequent publications (Hogg et al 2017) have shown that only ~30% of this should be attributed to ocean induced dynamic imbalance. Revise text to reflect known complexity. Discussion general – the authors have stated results from other regions of WAIS/AP, however this really isn't tied very coherently into how this impacts on the results they have presented on Fleming.
**We understand the concerns of the reviewer. We have added a sentence on the publication of Hogg et al. (2017):**
*For the glaciers on Western Palmer Land Hogg et al. (2017) showed that ~ 35% of the ice loss after 2009 can be attributed to dynamic thinning triggered by ocean driven melt.*
**Additionally, we have substantially rewritten the discussion section, following the suggestions of reviewer #1 and #2. The timing of acceleration found by us is coincident with the timing of recently published exceptional warm CDW intrusions in 2008/2009 and 2010/2011 to Wordie Bay due to phases of strong westerly winds during strong La Niña/+SAM (positive Southern Annular Mode) events. Hence, we could substantiate the link between ocean forcing and our observed dynamic changes.**

P12 L10 – as previously stated I do not think the authors have proven dynamic instability on Fleming. Imbalance maybe, but instability, no. Equally, attributing the signal to ocean induced dynamic forcing without properly evaluating any oceanographic or atmospheric data is poor. These interlinked processes are very complex, and really hard to disentangle. Although its entirely plausible that ocean forcing is responsible, I do not think the analysis presented in this paper has proven it.
**See answer P11L18**

P12 L18 – 'ocean forcing is likely to continue'. Do the authors present any evidence to support this statement, or is it just a guess?
**We thank the reviewer for this important remark. Oceanic forcing is likely to continue, since the SAM is expected to be shifted further poleward, which would foster conditions like in 2008/2009 and 2010/2011. We have changed the sentence into:**
*Pronounced oceanic forcing is likely to continue, since the SAM is forecasted to be shifted further poleward, which will foster conditions like those during the strong La Niña/+SAM events in 2008/2009 and 2010/2011 (Abram et al., 2014; Fogt et al., 2011; Walker and Gardner, 2017). Thus, further retreat of the grounding line and more dynamic thinning are expected on Fleming Glacier.*

**References**

Abram, N. J., Mulvaney, R., Vimeux, F., Phipps, S. J., Turner, J. and England, M. H.: Evolution of the Southern Annular Mode during the past millennium, Nature Climate change, 4, 564–569, doi: 10.1038/nclimate2235, 2014.

Burgess, E. W., Forster, R. R., Larsen, C. F. and Braun, M.: Surge dynamics on Bering Glacier, Alaska, in 2008–2011, The Cryosphere, 6, 1251–1262, doi: 10.5194/tc-6-1251-2012, 2012.

Corr, H. F. J., Doake, C. S. M., Jenkins, A. and Vaughan, D. G.: Investigations of an "ice plain" in the mouth of Pine Island Glacier, Antarctica, J. Glaciol., 47, 51–57, doi: 10.3189/172756501781832395, 2001.

Crabtree, R. D. and Doake, C.S.M.: Pine Island Glacier and Its Drainage Basin. Results From Radio Echo-Sounding, A. Glaciology., 3, 65–70, doi: 10.3189/S0260305500002548, 1982.

DeConto, R. M. and Pollard, D.: Contribution of Antarctica to past and future sea-level rise, Nature, 531, 591–597, doi: 10.1038/nature17145, 2016.

Depoorter, M. A., Bamber, J. L., Griggs, J. A., Lenaerts, J T M, Ligtenberg, S R M, van den Broeke, M R and Moholdt, G.: Calving fluxes and basal melt rates of Antarctic ice shelves, Nature, 502, 89–92, doi: 10.1038/nature12567, 2013.

Enderlin, E. M. and Howat, I. M.: Submarine melt rate estimates for floating termini of Greenland outlet glaciers (2000–2010), J. Glaciol., 59, 67–75, available at: https://www.cambridge.org/core/services/aop-cambridge-core/content/view/8BA26CC3DEBE21F24ABF703A41A97C3A/S0022143000203419a.pdf/div-class-title-submarine-melt-rate-estimates-for-floating-termini-of-greenland-outlet-glaciers-2000-2010-div.pdf, 2013.

Favier, L., Durand, G., Cornford, S. L., Gudmundsson, G. H., Gagliardini, O., Gillet-Chaulet, F., Zwinger, T., Payne, A. J. and Le Brocq, A. M.: Retreat of Pine Island Glacier controlled by marine ice-sheet instability, Nature Climate change, 4, 117–121, doi: 10.1038/nclimate2094, 2014.

Fogt, R. L., Bromwich, D. H. and Hines, K. M.: Understanding the SAM influence on the South Pacific ENSO teleconnection, Clim Dyn, 36, 1555–1576, 2011.

Fretwell, P., Pritchard, H. D., Vaughan, D. G., Bamber, J. L., Barrand, N. E., Bell, R., Bianchi, C., Bingham, R. G., Blankenship, D. D., Casassa, G., Catania, G., Callens, D., Conway, H., Cook, A. J., Corr, H. F. J., Damaske, D., Damm, V., Ferraccioli, F., Forsberg, R., Fujita, S., Gim, Y., Gogineni, P., Griggs, J. A., Hindmarsh, R. C. A., Holmlund, P., Holt, J. W., Jacobel, R. W., Jenkins, A., Jokat, W., Jordan, T., King, E. C., Kohler, J., Krabill, W., Riger-Kusk, M., Langley, K. A., Leitchenkov, G., Leuschen, C., Luyendyk, B. P., Matsuoka, K., Mouginot, J., Nitsche, F. O., Nogi, Y., Nost, O. A., Popov, S. V., Rignot, E., Rippin, D. M., Rivera, A., Roberts, J., Ross, N., Siegert, M. J., Smith, A. M., Steinhage, D., Studinger, M., Sun, B., Tinto, B. K., Welch, B. C., Wilson, D., Young, D. A., Xiangbin, C. and Zirizzotti, A.: Bedmap2: improved ice bed, surface and thickness datasets for Antarctica, The Cryosphere, 7, 375–393, doi: 10.5194/tc-7-375-2013, 2013.

Fricker, H. Amanda: Redefinition of the Amery Ice Shelf, East Antarctica, grounding zone, J. Geophys. Res., 107, doi: 10.1029/2001JB000383, 2002.

Gardner, A. S., Moholdt, G., Scambos, T., Fahnestock, M. A., Ligtenberg, S., van den Broeke, Michiel and Nilsson, J.: Increased West Antarctic ice discharge and East Antarctic stability over the last seven years, The Cryosphere Discuss., 1–39, doi: 10.5194/tc-2017-75, 2017.

Griggs, J. A. and Bamber, J. L.: Antarctic ice-shelf thickness from satellite radar altimetry, Journal of Glaciology, 57, 485–498, doi: 10.3189/002214311796905659, 2011.

Hogg, A. E., Shepherd, A., Cornford, S. L., Briggs, K. H., Gourmelen, N., Graham, J. A., Joughin, I., Mouginot, J., Nagler, T., Payne, A. J., Rignot, E. and Wuite, J.: Increased ice flow in Western Palmer Land linked to ocean melting, Geophys. Res. Lett., 353, 283, doi: 10.1002/2016GL072110, 2017.

Krabill, W. B., Abdalati, W., Frederick, E. B., Manizade, S. S., Martin, C. F., Sonntag, J. G., Swift, R. N., Thomas, R. H. and Yungel, J. G.: Aircraft laser altimetry measurement of elevation changes of the greenland ice sheet. Technique and accuracy assessment, Journal of Geodynamics, 34, 357–376, doi: 10.1016/S0264-3707(02)00040-6, 2002.

Martin, C. F., Krabill, W. B., Manizade, S. S., Russell, R. L., Sonntag, J. G., Swift, R. N. and Yungel, J. K.: Airborne topographic mapper calibration procedures and accuracy assessment, Hanover, Tech. Rep. NASA/TM-2012-215891, 32 pp., 2012.

Miranda, N., Rosich, B., Santella, C. and Grion, M.: Review of the impact of ERS-2 piloting modes on the SAR Doppler stability, Proceedings Fringe '03, 1–5, 2003.

Moon, T., Joughin, I., Smith, B., Broeke, M. R., Berg, W. Jan, Noël, B. and Usher, M.: Distinct patterns of seasonal Greenland glacier velocity, Geophysical Research Letters, 41, 7209–7216, available at: http://onlinelibrary.wiley.com/doi/10.1002/2014GL061836/full, 2014.

Münchow, A., Padman, L. and Fricker, H. A.: Interannual changes of the floating ice shelf of Petermann Gletscher, North Greenland, from 2000 to 2012, J. Glaciol., 60, 489–499, available at: https://www.cambridge.org/core/services/aop-cambridge-core/content/view/5DA6EADBD7E2C73FC0B79505689D4D3D/S0022143000205960a.pdf/interannual_changes_of_the_floating_ice_shelf_of_petermann_gletscher_north_greenland_from_2000_to_2012.pdf, 2014.

Rankl, M., Fürst, J. Jakob, Humbert, A. and Braun, M. Holger: Dynamic changes on Wilkins Ice Shelf during the 2006-2009 retreat derived from satellite observations, The Cryosphere Discuss., 1–17, doi: 10.5194/tc-2016-218, 2016.

Rankl, M., Kienholz, C. and Braun, M.: Glacier changes in the Karakoram region mapped by multimission satellite imagery, The Cryosphere, 8, 977–989, doi: 10.5194/tc-8-977-2014, 2014.

Rignot, E., Casassa, G., Gogineni, P., Kanagaratnam, P., Krabill, W., Pritchard, H. D., Rivera, A., Thomas, R., Turner, J. and Vaughan, D. G.: Recent ice loss from the Fleming and other glaciers, Wordie Bay, West Antarctic Peninsula, Geophys. Res. Lett., 32, doi: 10.1029/2004GL021947, 2005.

Rignot, E., Jacobs, S., Mouginot, J. and Scheuchl, B.: Ice-shelf melting around Antarctica, Science (New York, N.Y.), 341, 266–270, doi: 10.1126/science.1235798, 2013.

Rignot, E., Mouginot, J., Morlighem, M., Seroussi, H. and Scheuchl, B.: Widespread, rapid grounding line retreat of Pine Island, Thwaites, Smith, and Kohler glaciers, West Antarctica, from 1992 to 2011, Geophys. Res. Lett., 41, 3502–3509, doi: 10.1002/2014GL060140, 2014.

Rignot, E., Mouginot, J. and Scheuchl, B.: Antarctic grounding line mapping from differential satellite radar interferometry, Geophys. Res. Lett., 38, n/a, doi: 10.1029/2011GL047109, 2011.

Schoof, C.: Marine ice sheet stability, J. Fluid Mech., 698, 62–72, doi: 10.1017/jfm.2012.43, 2012.

Seehaus, T., Marinsek, S., Helm, V., Skvarca, P. and Braun, M.: Changes in ice dynamics, elevation and mass discharge of Dinsmoor–Bombardier–Edgeworth glacier system, Antarctic Peninsula, Earth and Planetary Science Letters, 427, 125–135, doi: 10.1016/j.epsl.2015.06.047, 2015.

Seehaus, T., Marinsek, S., Skvarca, P., van Wessem, J. M., Reijmer, C. H., Seco, J. L. and Braun, M. H.: Dynamic Response of Sjögren Inlet Glaciers, Antarctic Peninsula, to Ice Shelf Breakup Derived from Multi-Mission Remote Sensing Time Series, Front. Earth Sci., 4, F01005, doi: 10.3389/feart.2016.00066, 2016.

Sundal, A. Venke, Shepherd, A., NIENOW, P., Hanna, E., Palmer, S. and Huybrechts, P.: Melt-induced speed-up of Greenland ice sheet offset by efficient subglacial drainage, Nature, 469, 521–524, doi: 10.1038/nature09740, 2011.

Tinto, K. J. and Bell, R. E.: Progressive unpinning of Thwaites Glacier from newly identified offshore ridge. Constraints from aerogravity, Geophys. Res. Lett., 38, n/a-n/a, doi: 10.1029/2011GL049026, 2011.

Walker, C. C. and Gardner, A. S.: Rapid drawdown of Antarctica's Wordie Ice Shelf glaciers in response to ENSO/Southern Annular Mode-driven warming in the Southern Ocean, Earth and Planetary Science Letters, 476, 100–110, doi: 10.1016/j.epsl.2017.08.005, 2017.

Wendt, J., Rivera, A., Wendt, A., Bown, F., Zamora, R., Casassa, G. and Bravo, C.: Recent ice-surface-elevation changes of Fleming Glacier in response to the removal of the Wordie Ice Shelf, Antarctic Peninsula, Annals of Glaciology, 51, 97–102, doi: 10.3189/172756410791392727, 2010.

Zhao, C., King, M. A., Watson, C. S., Barletta, V. R., Bordoni, A., Dell, M. and Whitehouse, P. L.: Rapid ice unloading in the Fleming Glacier region, southern Antarctic Peninsula, and its effect on bedrock uplift rates, Earth and Planetary Science Letters, 473, 164–176, doi: 10.1016/j.epsl.2017.06.002, 2017.

Zwally, H. Jay, Abdalati, W., Herring, T., Larson, K., Saba, J. and Steffen, K.: Surface melt-induced acceleration of Greenland ice-sheet flow, Science (New York, N.Y.), 297, 218–222, doi: 10.1126/science.1072708, 2002.

---

## Referee Report (RR1)

**Reviewer #3, Re-review of: Recent dynamic changes on Fleming Glacier after the disintegration of Wordie Ice Shelf. Friedel et al., 2017**

**Summary**

This revised paper on Fleming Glacier is much improved from its original form. For example, vertically registering the TSX DEM's over sea ice has now been discarded in favor of using ground control points on stable mountains peaks. However, further information on the magnitude of this correction still needs to be provided. While many of the the methodological concerns have been addressed, the authors have not improved their approach to characterizing their measurement error. It is not unreasonable for a reviewer to request a spatially variable error estimate for the datasets presented, therefore the authors must calculate one before this paper is ready for publication, rather than to 'assume an error' as they have stated in their comments. As noted by the authors, the text throughout this paper, particularly in the discussion section, has been completely overhauled which has improved the clarity of the paper. Despite these significant improvements, and the quality of dense ice velocity time series, the authors have not convinced me that the ice thinning rates from the airborne data and DEM differencing are robust. The data has a huge noise range on it which is significantly greater than the stated error, and simply fitting a polynomial through it to get a plausible but unvalidated mean value, doesn't provide enough evidence to conclude with confidence that the magnitude of the thinning signal has been correctly characterized. Given that the authors acknowledge that their result contradicts the previously published estimate by Walker and Gardener (2017), the accuracy of the thinning estimate does need to be properly interrogated before this result is published.

**Specific Edits**

Original comment P5 L6 – The authors have responded to this comment. Although the coverage of ice velocity measurements is different for each image pair, it is possible to state % removed relative to the original pre-filtered result. The approach of only stating % removed on the fast flowing ice tongue is better than nothing, but this number will be biased low because as the authors acknowledge its harder to get good tracking results further inland. In my view, the fact that the measurements are poor in the interior, isn't a good reason for excluding this region from the filter removed stats.

Original comment P5 L16 – The authors have not responded to this important concern. The original comment was that the sensor accuracy is not an estimate of the measurement error on the ice thinning rates. As previously stated 0.2 m is the accuracy of the original point elevation measurements, not the *elevation change* measurement. The authors are also using the elevation data after re-gridding it so they need to state the accuracy of the gridded dataset instead of the accuracy of the raw point data. The elevation change error is different for different sensors, so the authors should provide formal error statistics for each dataset.

Original comment P6 L2/3 – The authors have only partially addressed this comment. Although the method for vertical registration has been completely overhauled to use stable ground control points rather than variable sea ice, the size of the vertical correction has not been stated in the paper. This is important because if the authors are adjusting the DEM heights by 10's of meters, to then measure a few meters of elevation change, then this would indicate that the raw data may not be suitable for the task. Additionally, is the vertical co-registration spatially variable, or is the mean of all ground control points used, and again if it is spatially variable the range of values should be stated. The precise method employed must be stated more clearly.

Original comment P6 L20 – The authors have not sufficiently addressed this comment. As other reviewers also pointed out, having a penetration bias account for 50% of the signal is a significant error therefore it can't be dismissed without a proper solution. The authors statement that they 'have to deal with the data they have in hand' is just not true. There are other available methods of measuring elevation change, such as using altimetry data. The TSX DEM differencing results could be inter-compared with altimetry elevation change from the same time period to establish the extent to which

the dhdt numbers can be trusted.

Original comment P6 L27 – The authors have not addressed this comment. Just because the study area is small, it doesn't mean the error is also small. Mountain glaciers and ice caps are significantly smaller than this Antarctic ice sheet drainage basin, but spatially variable error measurements are still made. It is a completely reasonable reviewer request to ask for a spatially variable error estimate, therefore the authors response to just 'assume an error', rather than to calculate one, is not acceptable.

Original comment P8 L14 – Wording still not correct English, suggest edit to: *Since 2008 the glacier tongue has not advanced seaward of the 1996 grounding line position.*

Original comment P8 L28 – yes fine.

P13, L26-27 – The authors state the following reason for the disagreement of their elevation change result compared with Walker and Gardener (2017), as: 'Their approach of averaging elevation change in 5 km intervals probably filtered out the positive trend that is most prominent on the lowest 3 km of the profile.' It would be trivial for the authors to test this hypothesis, so I suggest they do this rather than postulate, as this new result conflicts with the published literature.

Figure 6 - The wording 'Likely recent grounding line', for the authors new grounding line is not good. It either is a measurement of the new grounding line position, or its not. A time stamp for the new line should also be provided. This lack of commitment to the measurement limits usefulness of the information.

Figure S8 - This figure is incorrect. In 1) there is no dynamic thinning yet because the ice shelf is still providing buttressing force, and the ice velocities are stable. If the ice was dynamically thinning, then the ice velocity would have accelerated. There should be dynamic thinning in 2). And there should also be dynamic thinning in 3) if the ice velocities have continued to accelerate. This schematic is incorrect as it stands, and doesn't really add anything to the discussion so I would just remove the figure.

---

## Author Response (AR2)

**Response to the Re-review of:**

"Recent dynamic changes on Fleming Glacier after the disintegration of Wordie Ice Shelf. Friedl et al., 2017 "

by Peter Friedl et al.

**Anonymous Referee #1**

**We would like to thank the reviewer for taking again the time to comment on our revised manuscript. Our responses are written in bold face type and changes in the manuscript are written in *blue*.**

General comments:

I wish to thank the authors for their efforts in revising the manuscript and preparing the detailed response to the reviews. The revised version addresses adequately the main issues raised by the reviewers. Since submission of the first version of the manuscript two other papers on the dynamic behaviour of Wordie Ice Shelf glaciers were published (Walker and Gardner, 2017; Zhao et al., 2017). The paper of Walker and Gardner covers same topics as this manuscript, dealing with the acceleration and drawdown of the glaciers in connection with break-up of the ice shelf. Beyond that, Walker and Gardner present also a detailed analysis on oceanic and atmospheric conditions and a thorough discussion of driving mechanisms for the observed glacier changes. Friedl et al. present a detailed time series of flow velocities and spatial details on surface elevation change, of significant interest for characterization of glacier behaviour complementary to the material reported in the other publications. However, the discussion and conclusion sections lacks a clear message on the added value of these data for advancing the understanding of flow dynamics, mass balance and driving factors for the observed changes. Main conclusions refer to oceanic and atmospheric driving mechanisms that have been elaborated and reported in much more detail by Walker and Gardner. The relevance of the presented material for further advancing the knowledge on the dynamic response and mass budget of Fleming Glacier should be better worked out.

**We have added some statements on the added value of our data and results to the discussion and conclusion sections. In particular our dense time series of surface velocities is the most detailed information on glacier velocity at Fleming Glacier available so far. It enabled us to precisely date and characterize two pronounced phases of glacier acceleration for the first time. This information is very important for the interpretation of the observed changes. Furthermore we are the first who estimate the recent position of the grounding line on the central part of the glacier, by combining new information on hydrostatic equilibrium with new data on elevation change (TSX/TDX, ICESat), ice velocity and modelled bedrock topography. Whereas earlier publications rely on the grounding line position of 1996, we show that the grounding line had substantially retreated and that this is a key factor for the explanation of recent glacier behaviour. We also see a positive trend of elevation change rates towards the glacier front between 2011 and 2014, which is a sign of floatation of the glacier tongue and which has not been reported before. By relating velocity change and elevation change to grounding line retreat and oceanic forcing we complement previously published data and add new information to the understanding of the pronounced changes at Fleming Glacier and their drivers.**

Further comments:

P2, L18 to L21: Please specify the time periods for the different numbers on basal melt rates and the ratios basal melt/calving flux. This might possibly explain the different numbers.
**Rignot et al. (2013) estimated the basal melt rates for the period 2003-2008 whereas Depoorter et al. (2013) corrected all input data to the year 2009. Differences may also result from different assumptions on ice thickness. We have added the time periods of the basal melt rates throughout the text.**

P3, L3-L4 and P4, L15-L17: The statement " … feature tracking in 2014 and 2015 revealed that the glacier had sped up by ~400–500 m/a which is the largest acceleration in ice flow recorded across all of Antarctica" is incorrect, apparently based on an incomplete sample. This speed up reported by Walker and Gardner (2017) refers to the period 2008 to 2014. In the abstract (P1, L20) Friedl et al. report "a remarkable median speedup of ~1.3 m/d (~30 %) between 2007 and 2011. Much stronger acceleration events are not uncommon for tidewater calving glaciers. For example, the ice flow near the calving front of Hektoria Glacier (Larsen B embayment) accelerated between 2008 and 2011 from 1.4 m/d to 4.2 m/d (300%) (The Cryosphere Discuss., https://doi.org/10.5194/tc-2017-259.. I am sure, detailed analysis of ice motion time series would depict a considerable number of Antarctic glaciers with speed-up >30% during that period.
**We thank the reviewer for pointing this out. We have removed the term "remarkable" in the abstract and in the discussion section. Moreover, we have removed the phrases stating the largest acceleration across Antarctica on P3, L3-L4 and P4, L15-L17**

P9, L23: The term "median absolution deviation" is incorrect. Should rather be "absolute"?
**We are sorry for this typo. We have changed it to "median absolute deviation".**

P9, L25: Not sure if showing only the median velocity of the 16 km profile is the optimum choice. Would be of interest highlighting also velocity changes in subsections of the profile.
**We have largely overhauled the velocity results section by now discussing velocity changes in subsections of the profile more intensively. For this purpose we have also changed Figures 3 a, b. With regard to the reviewer`s comment on P13, L14-15, we have added velocity changes between Jan. 2008–Apr. 2008 and 2010–2011 to the figure and discuss the changes in the text. In order to make the figure tidier, we have binned absolute and relative velocity change by 1 km intervals.**

P13, L14-L15: "Rapidity of acceleration" is not evident in Fig. 2, showing a gradual overall acceleration of 30% between 2007 and 2011 (see also comment P2, L18).
**We assume that the reviewer refers to comment P9, L25 instead of P2, L18. Apart from that, same answer as P9, L25.**

P13, L25-26: There is no "contrast to Walker and Gardner (2017)" regarding the trend of elevation change. Differences are due to the different spatial filtering and the start of the profile further upstream. Besides, Fig. 5a (P27) shows negative dh/dt also in the lowest section, not a positive elevation change for the same ATM data set.
**The reviewer is right that differences to Walker and Gardner (2017) mainly result from their approach of excluding all data up to 5.5 km landward of the 1996 grounding line. Of course for 2011-2014 the elevation change rates in the lowest section are still negative, but the trend in this subsection of the profile clearly goes towards lower ice thinning rates. This difference to**

previous observations (2004 -2008) is important for the explanation of the glacier`s recent behaviour, since this signal now indicates floatation of the adjacent glacier tongue. The observation is in sharp contrast to the statement of Walker and Gardner, 2017 (Supplement, P1 L 37) that 'in all cases, the elevation change rates illustrate a clear trend towards increased surface lowering close to the glacier front'. For a detailed comparison between our data and the data published by Walker and Gardner (2017) we would like to refer the reviewer to our response to reviewer #3. We have changed the section into:

*This has neither been observed by Zhao et al. (2017), who did not specifically analyse elevation change for this time period, nor by Walker and Gardner (2017), who calculated elevation change from the same OIB ATM dataset. The latter binned elevation change in 5 km intervals and excluded all data up to 5.5 km landward of the 1996 grounding line, which is the part of the profile where the trend towards lower ice thinning is most prominent.*

P13, L28: Please specify the periods for the basal melt rates.
**See answer P2, L18 to L21**

P14, L19: In Fig. 8 a retrograde bedrock slope is not obvious for these glaciers. It shows a complex pattern of ridges and depressions.
**If looking at Fig. S6 b (0-6 km) a retrograde bed slope is obvious for the central part of the trough. We have added a reference to Fig. S6 b to the text:**
*Furthermore, the bed topography reveals that the trough underneath the joint Airy-Rotz-Seller-Fleming glacier tongue has a retrograde slope on its central part (Fig. S6 b, 0–6 km).*

**References**

Depoorter, M. A., Bamber, J. L., Griggs, J. A., Lenaerts, J T M, Ligtenberg, S R M, van den Broeke, M R and Moholdt, G.: Calving fluxes and basal melt rates of Antarctic ice shelves, Nature, 502, 89–92, doi: 10.1038/nature12567, 2013.

Rignot, E., Jacobs, S., Mouginot, J. and Scheuchl, B.: Ice-shelf melting around Antarctica, Science (New York, N.Y.), 341, 266–270, doi: 10.1126/science.1235798, 2013.

Walker, C. C. and Gardner, A. S.: Rapid drawdown of Antarctica's Wordie Ice Shelf glaciers in response to ENSO/Southern Annular Mode-driven warming in the Southern Ocean, Earth and Planetary Science Letters, 476, 100–110, doi: 10.1016/j.epsl.2017.08.005, 2017.

Zhao, C., King, M. A., Watson, C. S., Barletta, V. R., Bordoni, A., Dell, M. and Whitehouse, P. L.: Rapid ice unloading in the Fleming Glacier region, southern Antarctic Peninsula, and its effect on bedrock uplift rates, Earth and Planetary Science Letters, 473, 164–176, doi: 10.1016/j.epsl.2017.06.002, 2017.

**Response to the Re-review of:**

"Recent dynamic changes on Fleming Glacier after the disintegration of Wordie Ice Shelf. Friedl et al., 2017 "

by Peter Friedl et al.

**Anonymous Referee #3**

**We would like to thank the reviewer for taking again the time to comment on our revised manuscript. Our responses are written in bold face type and changes in the manuscript are written in** *blue***.**

This revised paper on Fleming Glacier is much improved from its original form. For example, vertically registering the TSX DEM's over sea ice has now been discarded in favor of using ground control points on stable mountains peaks. However, further information on the magnitude of this correction still needs to be provided. While many of the the methodological concerns have been addressed, the authors have not improved their approach to characterizing their measurement error. It is not unreasonable for a reviewer to request a spatially variable error estimate for the datasets presented, therefore the authors must calculate one before this paper is ready for publication, rather than to 'assume an error' as they have stated in their comments. As noted by the authors, the text throughout this paper, particularly in the discussion section, has been completely overhauled which has improved the clarity of the paper. Despite these significant improvements, and the quality of dense ice velocity time series, the authors have not convinced me that the ice thinning rates from the airborne data and DEM differencing are robust. The data has a huge noise range on it which is significantly greater than the stated error, and simply fitting a polynomial through it to get a plausible but unvalidated mean value, doesn't provide enough evidence to conclude with confidence that the magnitude of the thinning signal has been correctly characterized. Given that the authors acknowledge that their result contradicts the previously published estimate by Walker and Gardener (2017), the accuracy of the thinning estimate does need to be properly interrogated before this result is published.

**Specific Edits**
Original comment P5 L6 – The authors have responded to this comment. Although the coverage of ice velocity measurements is different for each image pair, it is possible to state % removed relative to the original pre-filtered result. The approach of only stating % removed on the fast flowing ice tongue is better than nothing, but this number will be biased low because as the authors acknowledge its harder to get good tracking results further inland. In my view, the fact that the measurements are poor in the interior, isn't a good reason for excluding this region from the filter removed stats.
**We have added a column to Tab. S3 which contains the % of removed velocities for each complete velocity field. We have changed the caption of Tab. S3 accordingly.**

Original comment P5 L16 – The authors have not responded to this important concern. The original comment was that the sensor accuracy is not an estimate of the measurement error on the ice thinning rates. As previously stated 0.2 m is the accuracy of the original point elevation measurements, not the elevation change measurement. The authors are also using the elevation data after re-gridding it so they need to state the accuracy of the gridded dataset instead of the accuracy of the raw point data. The

elevation change error is different for different sensors, so the authors should provide formal error statistics for each dataset.

**We have calculated uncertainty statistics for the LiDAR elevation measurements after re-gridding and how this propagates into uncertainties in elevation change rates. We have added some sentences on this to the text:**

*The uncertainty of the gridded measurements can be attributed to both physical topographical features and measurement error. We approximated the uncertainty of each gridded median surface elevation similar as recommended in the IceBridge ATM L2 user guide (https://nsidc.org/data/ILATM2/versions/2/) by calculating the normalized median absolute deviation (NMAD) of all measurements in a 50 m grid cell. For each LiDAR dataset the median NMAD value of all grid cells was taken as the total uncertainty in surface elevation. Total uncertainties in surface elevation were 0.46 m (2004), 0.39 m (2008), 0.38 m (2011) and 0.43 m (2014). For each pair of overlapping gridded median surface elevations we derived the uncertainty in elevation change rate by calculating the root of the sum squares (RSS) of the corresponding NMAD values and dividing the result by the difference in acquisition time (years). The total uncertainty in elevation change rate for a complete dataset was calculated as the median of the elevation change rate uncertainties of all grid cells. The total uncertainties in elevation change rate were 0.32 m $a^{-1}$ for 2004–2008 and 0.24 m $a^{-1}$ for 2011–2014.*

**We have further estimated the uncertainties in ICESat elevation change rates and have added a note on this to the text:**

*By calculating the RSS of the ICESat elevation uncertainties and dividing the result by the time interval between the measurements (years), we estimated the uncertainty in ICESat elevation change rates to be 0.13 m $a^{-1}$.*

Original comment P6 L2/3 – The authors have only partially addressed this comment. Although the method for vertical registration has been completely overhauled to use stable ground control points rather than variable sea ice, the size of the vertical correction has not been stated in the paper. This is important because if the authors are adjusting the DEM heights by 10's of meters, to then measure a few meters of elevation change, then this would indicate that the raw data may not be suitable for the task. Additionally, is the vertical co-registration spatially variable, or is the mean of all ground control points used, and again if it is spatially variable the range of values should be stated. The precise method employed must be stated more clearly.

**We have adjusted each DEM according to the median of all ground control measurements over stable areas (n = 35452). Hence, the adjustment was not spatially variable. The vertical correction was 0.48 m for the 2014 DEM and -5.1 m for the 2011 DEM. We have added a sentence on this to the manuscript:**

*For this purpose the vertical offset between the DEMs and the TanDEM-X global DEM was measured over stable areas (i.e. tops of nunataks and rock outcrops, which were not affected by image distortions) at altitudes between 150 m and 1000 m (Fig. S1). Both DEMs were adjusted according to the median values of all ground control measurements (n= 35452), which were -5.1 m for the 2011 DEM and 0.48 m for the 2014 DEM.*

**However we also would like to note that the amount of vertical adjustment does not necessarily reflect the quality of the DEM. The amount of vertical adjustment strongly depends on the reference height value used for the phase to height conversion of the interferogram. At a single location in the interferogram usually a roughly estimated elevation value is assigned to the corresponding phase difference. The location of the reference point and the assigned height value may vary between different interferograms. According to the phase-to-height-sensitivity, the unwrapped phase differences in the interferogram are then converted to surface heights relative to this elevation value (e.g. 100 m). However, if the real height at the reference point is largely different (e.g. 60 m), all values in the DEM have a huge constant offset to reality (e.g. 60 m). Nonetheless, also in this case the DEM still contains the correct relative height values. After correcting for the offset during vertical registration (e.g. by subtracting 60 m from the DEM),**

**the raw data is leveled to the correct surface height and contains valid information of elevation. In this case it doesn`t matter for the quality of the result whether the vertical correction was 60 m or 0.60 m.**

Original comment P6 L20 – The authors have not sufficiently addressed this comment. As other reviewers also pointed out, having a penetration bias account for 50% of the signal is a significant error therefore it can't be dismissed without a proper solution. The authors statement that they 'have to deal with the data they have in hand' is just not true. There are other available methods of measuring elevation change, such as using altimetry data. The TSX DEM differencing results could be inter-compared with altimetry elevation change from the same time period to establish the extent to which the dhdt numbers can be trusted.

**We now compare the penetration bias corrected TSX/TDX elevation change rates with an independent dataset of 2011-2014 ATM elevation change rates that goes across Fleming Glacier. This dataset is a subsection of the complete OIB flight tracks in 2011 and 2014 and to our knowledge the only existing airborne altimetry data over Fleming Glacier for this time period. The elevation change in the TSX/TDX and the ATM datasets are in good agreement. The RMSE of the cubic fits of both datasets is 0.24 m a⁻¹. At least this gives us a rough estimate of the remaining uncertainties due to uncorrected penetration depth biases and we assume that 0.3 m a⁻¹ is a reasonable value. The value is similar to what has been reported by Rott et al. (2017) for a comparison between TSX/TDX dh/dt and ATM dh/dt measurements over glaciers in the Larsen A and B embayments. We have added Figure S4 e to the supplement, showing the comparison of the TSX/TDX data with the independent ATM data. We have also added the ATM validation track to Figure 4. Furthermore we have added a note on this to the text:**

*Nevertheless, penetration depths of the radar signal may be spatially variable at the same altitude e.g. due to local differences in surface melt or ice properties. Hence we checked the validity of the applied penetration depth correction by comparing the corrected TSX/TDX elevation change rates with an independent validation track of 2011–2014 ATM dh/dt rates (see Fig. 4 for location). Figure S4 e shows that the elevation change rates of both datasets are in good agreement. The RMSE between the cubic fits of the data was 0.24 m a⁻¹. Hence we estimate the uncertainty due to remaining penetration depth differences to be 0.3 m a⁻¹.*

Original comment P6 L27 – The authors have not addressed this comment. Just because the study area is small, it doesn't mean the error is also small. Mountain glaciers and ice caps are significantly smaller than this Antarctic ice sheet drainage basin, but spatially variable error measurements are still made. It is a completely reasonable reviewer request to ask for a spatially variable error estimate, therefore the authors response to just 'assume an error', rather than to calculate one, is not acceptable. **See previous answer.**

Original comment P8 L14 – Wording still not correct English, suggest edit to: *Since 2008 the glacier tongue has not advanced seaward of the 1996 grounding line position.*
**We have changed the sentence accordingly.**

Original comment P8 L28 – yes fine.

P13, L26-27 – The authors state the following reason for the disagreement of their elevation change result compared with Walker and Gardener (2017), as: 'Their approach of averaging elevation change in 5 km intervals probably filtered out the positive trend that is most prominent on the lowest 3 km of

the profile.' It would be trivial for the authors to test this hypothesis, so I suggest they do this rather than postulate, as this new result conflicts with the published literature.

**We have applied the approach of Walker and Gardner (2017) of averaging measurements in 5 km intervals to our re-gridded data (Figure 1), which allows us to fully reproduce their results for 2011–2014 (Figure 2 c, Walker and Gardner, 2017). It is obvious that excluding all data up to 5.5 km landward of the 1996 grounding line almost completely eliminates the trend towards decreasing ice thinning in the lower areas of the glacier tongue. However, if extending the 5 km averaging approach to the complete dataset, the trend is clearly visible. Hence, the statement of Walker and Gardner, 2017 (Supplement, P1 L 37) that 'in all cases, the elevation change rates illustrate a clear trend towards increased surface lowering close to the glacier front' is not true if considering all data.**
**We have changed the sentence by also taking into account the comment of reviewer #1 to:**
*This has neither been observed by Zhao et al. (2017), who did not specifically analyse elevation change for this time period, nor by Walker and Gardner (2017), who calculated elevation change from the same OIB ATM dataset. The latter binned elevation change in 5 km intervals and excluded all data up to 5.5 km landward of the 1996 grounding line, which is the part of the profile where the trend towards lower ice thinning is most prominent.*

[Figure]

**Figure 1:** Elevation change rates on Fleming Glacier in 2011–2014 plotted against distance from the 1996 grounding line. Grey dots: re- gridded elevation change rates from OIB ATM LiDAR measurements in 2011 and 2014. Black line: Cubic function fitted to median filtered elevation change rates. Pink dots: Elevation change rates averaged in 5 km dots as shown by Walker and Gardner, 2017. Pink dotted lines: margins of the 5 km bins used by Walker and Gardner, 2017. Red dot with dotted line: Average elevation change rate in a 5 km interval, if applied to the data excluded by Walker and Gardner, 2017.

Figure 6 - The wording 'Likely recent grounding line', for the authors new grounding line is not good. It either is a measurement of the new grounding line position, or its not. A time stamp for the new line

should also be provided. This lack of commitment to the measurement limits usefulness of the information.

**We have changed the wording in Figures 6 and S7 to 'Interpolated recent GL (2014)'.**

Figure S8 - This figure is incorrect. In 1) there is no dynamic thinning yet because the ice shelf is still providing buttressing force, and the ice velocities are stable. If the ice was dynamically thinning, then the ice velocity would have accelerated. There should be dynamic thinning in 2). And there should also be dynamic thinning in 3) if the ice velocities have continued to accelerate. This schematic is incorrect as it stands, and doesn't really add anything to the discussion so I would just remove the figure.

**We have removed the figure.**

Figure 2 shows that glacier velocities were rather stable between 1994 and January 2008 although almost all of the remaining floating tongue got lost between 1998 and 1999. After 1999 the glacier front remained comparatively steady close to the grounding line location in 1996 for almost 10 years. The NMAD of the median velocities between 1994 and January 2008 was 0.06 m d$^{-1}$ (Fig. 2). Figure 3a reveals A comparison of the velocities on 1995-10-27 and 2007-10-23 (Fig. 3a) along the centreline profile reveals that the median velocity difference between 1997 and January 2008 along the centreline profile was just 0.07 m d$^{-1}$ with maximum velocity differences not exceeding 0.2 m d$^{-1}$.

Between January and April 2008 a rapid and almost constant acceleration of Fleming Glacier of ~0.4 m d$^{-1}$ was noticeable along the centreline until ~13 km upstream (Fig. 3a) and velocities >5 m d$^{-1}$ propagated ~8 km inland (Fig. 2, orange and red colours). The median relative increase in surface velocity between 3 and 13 km upstream was ~8 %, with a maximum increase of ~10 % at ~13 km upstream (Fig. 3b). , which propagated ~8 km inland. Simultaneously, the front of Fleming Glacier retreated behind the 1996 grounding line for the first time. Since 2008 the glacier tongue has not advanced seaward of the 1996 grounding line position (Fig. 2).

The velocity pattern persisted until March 2010, when a second phase of acceleration began. Our velocity time series shows that velocities >5 m d$^{-1}$ the acceleration gradually propagated further inland within one year, until they reached its final extension ~12 km upstream in early 2011. Velocity change during this time period increased towards the glacier front and reached a maximum value of . Fig. 3a, b show absolute and relative velocity changes along the centreline profile for the periods 1995-2007, 2007-2011 and 2011-2015. In 2011 the location of the glacier front reached its most inland position. ~0.9 m d$^{-1}$ or ~16 % at ~6 km upstream. Consequently velocities measured on sea ice or ice mélange in 2011 (seaward of the

 Large changes in surface velocity close to the 2011 front in the periods January 2008–2011, 2010–2011 and 2011–2016 were ignored, since they do not represent real dynamic change, but result from comparing the inherently higher frontal velocities in 2011 with lower velocities of the floating glacier tongue in 2008, 2010 and 2016. If looking at the complete period between January 2008 and 2011

5 the increase in median surface velocity between ~4 and ~7 km upstream  was ~1.3 m d$^{-1}$ or ~32 % (Fig. 3a, b). If ignoring velocity change in the vicinity of the 2011 glacier front, the highest  acceleration values of >1.4 m d$^{-1}$ or ~28 % ~~(~32–35 %)~~ were recorded at ~ 6 km ~~between ~7 and ~11 kmbrises significantly at ~7 km and~~ abruptly drops at ~12 km (Fig. 3 a, b). ~~Peak absolute acceleration values of ~1.6 m d$^{-1}$ were found at ~8 km.~~ 
[revised manuscript text omitted]
 do not see the positive trend of elevation change for the same OIB ATM dataset. Their approach of averaging binned elevation change in 5 km intervals and excluded all data up to 5.5 km landward of the 1996 grounding line, which is the part of the profile where the trend towards lower ice thinning probably filtered out the positive trend that is most prominent. on the lowest 3 km of the profile.

Rignot et al. (2013) and Depoorter et al. (2013) reported high basal melt rates of 23.6 ± 10 m a$^{-1}$ (2003–2008) and 14.79 ± 5.26 m a$^{-1}$ (2009) for the remaining parts of Wordie Ice Shelf, respectively. The magnitude of basal thinning is comparable to those found for ice shelves in the Amundsen Sea sector, where the influx of relatively warm CDW onto the continental shelf is thought to be the dominant driver for recent substantial grounding line retreat, acceleration and dynamic thinning of several glaciers (Turner et al., 2017). Periodical pulses of warm CDW are also known to flood onto the continental shelf of Marguerite Bay (Holland et al., 2010). Significant warming of Antarctic Continental Shelf Bottom Water (ASBW) of 0.1° to 0.3°C decade$^{-1}$ since the 1990s were recorded in the Bellingshausen Sea region and linked to increased warming and shoaling of CDW (Schmidtko et al., 2014). Cook et al. (2016) proposed that oceanic melt induced by an increased shoaling of relatively warm CDW is responsible for an accelerated frontal retreat of tidewater glaciers in the south-western Antarctic Peninsula since the 1990s. Other studies reported considerable thinning of the nearby George VI Ice Shelf (Hogg et al., 2017; Holt et al., 2013) and other ice shelves on the south-western Antarctic Peninsula (e.g. Rignot et al., 2013) due to increased basal melt. The onset of Fleming Glacier`s speedup between January and –April 2008 corresponds well with observations of Wouters et al. (2015). They reported first signs of a near simultaneous increase of ice mass loss for glaciers all across the western Antarctic Peninsula south of -70° since 2008 and an unabated rapid ice loss since 2009. For the glaciers on Western Palmer Land Hogg et al. (2017) showed that ~35% of the ice loss after 2009 can be attributed to dynamic thinning triggered by ocean driven melt.

Walker and Gardner (2017) found that in 2008/2009 and 2010/2011exceptional warm water intrusions into Wordie Bay occurred due to upwelling CDW in response to phases of anomalously strong north-westerly winds during strong La Niña and positive SAM (Southern Annular Mode) events. Highest temperatures were not only recorded at depths between 100–200 m but also at below 400 m, which is close to whereere Fleming Glacier is grounded. The coincident timing with the two phases of glacier acceleration substantiates the link between ocean warming and our observed dynamic changes. It is very

likely that submarine ice melting was increased during phases of strong CDW upwelling and that this has triggered unpinning from the 1996 grounding line position in 2008 as well as further gradual grounding line retreat in 2010–2011.

The strong basal melt rates proposed by Rignot et al. (2013) for 2003–2008 and Depoorter et al. (2013) further suggest that basal melt probably has already occurred prior to 2008. This, together with increased dynamic thinning towards the ice front

5   between 2004 and 2008 has likely weakened the ice at the pinning point, which may have fostered unpinning in 2008. Furthermore, the bed topography reveals that the trough underneath the joint Airy-Rotz-Seller-Fleming glacier tongue has a retrograde slope on its central part (Fig. S6 b, 0–6 km). Such a bed topography is known to be an unstable configuration for the glacier (e.g. DeConto and Pollard, 2016; Favier et al., 2014; Rignot et al., 2014; Schoof, 2012), which may have promoted gradual grounding line retreat between 2010 and 2011. Figure S8 summarizes our interpretation of grounding line

10  retreat at Fleming Glacier.

The bedrock topography of Fleming Glacier also shows a retrograde bed slope starting at ~3–4 km upstream of the current grounding line, which transitions into a pronounced deep trough (up to 1100 m below sea level) at about 10 km upstream. Hence, if grounding line retreat exceeds the edge of this trough, destabilisation like on Thwaites Glacier and in the Pine Island Bay region is possible, which would involve further rapid grounding line retreat and amplified mass loss in the future

15  (Favier et al., 2014; Rignot et al., 2014).

**7 Conclusions**

We present a detailed history of the glacier dynamics of Fleming Glacier after the retreat and disintegration of Wordie Ice Shelf. by analysing glacier extent, surfaces velocities, elevation change rates and hydrostatic equilibrium. While previous studies analysed only rather limited amounts of velocity data (Rignot et al., 2005; Walker and Gardner, 2017; Wendt et al.,

20  2010; Zhao et al., 2017), we now show a complete time series of SAR surface velocities of Fleming Glacier at high temporal resolution for the period 1994–2016. These data enable a much better temporal constraint and characterisation of glaciological changes in the region. By combining this unique dataset with recently published data on oceanic forcing (Walker and Gardner, 2017) and new data on surface elevation change from TSX/TDX interferometry, ICESat laser altimetry and airborne LiDAR, as well as modelled bedrock topography and hydrostatic equilibrium, we are able to relate

25  precisely dated events of acceleration to increased dynamic ice thinning and ocean driven grounding line retreat. This complements previous studies in the region and provides new and more detailed information on the glaciological changes and their drivers. Especially our dense SAR time series enables us to precisely date events of velocity change jointly with glacier retreat, elevation changes and grounding line retreat.

[revised manuscript text omitted]

**Supplement**

**S1 Data used for front line delineation**

**Table S1:** Data used for mapping the front positions in Fig. 1, 2, 3, 6 and S6.

| Year | Platform | Reference | Date |
| --- | --- | --- | --- |
| 1966 | Aerial photography | Ferrigno, 2008 | 11/12 1966 |
| 1974 | Landsat-1 | Ferrigno, 2008 | 1974-01-06 |
| 1989 | Landsat-3 | Ferrigno, 2008 | 1989-02-20 |
| 1994 | ERS-1/2 | | 1994-02-01 |
| 1995 | ERS-1/2 | | 1995-10-27 |
| 1996 | ERS-1/2 | | 1996-02-10 |
| 1997 | ERS-1/2 | | 1997-01-29 |
| 1998 | ERS-1/2 | | 1998-01-30 |
| 1999 | ERS-1/2 | | 1999-11-10 |
| 2000 | ERS-1/2 | | 2000-02-20 |
| 2001 | Landsat-7 | | 2001-01-04 |
| 2002 | ERS-1/2 | | 2002-02-23 |
| 2003 | ERS-1/2 | | 2003-01-24 |
| 2004 | ERS-1/2 | | 2004-03-03 |
| 2005 | ERS-1/2 | | 2005-01-28 |
| 2007 | ERS-1/2 | | 2007-02-02 |
| 2008* | Envisat | Wendt et al., 2010 | 2008-04-13 |
| 2008* | Envisat | | 2008-11-08 |
| 2009 | ASTER | Wendt et al., 2010 | 2009-02-02 |
| 2010 | ERS-1/2 | | 2010-02-26 |
| 2011 | TSX/TDX | | 2011-11-23 |
| 2012 | TSX/TDX | | 2012-10-16 |
| 2013 | TSX/TDX | | 2013-12-08 |
| 2014 | TSX/TDX | | 2014-11-03 |
| 2015 | Sentinel 1a | | 2015-09-09 |
| 2016 | Sentinel 1a | | 2016-01-31 |

**\***The 2008-04-13 front position is shown in Fig. 1 and the 2008-11-08 front position is shown in Fig. 6 and S6.

**S2 Error estimation of surface velocity measurements**

The corresponding errors of the velocity measurements were estimated as described in detail in Seehaus et al. (2015). It is assumed that the resulting uncertainties for each velocity field are induced by two major sources: the coregistration process and the tracking algorithm itself. The error caused by residual inaccuracies of the coregistration ($\sigma_V^C$) was determined by calculating the median velocity for 19 to 64 points on stable non-moving surfaces (e.g. rock outcrops) (Fig. S1). The error induced by the tracking algorithm ($\sigma_V^T$) was estimated according to the following formula, modified from McNabb et al. (2012):

$$\sigma_V^T = \frac{C\Delta x}{z\Delta t} \tag{1}$$

where $C$ is the uncertainty of the tracking algorithm (assumed to be 0.4 pixels), $\Delta x$ is the image resolution (mean values for each sensor are listed in Tab. S2), $z$ is the oversampling factor (we applied a factor of two) and $\Delta t$ is the temporal baseline between the SAR images. The total error ($\sigma_V$) of the velocity measurement is the sum of $\sigma_V^C$ and $\sigma_V^T$. Table S3 lists the values $\Delta t$, $\sigma_V^C$, $\sigma_V^T$ and $\sigma_V$ for each velocity field. As in Seehaus et al. (2015) the quite large $\sigma_V^T$ values for ERS-1/2 measurements during one of the sensor`s "Tandem" or "Ice Phases", where the satellites orbited in 1- or 3-day repeat passes, were excluded from our estimation of $\sigma_V$.

[Figure]

**Figure S1:** Blue dots: Stable points used for the coregistration error estimation of the intensity tracking results; Black polygon: reference area for the calculation of the proportion of velocity measurements removed by the filter of Burgess et al. (2012) (values of each velocity field are listed in Tab. S.3). Orange polygons: stable areas on nunataks and hills for accuracy assessment of elevation change measurements. Pink polygons: stable areas on nunataks and hills for vertical height referencing of the TanDEM-X DEMs. Purple polygon: spatial coverage of the TanDEM-X DEMs. Background: Mosaic of two Landsat-8 „Natural Color" images, acquired on September 16, 2015 ©USGS.

**Table S2:** Ground range resolution $\Delta x_{GR}$, azimuth resolution $\Delta x_{AZ}$ and image resolution $\Delta x$ used in Formula 1 (S2) for calculating velocity errors. For most of the sensors $\Delta x_{GR}$ is coarser than $\Delta x_{AZ}$, except for Sentinel-1a that has a coarser resolution in azimuth direction and TSX/TDX which have fairly equal resolutions in both directions. In order to make a conservative estimate of velocity errors, always the coarser resolution value was chosen to be $\Delta x$.

| Sensor | $\Delta x_{GR}$ [m] | $\Delta x_{AZ}$ [m] | $\Delta x$ [m] |
|---|---|---|---|
| AMI SAR (ERS-1/2) | 20 | 4 | 20 |
| R1 (Radarsat 1) | 20 | 4 | 20 |
| ASAR (Envisat) | 17 | 4 | 17 |
| PALSAR (ALOS) | 7 | 3 | 7 |
| TSX/TDX(TerraSAR-X/TanDEM-X) | 2 | 2 | 2 |
| S1 (Sentinel-1a) | 3 | 14 | 14 |

**Table S3:** Uncertainty $\sigma_V$ of processed velocity fields and proportion of removed velocity measurements by  filter  Burgess et al. (2012) . Date: Mean date of SAR acquisitions; $\Delta t$: Time interval in days between repeat SAR acquisitions; $\sigma_V^C$: Uncertainty of image coregistration; $\sigma_V^T$: Uncertainty of intensity tracking processing; ERS velocity fields with $\Delta t \leq 3d$: $\sigma_V = \sigma_V^C$. The percentage of filtered velocities is listed for a reference area on the tongue of Fleming Glacier (Fig. S.1) and each complete velocity field. The table is continued on the next pages.

| Date [yyyy-mm-dd] | Sensor | $\Delta t$ [d] | $\sigma_V^C$ [m d$^{-1}$] | $\sigma_V^T$ [m d$^{-1}$] | $\sigma_V$ [m d$^{-1}$] | Removed by filter (area) [%] | Removed by filter (total) [%] |
|---|---|---|---|---|---|---|---|
| 1994-01-27 | AMI SAR | 3 | 0.13 | 1.33 | 0.13 | 1.47 | 20.58 |
| 1994-02-05 | AMI SAR | 3 | 0.19 | 1.33 | 0.19 | 1.01 | 22.36 |
| 1994-02-23 | AMI SAR | 3 | 0.14 | 1.33 | 0.14 | 3.85 | 22.04 |
| 1994-02-26 | AMI SAR | 3 | 0.34 | 1.33 | 0.34 | 4.41 | 19.4 |
| 1994-03-07 | AMI SAR | 3 | 0.2 | 1.33 | 0.2 | 1.93 | 23.96 |
| 1995-10-27 | AMI SAR | 1 | 0.25 | 4 | 0.25 | 1.34 | 14.48 |
| 1996-02-09 | AMI SAR | 35 | 0.12 | 0.11 | 0.23 | 3.88 | 11.19 |
| 1997-02-27 | AMI SAR | 35 | 0.12 | 0.11 | 0.23 | 4.91 | 22.04 |
| 2000-09-19 | R1 | 24 | 0.11 | 0.14 | 0.25 | 7.1 | 27.19 |
| 2000-10-13 | R1 | 24 | 0.11 | 0.14 | 0.25 | 4.58 | 27.32 |
| 2002-12-02 | AMI SAR | 35 | 0.19 | 0.11 | 0.3 | 1.64 | 15.37 |
| 2003-01-06 | AMI SAR | 35 | 0.13 | 0.11 | 0.24 | 1.83 | 14.78 |
| 2003-10-22 | R1 | 24 | 0.12 | 0.14 | 0.26 | 2.98 | 30.17 |
| 2003-11-15 | R1 | 24 | 0.19 | 0.14 | 0.33 | 4.21 | 28.82 |
| 2003-12-09 | R1 | 24 | 0.09 | 0.14 | 0.23 | 4.38 | 28.71 |
| 2004-02-19 | R1 | 24 | 0.08 | 0.14 | 0.22 | 2.63 | 29.25 |
| 2004-03-14 | R1 | 24 | 0.12 | 0.14 | 0.26 | 3.19 | 28.85 |
| 2004-04-07 | R1 | 24 | 0.17 | 0.14 | 0.31 | 3.36 | 28.55 |
| 2004-09-22 | R1 | 24 | 0.1 | 0.14 | 0.24 | 3.61 | 30.05 |
| 2004-10-16 | R1 | 24 | 0.21 | 0.14 | 0.35 | 3.91 | 29.8 |
| 2004-11-09 | R1 | 24 | 0.09 | 0.14 | 0.23 | 2.63 | 31.13 |
| 2004-12-03 | R1 | 24 | 0.14 | 0.14 | 0.28 | 3.49 | 30.63 |
| 2004-12-27 | R1 | 24 | 0.15 | 0.14 | 0.29 | 13.79 | 19.03 |
| 2005-01-20 | R1 | 24 | 0.26 | 0.14 | 0.4 | 11.63 | 17.7 |
| 2005-02-13 | R1 | 24 | 0.07 | 0.14 | 0.21 | 14.86 | 18.07 |
| 2005-04-26 | R1 | 24 | 0.19 | 0.14 | 0.33 | 2.59 | 30.13 |
| 2005-10-11 | R1 | 24 | 0.13 | 0.14 | 0.27 | 18.38 | 17.42 |
| 2005-11-04 | R1 | 24 | 0.17 | 0.14 | 0.31 | 1.75 | 30.48 |
| 2006-01-15 | R1 | 24 | 0.12 | 0.14 | 0.26 | 10.81 | 15.89 |
| 2006-02-08 | R1 | 24 | 0.16 | 0.14 | 0.3 | 1.7 | 29.42 |
| 2006-02-15 | ASAR | 35 | 0.15 | 0.11 | 0.26 | 3.29 | 11.15 |

| Date [yyyy-mm-dd] | Sensor | $\Delta t$ [d] | $\sigma_V^C$ [m d$^{-1}$] | $\sigma_V^T$ [m d$^{-1}$] | $\sigma_V$ [m d$^{-1}$] | Removed by filter (area) [%] | Removed by filter (total) [%] |
|---|---|---|---|---|---|---|---|
| 2006-03-04 | R1 | 24 | 0.06 | 0.14 | 0.2 | 1.23 | 30.23 |
| 2006-03-28 | R1 | 24 | 0.23 | 0.14 | 0.37 | 1.27 | 30.25 |
| 2006-04-21 | R1 | 24 | 0.17 | 0.14 | 0.31 | 1.25 | 30.13 |
| 2006-05-31 | ASAR | 35 | 0.09 | 0.11 | 0.2 | 3.35 | 8.81 |
| 2006-07-05 | ASAR | 35 | 0.11 | 0.11 | 0.22 | 3.2 | 8.74 |
| 2006-07-18 | PALSAR | 46 | 0.06 | 0.03 | 0.09 | 27.76 | 12.77 |
| 2006-08-09 | ASAR | 35 | 0.14 | 0.11 | 0.25 | 3.82 | 8.82 |
| 2006-11-03 | AMI SAR | 35 | 0.18 | 0.11 | 0.29 | 3.1 | 9.16 |
| 2007-04-10 | ASAR | 35 | 0.17 | 0.11 | 0.28 | 11.26 | 8.89 |
| 2007-05-15 | ASAR | 35 | 0.06 | 0.11 | 0.17 | 1.29 | 13.54 |
| 2007-05-16 | ASAR | 35 | 0.09 | 0.11 | 0.2 | 3.44 | 9.7 |
| 2007-06-19 | ASAR | 35 | 0.05 | 0.11 | 0.16 | 3.44 | 12.76 |
| 2007-06-20 | ASAR | 35 | 0.08 | 0.11 | 0.19 | 1.33 | 11.22 |
| 2007-07-25 | ASAR | 35 | 0.05 | 0.11 | 0.16 | 3.92 | 11.41 |
| 2007-08-28 | ASAR | 35 | 0.1 | 0.11 | 0.21 | 1.41 | 13.03 |
| 2007-08-29 | ASAR | 35 | 0.13 | 0.11 | 0.24 | 4.08 | 10.76 |
| 2007-10-02 | ASAR | 35 | 0.08 | 0.11 | 0.19 | 1.45 | 11.25 |
| 2007-10-03 | ASAR | 35 | 0.11 | 0.11 | 0.22 | 4.25 | 10.71 |
| 2007-10-23 | PALSAR | 46 | 0.05 | 0.03 | 0.08 | 20.19 | 12.04 |
| 2007-11-06 | ASAR | 35 | 0.06 | 0.11 | 0.17 | 1.63 | 13.35 |
| 2008-01-05 | R1 | 24 | 0.2 | 0.14 | 0.34 | 1.95 | 29.79 |
| 2008-04-30 | ASAR | 35 | 0.11 | 0.11 | 0.22 | 5.52 | 11.36 |
| 2008-06-03 | ASAR | 35 | 0.08 | 0.11 | 0.19 | 1.97 | 14.19 |
| 2008-07-08 | ASAR | 35 | 0.06 | 0.11 | 0.17 | 2.05 | 14.17 |
| 2008-08-12 | ASAR | 35 | 0.08 | 0.11 | 0.19 | 2.37 | 13.69 |
| 2008-09-16 | ASAR | 35 | 0.08 | 0.11 | 0.19 | 2.22 | 14.31 |
| 2008-10-21 | ASAR | 35 | 0.07 | 0.11 | 0.18 | 2.45 | 14.51 |
| 2009-02-06 | PALSAR | 46 | 0.15 | 0.03 | 0.18 | 26.83 | 14.54 |
| 2009-04-14 | ASAR | 35 | 0.22 | 0.11 | 0.33 | 6.95 | 7.04 |
| 2009-07-29 | ASAR | 35 | 0.09 | 0.11 | 0.2 | 6.57 | 11.92 |
| 2009-09-02 | ASAR | 35 | 0.09 | 0.11 | 0.2 | 6.44 | 10.57 |
| 2009-10-07 | ASAR | 35 | 0.09 | 0.11 | 0.2 | 6.63 | 10.48 |
| 2010-02-08 | ASAR | 35 | 0.08 | 0.11 | 0.19 | 4.69 | 11.03 |
| 2010-02-11 | ASAR | 35 | 0.19 | 0.11 | 0.3 | 0.02 | 5.38 |
| 2010-03-31 | ASAR | 35 | 0.09 | 0.11 | 0.2 | 5.16 | 9.52 |

| Date [yyyy-mm-dd] | Sensor | $\Delta t$ [d] | $\sigma_V^C$ [m d$^{-1}$] | $\sigma_V^T$ [m d$^{-1}$] | $\sigma_V$ [m d$^{-1}$] | Removed by filter (area) [%] | Removed by filter (total) [%] |
|---|---|---|---|---|---|---|---|
| 2010-05-05 | ASAR | 35 | 0.1 | 0.11 | 0.21 | 5.49 | 11.38 |
| 2010-06-09 | ASAR | 35 | 0.08 | 0.11 | 0.19 | 5.29 | 11.08 |
| 2010-07-14 | ASAR | 35 | 0.09 | 0.11 | 0.2 | 5.39 | 11.51 |
| 2010-08-18 | ASAR | 35 | 0.1 | 0.11 | 0.21 | 5.87 | 11.49 |
| 2010-09-22 | ASAR | 35 | 0.09 | 0.11 | 0.2 | 6.04 | 11.49 |
| 2010-10-27 | TSX/TDX | 33 | 0.01 | 0.01 | 0.02 | 3.7 | 12.01 |
| 2010-10-31 | PALSAR | 46 | 0.33 | 0.03 | 0.36 | 28.43 | 10.62 |
| 2010-11-18 | TSX/TDX | 11 | 0.02 | 0.04 | 0.06 | 4.06 | 11.68 |
| 2011-05-10 | AMI SAR | 3 | 0.25 | 1.33 | 0.25 | 2.52 | 19.79 |
| 2011-06-27 | AMI SAR | 3 | 0.36 | 1.33 | 0.36 | 6.46 | 20.24 |
| 2011-10-02 | TSX/TDX | 11 | 0.02 | 0.04 | 0.06 | 2.94 | 8.23 |
| 2011-11-15 | TSX/TDX | 11 | 0.03 | 0.04 | 0.07 | 5.02 | 10.86 |
| 2011-12-29 | TSX/TDX | 11 | 0.01 | 0.04 | 0.05 | 6.47 | 10.21 |
| 2012-03-15 | TSX/TDX | 11 | 0.02 | 0.04 | 0.06 | 3.55 | 9.64 |
| 2012-04-06 | TSX/TDX | 11 | 0.02 | 0.04 | 0.06 | 4.56 | 10.76 |
| 2012-04-17 | TSX/TDX | 11 | 0.02 | 0.04 | 0.06 | 3.66 | 9.34 |
| 2012-05-31 | TSX/TDX | 11 | 0.03 | 0.04 | 0.07 | 4.14 | 10.34 |
| 2012-06-11 | TSX/TDX | 11 | 0.01 | 0.04 | 0.05 | 4.01 | 10.41 |
| 2012-07-25 | TSX/TDX | 11 | 0.02 | 0.04 | 0.06 | 1.69 | 10.52 |
| 2012-09-07 | TSX/TDX | 11 | 0.02 | 0.04 | 0.06 | 4.53 | 10.29 |
| 2012-10-21 | TSX/TDX | 11 | 0.02 | 0.04 | 0.06 | 4.34 | 10.33 |
| 2012-12-04 | TSX/TDX | 11 | 0.02 | 0.04 | 0.06 | 3.37 | 9.84 |
| 2013-01-17 | TSX/TDX | 11 | 0.01 | 0.04 | 0.05 | 3.94 | 8.18 |
| 2013-03-13 | TSX/TDX | 11 | 0.03 | 0.04 | 0.07 | 3.8 | 9.63 |
| 2013-03-23 | TSX/TDX | 11 | 0.08 | 0.04 | 0.12 | 2.19 | 6.64 |
| 2013-04-15 | TSX/TDX | 11 | 0.03 | 0.04 | 0.07 | 3.47 | 10.15 |
| 2013-06-09 | TSX/TDX | 11 | 0.02 | 0.04 | 0.06 | 2.56 | 8.79 |
| 2013-06-20 | TSX/TDX | 11 | 0.02 | 0.04 | 0.06 | 3.94 | 8.76 |
| 2013-07-01 | TSX/TDX | 11 | 0.02 | 0.04 | 0.06 | 2.37 | 8.48 |
| 2013-07-12 | TSX/TDX | 11 | 0.02 | 0.04 | 0.06 | 2.2 | 8.48 |
| 2013-11-20 | TSX/TDX | 11 | 0.04 | 0.04 | 0.08 | 0.71 | 5.77 |
| 2013-12-13 | TSX/TDX | 11 | 0.03 | 0.04 | 0.07 | 4.25 | 8.29 |
| 2013-12-24 | TSX/TDX | 11 | 0.02 | 0.04 | 0.06 | 3.57 | 8.02 |
| 2014-01-26 | TSX/TDX | 11 | 0.01 | 0.04 | 0.05 | 3.2 | 7.72 |
| 2014-08-23 | TSX/TDX | 11 | 0.02 | 0.04 | 0.06 | 4.67 | 10.98 |

| Date | Sensor | $\Delta t$ | $\sigma_V^C$ | $\sigma_V^T$ | $\sigma_V$ | Removed by | Removed by |
|------|--------|-----------|--------------|--------------|------------|------------|------------|
| [yyyy-mm-dd] | | [d] | [m d$^{-1}$] | [m d$^{-1}$] | [m d$^{-1}$] | filter (area) [%] | filter (total) [%] |
| 2014-09-03 | TSX/TDX | 11 | 0.03 | 0.04 | 0.07 | 4.52 | 10.36 |
| 2014-12-11 | TSX/TDX | 11 | 0.01 | 0.04 | 0.05 | 4.05 | 6.79 |
| 2015-01-24 | TSX/TDX | 11 | 0.03 | 0.04 | 0.07 | 2.2 | 9.04 |
| 2015-09-03 | S1 | 12 | 0.06 | 0.23 | 0.29 | 2.47 | 19.85 |
| 2015-09-15 | S1 | 12 | 0.05 | 0.23 | 0.28 | 2.51 | 20.36 |
| 2015-10-21 | S1 | 12 | 0.15 | 0.23 | 0.38 | 2.51 | 19.74 |
| 2015-12-08 | S1 | 12 | 0.11 | 0.23 | 0.34 | 2.03 | 12.42 |
| 2015-12-20 | S1 | 12 | 0.07 | 0.23 | 0.3 | 2.13 | 13.99 |
| 2016-01-01 | S1 | 12 | 0.09 | 0.23 | 0.32 | 2.21 | 13.15 |
| 2016-01-13 | S1 | 12 | 0.05 | 0.23 | 0.28 | 2.08 | 13.79 |
| 2016-01-25 | S1 | 12 | 0.08 | 0.23 | 0.31 | 2.18 | 13.59 |
| 2016-02-06 | S1 | 12 | 0.08 | 0.23 | 0.31 | 2.15 | 12.85 |
| 2016-02-18 | S1 | 12 | 0.08 | 0.23 | 0.31 | 2.1 | 13.56 |
| 2016-03-01 | S1 | 12 | 0.09 | 0.23 | 0.32 | 2.15 | 12.63 |
| 2016-03-13 | S1 | 12 | 0.08 | 0.23 | 0.31 | 2.11 | 13.58 |
| 2016-04-06 | S1 | 12 | 0.09 | 0.23 | 0.32 | 2.01 | 12.64 |
| 2016-04-18 | S1 | 12 | 0.07 | 0.23 | 0.3 | 2.12 | 13.43 |
| 2016-06-05 | S1 | 12 | 0.12 | 0.23 | 0.35 | 1.97 | 13.02 |
| 2016-07-23 | S1 | 12 | 0.1 | 0.23 | 0.33 | 2.09 | 12.36 |
| 2016-08-04 | S1 | 12 | 0.08 | 0.23 | 0.31 | 2.07 | 12.51 |
| 2016-08-16 | S1 | 12 | 0.06 | 0.23 | 0.29 | 2.17 | 14.7 |

**S3 Uncertainty estimation of variables for the hydrostatic height anomaly calculations and assessment of error propagation**

In order to assess the propagation of uncertainties for the calculation of hydrostatic height anomalies, we estimated the error of each variable of Formula 2. The accuracy of the ATM elevations was estimated to be ± 0.2 m. The overall uncertainty of the EIGEN-6C4 geoid is 0.24 m (http://icgem.gfz-potsdam.de/ICGEM/). Accounting for an unknown additional error induced by kriging of the geoid values, we assumed a total accuracy of ± 0.5 m for $e$. Following the recommendation of the CReSIS Radar Depth Sounders (RDS) user guide (ftp://data.cresis.ku.edu/data/rds/rds_readme.pdf) we defined the error of $H_i$ as the sum of the RMS error of the sensor`s range resolution and the RMS error of the dielectric. Depending on the sensor, the uncertainty of $H_i$ varies between ~6 and ~14 m for an ice thickness of 750 m, which is the approx. mean ice thickness in the vicinity of the current grounding line measured on our OIB profiles. For $\rho_i$ we used a value of 917 kg m$^{-3}$, which is the standard density of pure ice (Benn and Evans, 2013). However, since impurities in the ice can cause this value to vary by around ± 5 kg m$^{-3}$ (Griggs and Bamber, 2011), we chose this rate to be the uncertainty of $\rho_i$. The global mean density of sea water is 1027 kg m$^{-3}$, but this value can vary locally. According to Griggs and Bamber (2011) we therefore assumed an error of ± 5 kg m$^{-3}$ for $\rho_w$. The firn density correction factor for pure glacier ice is 0. In situ values of about 10 m have been measured for $\delta$ on Larsen C Ice Shelf (Griggs and Bamber, 2009) and firn density correction factors > 20 m have been reported for areas of convergent flow on the Ross Ice Shelf (Bamber and Bentley, 1994). Modelled firn densities (van den Broeke, Michiel et al., 2008) indicate a firn correction factor of ~17 ± 4 m for the glacier tongue of the Airy-Rotz-Seller-Fleming glacier system.  In order to quantify the total error of $\Delta e$ and to consider the propagation of uncertainties, we run a Monte Carlo simulation based on Formula 1 and 2 with 100.000 runs. For all possible sensor-depending errors of $H_i$=750 m the Monte Carlo simulation yielded a standard deviation of ~< 12 m for $\Delta e$. Thus we assumed the total uncertainty of $\Delta e$ to be 12 m. Consequently, we assigned locations on our OIB and PIB profiles to be freely floating ice, if the calculated values of $\Delta e$ lay within this range. In the vicinity of the grounding zone the TDX global DEM has a minimum slope of 1°, and therefore we estimate the uncertainty in the horizontal position of the transition from grounded to freely floating ice to be ~350 m.

**S4 Penetration bias correction of TSX/TDX 2011–2014 elevation change rates**

**Figure S4 a–c:** Penetration bias correction of TSX/TDX 2011–2014 elevation change rates

**(a)** Comparison of yearly elevation change rates obtained from OIB ATM LiDAR measurements (2011-11-17/2014-11-10) and TSX/TDX DEMs (2011-11-21/2014-11-03) after vertical registration of the TSX/TDX DEMs. Data is plotted against absolute ellipsoidal elevations from the resampled Bedmap 2 DEM (Fretwell et al., 2013). Grey dots: elevation change rates between 2011 and 2014 from OIB ATM. Black line: cubic function fitted to the OIB ATM measurements. Light blue dots: elevation change rates between 2011 and 2014 from TSX/TDX. Blue line: cubic function fitted to the TSX/TDX measurements

[Figure]

**(b)** Penetration bias correction model for TSX/TDX change rates. Data is plotted against absolute ellipsoidal elevations from the resampled Bedmap 2 DEM (Fretwell et al., 2013). Black dots: differences between TSX/TDX elevation change rates ($r_{DEM}$) and OIB ATM rates ($r_{ATM}$). Green line: local polynomial model fitted to the measurements.

[Figure]

**(c)** Comparison of yearly elevation change rates obtained from OIB ATM LiDAR measurements (2011-11-17/2014-11-10) and TSX/TDX DEMs (2011-11-21/2014-11-03) after vertical registration and penetration depth bias correction of the TSX/TDX elevation change map. Data is plotted against absolute ellipsoidal elevations from the resampled Bedmap 2 DEM (Fretwell et al., 2013). Grey dots: elevation change rates between 2011 and 2014 from OIB ATM. Black line: cubic function fitted to the OIB ATM measurements. Light blue dots: elevation change rates between 2011 and 2014 from TSX/TDX. Blue line: cubic function fitted to the TSX/TDX measurements

[Figure]

**(d)** Comparison of median filtered backscatter values of TSX/TDX acquisitions on 2011-11-21 and 2014-11-03. Data is plotted against absolute ellipsoidal elevations from the resampled Bedmap 2 DEM (Fretwell et al., 2013). Light pink dots: backscatter of the master image on 2011-11-21. Brown line: cubic function fitted to the 2011-11-21 backscatter values. Light blue dots: backscatter of the master image on 2014-11-03. Dark blue line: cubic function fitted to the 2014-11-03 backscatter values.

[Figure]

**(e)** Comparison of yearly elevation change rates along a validation track of OIB ATM LiDAR measurements (2011-11-17/2014-11-10) (Figure 4) and elevation change rates from the penetration depth corrected differential TSX/TDX DEM (2011-11-21/2014-11-03). Data is plotted against absolute ellipsoidal elevations from the resampled Bedmap 2 DEM (Fretwell et al., 2013). Grey dots: elevation change rates between 2011 and 2014 from OIB ATM. Black line: cubic function fitted to the OIB ATM measurements. Light blue dots: elevation change rates between 2011 and 2014 from TSX/TDX. Blue line: cubic function fitted to the TSX/TDX measurements

[Figure]

**S5 Results of the hydrostatic height anomaly calculations**

**Figure S5 a–e:** Fulfillment of the hydrostatic equilibrium assumption from hydrostatic height anomaly calculations along PIB and OIB profiles. Profiles are to read from left to right (in upstream direction). Dates of PIB and OIB flights: a) 2002-11-26, b) 2004-11-18, c) 2011-11-17, d) 2014-11-16, e) 2014-11-10. Purple dots: PIB/OIB ice surface/bottom elevations. Ice surface elevation is taken from PIB/OIB ATM measurements and ice bottom elevation is calculated by subtracting OIB/PIB ice thickness from ice surface elevation. Brown line: Bedrock elevation from Huss and Farionotti, 2014. Yellow line: Bedrock elevation from Bedmap 2 (Fretwell et al., 2013). Red and blue dots: calculated ice surface elevation in hydrostatic equilibrium $e_{he}$ and information on hydrostatic equilibrium (blue: freely floating ice, red: grounded ice)

[Figure]

[Figure]

[Figure]

[Figure]

**S6 Estimation of grounding line positions along profiles of surface velocity, bedrock topography, ice elevations and hydrostatic height anomalies**

**Figure S6 a–d:** Estimated grounding line positions on profiles a–d (for location see Fig. 6) based on surface velocities and elevation change rates from TDX (black line). Distances for Fig. S6 a–d are relative to the 1996 grounding line. F: Front, GL: Grounding Line. Brown line: Bedrock elevation from Huss and Farinotti (2014). Yellow line: Bedrock elevation from Bedmap 2 (Fretwell et al. 2013).

[Figure]

Figure S6 a exhibits that in contrast to Fleming Glacier, on Airy Glacier velocity data extracted along Profile a (Fig. 6) does not show signs of acceleration between 2007 and 2008. Furthermore in 2008 the front was still located at the 1996 grounding line. Hence, we assume that in 2008 Airy Glacier was still grounded at the 1996grounding line position. However, in 2011 the front had retreated behind the 1996 grounding line. This shows that the grounding line must have had retreated from its 1996 position by this time. An abrupt change in the 2011–2014 TSX/TDX elevation change rates is visible at 4–5 km upstream, which coincides with the edge of the subglacial trough in the bedrock data. A similar extent of accelerated surface

velocities is visible on the velocity profiles in 2011 and 2015. Hence, this location is likely the recent location of the grounding line. A distinct area of low elevation change rates which is visible on the TSX/TDX dh/dt map (Fig. S7) suggests that the entire Airy Glacier tongue may be currently floating up to ~4 km upstream of the 1996 grounding line. However, this is in contrast to the buoyancy calculations (Fig. 6, Track 3) which indicate that in 2011 the ice was grounded on a hill ~2 km
5    upstream, which reaches to the subglacial trough. Since we assume that the hydrostatic height anomalies from 2011 are the most reliable source of information for grounding line detection we have, we delineated the recent grounding line accordingly. However, we cannot rule out that the ice which rested on the hill in 2011 is afloat today.

[Figure]

Figure S6 b depicts data extracted along Profile b (Fig. 6), close to the confluence of Fleming and Seller Glacier. Here
10   velocities show acceleration between 2007 and 2008. While the glacier had accelerated, the glacier front had retreated behind the 1996 grounding line. Hence, in 2008 the grounding line must have been located upstream of the 1996 position. The

upstream extent of acceleration visible on the velocity profile suggests that in 2008 the grounding line had not retreated as far as in 2011 and 2015, yet. A reasonable estimate would be that the grounding line in 2008 was located on a gentle hill apparent in the bedrock data close to the 2011 front, ~4 km upstream. However, given the limitations of the modelled bedrock topography (Sect. 3) and the lack of further information, the exact grounding line position in 2008 remains vague.

5    In 2011 the glacier had substantially accelerated and in 2015 high velocities had further propagated inland by ~1 km up to the edge of the subglacial trough at ~9 km upstream. The limit of high velocities in 2015 coincides with a marked change of the TSX/TDX 2011–2014 elevation change rates, indicating that at this position the ice starts to float. Based on these evidences, we estimate that the grounding line position in 2011 was likely already located at the edge of the subglacial trough and had probably further retreated by ~1 km in 2015.

[Figure]

Figure S6 c reveals that velocities extracted along Profile 3 (Fig. 6) show an upstream propagation of high velocities between 2007 and 2008 of up to 11 km. In 2008, the glacier front had retreated behind the 1996 grounding line. This indicates that in 2008 the grounding line must have been located upstream from its 1996 position. Since the acceleration had not yet propagated as far upstream as in 2011 and 2015, we assume that the 2008 grounding line was located more seawards than in 2011 and 2015 respectively. A possible grounding line location in 2008 is a smaller hill visible in the bedrock data close to the front in 2011 at ~2, 5 km upstream. However, due to the limitations of the bedrock data (Sect. 3) and the lack of further evidence, the precise 2008 position remains unclear.

In 2011 and 2015 the glacier had markedly accelerated, indicating that the grounding line had further retreated. The 2011/2014 TSX/TDX dh/dt profile shows a slight drop in ice thinning rates at ~6 km upstream of the 1996 grounding line which coincides with the edge of the subglacial trough in the bedrock data. Although the change of elevation change rates is not as pronounced as on the other profiles, hydrostatic height anomalies along an OIB-flight path running close to this position (Fig. 6, Track 4) show that in 2014 the glacier was freely floating downstream. OIB data acquired in 2011 and 2014 (Fig. 6, Track 3 and 5) indicate that the ice is currently grounded upstream of the hill chain. We hence estimate the recent grounding line position to be located at the edge of the subglacial trough ~6 km upstream.

[Figure]

Figure S6 d shows that between 2007 and 2008 a pronounced acceleration is detected along Profile d (Fig. 6). In 2008 the glacier front had retreated behind the 1996 grounding line for the first time. Thus, the grounding line in 2008 must have been located upstream of the 1996 position in 2008. The velocity patterns look similar in 2008, 2011 and 2015, suggesting that no further grounding line retreat occurred after 2008. We hence assume that the 2008 grounding line position is more or less identical with the recent location. However, given the lack of further evidence the precise grounding line position of 2008 remains vague.

A sharp increase in surface velocities is visible at 3–4 km upstream in 2011 and 2015. This location is consistent with a pronounced change in ice thinning rates visible on the TSX/TDX 2011–2014 dh/dt profile and on the map (Fig. S7), which indicates that the ice is currently floating downstream. On the other hand, hydrostatic height anomalies along OIB tracks 4

and 5 in Figure 6 reveal that the ice is grounded further upstream. Additionally the 2011 front was located ~ 2 km upstream of the grounding line. Combining all of the information, we estimate the recent grounding line to be located at a position ~3–4 km upstream. However, the modeled bedrock data suggests that the grounding line is not situated on the upslope but on the downslope side of a subglacial hill. We hence cannot rule out that the hill may be shifted to the north in the modelled bedrock data, or that – if the modelled bedrock topography is correct - the recent grounding line is located ~ 1 km further to the south.

**S7 Recent grounding line and TSX/TDX 2011–2014 elevation change**

**Figure S7:** Glacier surface elevation change on Fleming Glacier between 2011 and 2014 derived from TSX/TDX bistatic and monostatic acquisitions with final solution of the   recent (2014) grounding line  (pink line). Brown line: grounding line in 1996 from Rignot et al. (2005) and Rignot et al. (2011). Background: Mosaic of two Landsat-8 „Natural Color" images, acquired on 2015-09-16 ©USGS.

[Figure]

S8 Evolution of the grounding line before 2008-today

Figure S8: Schematic drawing of the ungrounding history of Fleming Glacier from before 2008 to today. Interpretation is based on the data presented in this study and in the literature. Figures are not drawn to scale.

[Figure]